# Non-Stationary Dueling Bandits Under a Weighted Borda Criterion

**Joe Suk**                                                                                    *joe.suk@columbia.edu*
*Columbia University*

**Arpit Agarwal**\*                                                                            *agarpit@outlook.com*
*Columbia University*

**Reviewed on OpenReview:** *https://openreview.net/forum?id=XXXX*

## Abstract

In $K$-armed dueling bandits, the learner receives preference feedback between arms, and the regret of an arm is defined in terms of its suboptimality to a *winner* arm. The *non-stationary* variant of the problem, motivated by concerns of changing user preferences, has received recent interest (Saha & Gupta, 2022; Buening & Saha, 2023; Suk & Agarwal, 2023). The goal here is to design algorithms with low *dynamic regret*, ideally without foreknowledge of the amount of change.

The notion of regret here is tied to a notion of *winner arm*, most typically taken to be a so-called Condorcet winner or a Borda winner. However, the aforementioned results mostly focus on the Condorcet winner. In comparison, the Borda version of this problem has received less attention which is the focus of this work. We establish the first optimal and adaptive dynamic regret upper bound $\tilde{O}(\tilde{L}^{1/3}K^{1/3}T^{2/3})$, where $\tilde{L}$ is the unknown number of significant Borda winner switches.

We also introduce a novel *weighted Borda score* framework which generalizes both the Borda and Condorcet problems. This framework surprisingly allows a Borda-style regret analysis of the Condorcet problem and establishes improved bounds over the theoretical state-of-art in regimes with a large number of arms or many spurious changes in Condorcet winner. Such a generalization was not known and could be of independent interest.

## 1 Introduction

In $K$-armed *dueling bandits* problem, a learner relies on relative feedback between arms, as opposed to the reward feedback model of the more well-studied multi-armed bandit problem (see Sui et al., 2018; Bengs et al., 2021, for surveys). This problem has application in information retrieval, recommendation systems, etc, where relative feedback is easy to elicit, while real-valued feedback is difficult to obtain or interpret. For example, the availability of implicit user feedback comparing the output of two information retrieval algorithms allows one to automatically tune the parameters of these algorithms using the framework of dueling bandits (Radlinski et al., 2008; Liu, 2009).

Formally, at round $t \in [T]$, the learner pulls a *pair* of arms and observes *relative feedback* between these arms indicating which was preferred. The feedback is drawn randomly according to a pairwise mean preference matrix and the regret is measured in terms of the sub-optimality of choices to a 'winner' arm. Unlike classical MAB, there are different contending notions of winner arm in dueling bandits, and the underlying theory depends critically on the chosen notion. Most early work in dueling bandits considered the *Condorcet winner* (CW) (Urvoy et al., 2013; Ailon et al., 2014; Zoghi et al., 2014; 2015b; Komiyama et al., 2015) which is an arm that 'stochastically beats' every other arm. An alternative line of works focuses on the *Borda winner*

---

\*The author is currently at FAIR, Meta. Work done while the author was at Columbia University.

(BW), an arm maximizing the probability of defeating a uniformly at random chosen comparator (Urvoy et al., 2013; Jamieson et al., 2015; Ramamohan et al., 2016; Falahatgar et al., 2017; Lin & Lu, 2018; Heckel et al., 2018; Saha et al., 2021).

Both Borda and Condorcet carry their own notions of suboptimality/regret. While the Condorcet winner may not always exist, the Borda winner is always well-defined. On the other hand, the Borda winner may not satisfy the *independence of clones* property (Schulze, 2011), i.e. the Borda winner may change by adding identical copies of an arm to the model. Thus, it's difficult to prefer one winner notion over the other.[1]

In the more challenging *non-stationary* dueling bandit problem, preferences may change over time. Saha & Gupta (2022) first studied this problem and provided an algorithm that achieves a nearly optimal *Condorcet dynamic regret*[2] of $\tilde{O}(\sqrt{S_{\mathrm{C}}KT})$, where $S_{\mathrm{C}}$ is the number of changes in Condorcet Winner. For the Borda problem, they showed a dynamic regret bound of $\tilde{O}(S_{\mathrm{B}}^{1/6}K^{-1/3}T^{5/6} + S_{\mathrm{B}}^{1/2}K^{1/3}T^{2/3})$ in terms of $S_{\mathrm{B}}$ switches in Borda winner. While the Condorcet rate of $\sqrt{S_{\mathrm{C}}KT}$ is minimax optimal, up to log terms, the reported Borda rate is suboptimal as confirmed by our lower bound (Theorem 7). Furthermore, the aforementioned procedures (for both winner settings) are not *adaptive*, requiring knowledge of the number of switches. Moreover, the dependence on $S_{\mathrm{C}}, S_{\mathrm{B}}$ is pessimistic, as these may count insignificant changes (e.g., if there are $\Omega(T)$ spurious changes in CW) which do not properly capture the difficulty of non-stationarity.

Addressing these limitations, Buening & Saha (2023) propose for the CW setting a notion of *significant CW switches* (only counting those switches in CW which truly pose challenging non-stationarity). Under the classical strong stochastic transitivity (SST) and stochastic triangle inequality (STI) conditions on the preferences[3] (Assumption 1), they achieve a parameter-free dynamic regret bound of $\tilde{O}(K\sqrt{\tilde{S}_{\mathrm{C}}T})$ in terms of $\tilde{S}_{\mathrm{C}}$ such switches. A subsequent work (Suk & Agarwal, 2023) achieved the optimal dependence on $K$ of $\tilde{O}(\sqrt{\tilde{S}_{\mathrm{C}}KT})$. Naturally, one may ask whether an analogous result is possible for the Borda problem, leading to our first question (answered in the affirmative by our Theorem 2):

**Question #1.** *Can we attain adaptive and optimal Borda dynamic regret in terms of a "Borda notion" of significant winner switches?*

Now, turning back to the Condorcet problem, we again note that the $\tilde{O}(\sqrt{\tilde{S}_{\mathrm{C}}KT})$ rate was only established under SST and STI, which assumes a linear ordering on arms and monotonicity/transitivity conditions on preferences. While such assumptions are well-studied in earlier dueling bandit works (Yue & Joachims, 2011; Yue et al., 2012), they're arguably unrealistic as arm preferences may not even be totally ordered in application (e.g., cyclic preferences in a tournament). Despite this, Suk & Agarwal (2023) showed the $\sqrt{\tilde{S}_{\mathrm{C}}KT}$ rate is in fact unattainable without SST∩STI.

Buening & Saha (2023) show, outside of SST∩STI, a dynamic regret upper bound of $\tilde{O}(K\sqrt{ST})$ in terms of the coarser count $S \geq \tilde{S}_{\mathrm{C}}$ of total CW switches. However, when compared to the optimal stationary regret rate $\sqrt{KT}$, it's unclear if the dependence of $K$ in this result is optimal and also whether an *intermediate notion of significant non-stationarity* (counting between $\tilde{S}_{\mathrm{C}}$ and $S$ switches in CW) can be learned adaptively. In other words, one hopes for a regret rate of $\tilde{O}(\sqrt{S_{\mathrm{int}}KT})$ in terms of some count $S_{\mathrm{int}} \in [\tilde{S}_{\mathrm{C}}, S]$ capturing the learnable significant non-stationarity, thus resolving the gap in the findings of Suk & Agarwal (2023) and Buening & Saha (2023).

**Question #2.** *Can we learn a count of enhanced changes for switching Condorcet dueling bandits and improve the $K\sqrt{ST}$ rate?*

In fact, we argue this is a difficult problem. Even in the stationary dueling bandit problem, the minimax optimal regret rate of $\sqrt{KT}$ is only known to be achievable using a sparring reduction to adversarial bandit algorithms (Dudik et al., 2015; Balsubramani et al., 2016; Saha & Gaillard, 2022). On the other hand, the adaptive procedures of Buening & Saha (2023); Suk & Agarwal (2023) crucially rely on stochastic elimination-style algorithms. Thus, it's unclear how to simultaneously attain a sharp dependence on $K$ and handle unknown non-stationarity.

---

[1] The celebrated Arrow's impossibility theorem from social choice theory (Arrow, 1950) shows no single winner notion satisfies a class of "reasonable" axiomatic requirements.

[2] measured to a time-varying sequence of Condorcet winners.

[3] These assume the preference matrix admits a total ordering on arms which obey monotonicity and transitivity conditions.

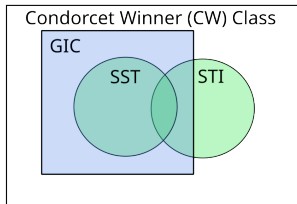

Figure 1: GIC vs. SST, STI.

Our second contribution is to make progress in answering the above question. In particular, we introduce a new count $S_{\mathrm{approx}}$ of *approximate CW changes* and show an adaptive Condorcet regret bound of $S_{\mathrm{approx}}^{1/3}K^{2/3}T^{2/3}$ under a weaker *general identifiability condition* (GIC) (Condition 1). GIC mandates that there's an arm stochastically dominating any other arm with the largest margin, and is weaker than SST∩STI (see Figure 1 for comparison). This yields an improvement over the $K\sqrt{ST}$ rate of Buening & Saha (2023) under GIC when the number of arms $K$ is large or when $S_{\mathrm{approx}} \ll S$.

Both our discussion for non-stationary Borda and Condorcet dueling bandits can also go through an alternative non-stationarity measure $V_T$ measuring the *total variation* of preferences over time. For the Borda setting, we give the first optimal dynamic regret bound in terms of $V_T$ (Theorem 2). For the Condorcet setting, we give the first adaptive dynamic regret bound, which was only shown previously under the restrictive SST∩STI assumption (Buening & Saha, 2023; Suk & Agarwal, 2023).

Both our answers to Questions #1 and 2 involve a new unified regret minimization framework, called *weighted Borda scores*, for dueling bandits. This framework generalizes both the Condorcet and Borda problems and, in particular, allows for the Condorcet regret to be recast as a Borda-like regret. Such a relationship was not known before and may be of independent interest. We note that, within this framework, showing a Condorcet regret bound involves additional complications beyond the Borda-style regret analysis as the *reference weight* (a notion generalizing the uniform weight of the Borda score; cf. Section 6) on arms is unknown and must be efficiently estimated. Further details on the challenges of this Condorcet regret analysis are found in Section 7.

A more expansive account of related works on dueling bandits is deferred to Appendix J.

## 1.1 Tabular Summary of Contributions

| Notation | Non-Stationarity Measure |
|:---:|:---:|
| $S_{\mathrm{B}}$ | Borda winner switches |
| $\tilde{S}_{\mathrm{B}}$ | Significant BW switches (Definition 1) |
| $S_{\mathrm{C}}$ | Condorcet winner switches |
| $\tilde{S}_{\mathrm{C}}$ | Significant CW switches (Buening & Saha, 2023) |
| $S_{\mathrm{approx}}$ | Approximate CW switches (Definition 4) |
| $V_T$ | Total Variation in preferences (Saha & Gupta, 2022) |

Figure 2: Glossary of Non-Stationarity Measures

## 2 Setup – Non-stationary Dueling Bandits

We consider $K$-armed dueling bandits with horizon $T$. At round $t \in [T]$, the pairwise preference matrix is denoted by $\mathbf{P}_t \in [0,1]^{K \times K}$, where $(i,j)$-th entry $P_t(i,j)$ encodes the likelihood of observing a preference for arm $i$ in a direct comparison with arm $j$. In the *stationary* dueling bandit problem, the preference matrices $\mathbf{P}_t \equiv \mathbf{P}$ are unchanging in time, whereas in the *non-stationary* problem, the preference matrices $\mathbf{P}_t$ may change arbitrarily from round to round. At round $t$, the learner selects a pair of actions $(i_t, j_t) \in [K] \times [K]$ and observes the feedback $O_t(i_t, j_t) \sim \mathrm{Ber}(P_t(i_t, j_t))$ where $P_t(i_t, j_t)$ is the underlying preference of arm $i_t$ over $j_t$. We next outline the two main formulations of dueling bandits (Borda and Condorcet).

| | Model | Regret Upper Bound | Adaptive? |
|---|---|---|---|
| **Borda** | | | |
| Saha & Gupta (2022) | None | $S_{\mathrm{B}}^{1/6}K^{-1/3}T^{5/6} + S_{\mathrm{B}}^{1/2}K^{1/3}T^{2/3}$ | No |
| Saha & Gupta (2022) | None | $V_T^{1/8}K^{-3/8}T^{7/8} + V_T^{7/8}K^{5/24}T^{19/24}$ | No |
| Our Work (Theorem 2) | None | $\min\{\tilde{S}_{\mathrm{B}}^{1/3}K^{1/3}T^{2/3}, V_T^{1/4}K^{1/4}T^{3/4} + K^{1/3}T^{2/3}\}$ | Yes |
| Lower Bound (Theorems 7 and 8) | None | $\min\{\tilde{S}_{\mathrm{B}}^{1/3}K^{1/3}T^{2/3}, V_T^{1/4}K^{1/4}T^{3/4} + K^{1/3}T^{2/3}\}$ | |
| **Condorcet** | | | |
| Saha & Gupta (2022) | C.W. | $\sqrt{S_{\mathrm{C}}KT}$ | No |
| Saha & Gupta (2022) | C.W. | $V_T^{1/3}K^{1/3}T^{2/3} + \sqrt{KT}$ | No |
| Buening & Saha (2023) | SST∩STI | $\min\{K\sqrt{\tilde{S}_{\mathrm{C}}T}, V_T^{1/3}(KT)^{2/3} + K\sqrt{T}\}$ | Yes |
| Suk & Agarwal (2023) | SST∩STI | $\min\{\sqrt{\tilde{S}_{\mathrm{C}}KT}, V_T^{1/3}K^{1/3}T^{2/3} + \sqrt{KT}\}$ | Yes |
| Buening & Saha (2023) | C.W. | $K\sqrt{S_{\mathrm{C}}T}$ | Yes |
| Our Work (Theorem 5) | GIC | $\min\{S_{\mathrm{approx}}^{1/3}K^{2/3}T^{2/3}, V_T^{1/4}K^{1/2}T^{3/4} + K^{2/3}T^{2/3}\}$ | Yes |
| Lower Bound (Saha & Gupta, 2022) | C.W. | $\min\{\sqrt{S_{\mathrm{C}}KT}, V_T^{1/3}K^{1/3}T^{2/3} + \sqrt{KT}\}$ | |

Table 1: Comparison of dynamic regret upper bounds. See Subsection 5.2 for a detailed comparison of our $K^{2/3}S_{\mathrm{approx}}^{1/3}T^{2/3}$ rate vs. the $K\sqrt{S_{\mathrm{C}}T}$ rate of Buening & Saha (2023). See Figure 1 for a comparison of the model classes CW, GIC, SST∩STI. Our regret upper bound for the Borda model is tight (Theorem 7).

**Borda Criterion.** Let $b_t(a) \doteq \frac{1}{K}\sum_{a'\in[K]} P_t(a,a')$ be the **Borda score** of arm $a$. Then, $a_t^{\mathrm{B}} \doteq \mathrm{argmax}_{a\in[K]} b_t(a)$ is the **Borda winner** (BW) at round $t$. Then, the **Borda dynamic regret** is

$$\mathrm{Regret}^{\mathrm{B}} \doteq \sum_{t=1}^{T} b_t(a_t^{\mathrm{B}}) - \frac{1}{2}\left(b_t(i_t) + b_t(j_t)\right). \tag{1}$$

We let $\delta_t^{\mathrm{B}}(a) \doteq b_t(a_t^{\mathrm{B}}) - b_t(a)$ be the instantaneous dynamic Borda regret of arm $a$ at round $t$.

**Condorcet Criterion.** Here, one assumes the existence of a **Condorcet winner** (CW) $a_t^{\mathrm{C}}$ such that $P_t(a_t^{\mathrm{C}},a) \geq 1/2$ for all arms $a \in [K]$. Then, the *Condorcet dynamic regret* is defined as

$$\mathrm{Regret}^{\mathrm{C}} \doteq \sum_{t=1}^{T} \frac{P_t(a_t^{\mathrm{C}},i_t) + P_t(a_t^{\mathrm{C}},j_t) - 1}{2}.$$

## 2.1 Non-Stationarity Measures

Following the discussion in Section 1, key in works on non-stationary (dueling) bandits is a measure of non-stationarity which captures the difficulty of the problem. Speaking plainly, the higher the amount of non-stationarity the more difficult the problem is, and so larger regret rates are expected. It is thus crucial that such a measure of change properly captures the difficulty of the problem. However, the only other work dealing with Borda dynamic regret (Saha & Gupta, 2022) relies on coarse non-stationarity measures, such as the aggregate number $L \doteq \sum_{t=2}^{T} \mathbf{1}\{\mathbf{P}_t \neq \mathbf{P}_{t-1}\}$ or magnitude $V_T \doteq \sum_{t=2}^{T} \max_{a,a'\in[K]} |P_t(a,a') - P_{t-1}(a,a')|$ of changes (see Figure 2).

To contrast, in non-stationary MAB and Condorcet dueling bandits, it is now recognized (Suk & Kpotufe, 2022; Buening & Saha, 2023; Suk & Agarwal, 2023) that tighter non-stationary measures, so-called *significant shifts*, which capture only those changes which are detrimental to performance can in fact be adaptively learned. The main intuition behind a significant shift is that, for any notion of regret (e.g., MAB, Condorcet, Borda), one can track when each arm accrues significant regret (i.e., surpassing the minimax rate) and define a significant shift as occurring when every arm has accrued significant regret. Thus, our Definition 1 has a similar form as Definition 1 of Suk & Kpotufe (2022) or as the corresponding notion for Condorcet winner dueling bandits (Buening & Saha, 2023). The key difference here is that we use the Borda notion of regret and compare to the Borda minimax regret rate ($T^{2/3}$ vs. $T^{1/2}$).

**Definition 1.** *Define an arm $a$ as having* **significant Borda regret** *on $[s_1, s_2]$ if*

$$\sum_{s=s_1}^{s_2} \delta_s^B(a) \geq K^{1/3} \cdot (s_2 - s_1)^{2/3}. \tag{2}$$

*Define* **significant BW switches** *(abbrev. SBS) as follows: let $\tau_0 = 1$ and the $(i+1)$-th sig. shift $\tau_{i+1}$ is recursively defined as the smallest $t > \tau_i$ such that for each arm $a \in [K]$, $\exists [s_1, s_2] \subseteq [\tau_i, t]$ such that arm $a$ has significant regret over $[s_1, s_2]$. We refer to the interval of rounds $[\tau_i, \tau_{i+1})$ as an* **SBS phase**[4]. *Let $\tilde{S}_B$ be the number of SBS phases elapsed in $T$ rounds. By convention, let $\tau_{\tilde{S}_B+1} \doteq T + 1$.*

**Remark 1.** *If there is a fixed Borda winner $a_t^B \equiv a^B$, then it does not incur significant regret. Thus, the number $\tilde{S}_B$ of SBS is smaller than the number $S_B$ of Borda winner switches, meaning Definition 1 is a tighter measure of non-stationarity.*

**Notation.** *As is common in works on non-stationary bandits (see Appendix J), we'll often conflate counts of changes (e.g., $L, S_B$) with the* **number of phases induced**, *appearing in regret rates, which is always one greater than the corresponding count. Note that this only affects constants in the regret rates.*

## 3   Dynamic Regret Lower Bounds

We briefly summarize the minimax dynamic regret rates for the Borda and Condorcet problems. Full statements and proofs are deferred to Appendices B and C. We note the arguments are fairly standard and mostly follow previous lower bound constructions for Borda (Theorem 16 Saha et al., 2021) and non-stationary dueling bandits (Theorems 5.1, 5.2 Saha & Gupta, 2022).

• **Borda Dynamic Regret**   As the minimax Borda regret rate over $n$ stationary rounds is $K^{1/3}n^{2/3}$ (Saha et al., 2021), it follows that the minimax dynamic regret rate over $L$ stationary phases of length $T/L$ is $LK^{1/3}(T/L)^{2/3} = K^{1/3}T^{2/3}L^{1/3}$ (Theorem 7). In fact, the number of SBS phases (Definition 1) may replace $L$ here. To our knowledge, this establishes the first lower bound on Borda dynamic regret.

• **Condorcet Dynamic Regret**   Here, since the minimax regret rate over stationary problems is of order $\sqrt{KT}$, the lower bound on Condorcet dynamic regret is of order $\sqrt{LKT}$ (Theorem 5.1; Saha & Gupta, 2022). Once again, $L$ here may be replaced by a tighter measure of non-stationarity (Suk & Agarwal, 2023).

## 4   Algorithmic Design

To minimize the weighted Borda dynamic regret, we rely on estimating $b_t(a, \mathbf{w})$ for weights $\mathbf{w}$ belonging to a *reference set of weights $\mathcal{W}$*. In what follows, we'll discuss the procedures in terms of a fixed $\mathcal{W}$. In the end, $\mathcal{W}$ can be flexibly specialized for each of the Borda and Condorcet regret problems (see Definition 4).

Following the high-level idea of prior works on non-stationary dueling bandits (Suk & Agarwal, 2023; Buening & Saha, 2023), we'll first design a base algorithm which works well in mildly non-stationary environments where there is no approximate winner change (Definition 4) and then randomly schedule different instances of this base algorithm to allow for detection of unknown non-stationarity.

One key deviation from the aforementioned prior non-stationary works is that we avoid using a successive elimination base algorithm. This is necessary since accurate estimation of the (weighted) Borda score $b_t(a, \mathbf{w})$, calls for some baseline uniform exploration, which rules out any hard elimination strategy.

Thus, we first design a new base algorithm, which mixes elimination with time-varying exploration.

### 4.1   Base Algorithm – Soft Elimination with WBS

**Estimating Weighted Borda Scores.**   At round $t$, we estimate the weighted Borda scores via importance-weighting for weight $\mathbf{w} \in \mathcal{W}$:

$$\hat{b}_t(a, \mathbf{w}) \doteq \mathop{\mathbb{E}}_{a' \sim \mathbf{w}} \left[ \frac{\mathbf{1}\{i_t = a, j_t = a'\} \cdot O_t(i_t, j_t)}{q_t(a) \cdot q_t(a')} \right], \tag{3}$$

---

[4]We conflate intervals $[a, b), [a, b]$ for $a, b \in \mathbb{N}$ with the set of natural numbers contained therein.

where $q_t$ is the play-distribution over arms at time $t$. Let $\hat{\delta}_t^{\mathbf{w}}(a', a) \doteq \hat{b}_t(a', \mathbf{w}) - \hat{b}_t(a, \mathbf{w})$ denote the induced estimator of the weighted Borda gap.

**Eliminating Arms** We evict arm $a$ from *the candidate arm set* $\mathcal{A}_t$ *at round* $t$ if for some $[s_1, s_2] \subseteq [1, t]$:

$$\max_{a' \in \cap_{s=s_1}^{s_2} \mathcal{A}_s} \sum_{s=s_1}^{s_2} \hat{\delta}_s^{\mathbf{w}}(a', a) \geq C \log(T) \cdot F([s_1, s_2]). \tag{4}$$

In the above $C > 0$ is a universal constant free of any problem-dependent parameters whose value can be determined from the analysis and the *eviction threshold* $F(I)$ for interval $I$ is determined using the estimation error bounds on $\sum_{s=s_1}^{s_2} \hat{b}_s(a, \mathbf{w})$ (see Definition 4 in Appendix E for exact formulas for $F(I)$).

These error bounds scale with the inverse play-probability $\min_a q_t^{-1}(a)$, calling for a careful design of the play distribution $q_t(\cdot)$, namely that which balances exploration and proper elimination.

**Novel Exploration Schedule.** We choose $q_t$ as a mixture of exploring the candidate set $\mathcal{A}_t$ and uniformly exploring all arms:

$$q_t \sim (1 - \eta_t) \cdot \text{Unif}\{\mathcal{A}_t\} + \eta_t \cdot \text{Unif}\{\mathcal{W}\}, \tag{5}$$

where $\text{Unif}\{\mathcal{W}\}$ is a further uniform mixture of playing according to the distributions in $\mathcal{W}$. The *learning rate* $\eta_t$ must then be carefully set to ensure both sufficient exploration (to reliably estimate the WBS) and safe regret. As before, the values of $\eta_t$ (see Definition 4) will depend on the Borda vs. Condorcet setting.

However, prior works on stationary Borda dueling bandits must set $\eta_t$ using knowledge of the horizon $T$ (Jamieson et al., 2015; Saha et al., 2021; Wu et al., 2023). In our non-stationary problem, this amounts to setting $\eta_t$ based on knowledge of the underlying non-stationarity (Saha & Gupta, 2022).

Thus, a key difficulty in targeting the regret rate of Theorem 2 *without knowledge of non-stationarity* is that the optimal oracle learning rate $\eta_t$ depends on the unknown phase lengths ($\tau_{i+1} - \tau_i$ for Borda scores; $\rho_{i+1} - \rho_i$ for Condorcet). To circumvent this, we employ a *time-varying learning rate* $\eta_t$ which depends on the current number of rounds elapsed. To our knowledge, such an idea has only been used in works on adversarial bandit for analyzing EXP3 (Seldin et al., 2013; Maillard, 2011).

### 4.2 Non-Stationary Meta-Algorithm

---

**Algorithm 1:** BOSSE($t_{\text{start}}, m_0$): (Weighted) **BO**rda **S**core **S**oft **E**limination

---

**Input:** Input set of weights $\mathcal{W}$, learning rate profile $\boldsymbol{\gamma} \doteq \{\gamma_t\}_{t=1}^T$, eviction threshold $F(\cdot)$.

**1 Initialize:** $t \leftarrow t_{\text{start}}$, active arm set $\mathcal{A}_{t_{\text{start}}} \leftarrow [K]$.

**2 while** $t \leq T$ **do**

**3**      **Set exploration rate for this round**: $\eta_t \leftarrow \gamma_{t-t_{\text{start}}}$.

**4**      **Update play distribution** $q_t$: let $q_t \in \Delta^K$ be the mixture $(1 - \eta_t) \cdot \text{Unif}\{\mathcal{A}_t\} + \eta_t \cdot \text{Unif}\{\mathcal{W}\}$.

**5**      Sample $i_t, j_t \sim q_t$ i.i.d..

**6**      Receive feedback $O_t(i_t, j_t) \sim \text{Ber}(P_t(i_t, j_t))$.

**7**      Increment $t \leftarrow t + 1$.

**8**      **Evict arms with large weighted Borda gaps:**

         $\mathcal{A}_t \leftarrow \mathcal{A}_t \setminus \big\{ a \in [K] : \exists [s_1, s_2] \subseteq [t_{\text{start}}, t], \mathbf{w} \in \mathcal{W} \text{ s.t. (4) holds with } a \in \cap_{s=s_1}^{s_2} \mathcal{A}_s \big\}$ ;

         $\mathcal{A}_{\text{global}} \leftarrow \mathcal{A}_{\text{global}} \setminus \big\{ a \in [K] : \exists [s_1, s_2] \subseteq [t_\ell, t], \mathbf{w} \in \mathcal{W} \text{ s.t. (4) holds with } a \in \cap_{s=s_1}^{s_2} \mathcal{A}_s \big\}$ ;

**9**

**10**      **if** $\exists m$ such that $B_{t,m} > 0$ **then**            /* Lines 10-13 only for use with METABOSSE */

**11**          Let $m \doteq \max\{m \in \{2, 4, \ldots, 2^{\lceil \log(T) \rceil}\} : B_{t,m} > 0\}$. ;      // Set maximum replay length.

**12**          Run BOSSE($t, m$). ;                     // Replay interrupts.

**13**      **Restart criterion: if** $\mathcal{A}_{\text{global}} = \emptyset$ **then** RETURN.;

**14**      **if** $t > t_{\text{start}} + m_0$ **then** RETURN. ;

---

---

**Algorithm 2:** Meta-BOSSE

---

**Input:** horizon $T$, input set of weights $\mathcal{W}$, learning rate profile $\boldsymbol{\gamma} \doteq \{\gamma_t\}_{t=1}^{T}$, eviction threshold $F(\cdot)$.

**1 Initialize:** round count $t \leftarrow 1$.

**2 Episode Initialization (setting global variables $t_\ell, \mathcal{A}_{\text{global}}, B_{s,m}$):**

**3**      $t_\ell \leftarrow t$. ;                                                    // $t_\ell$ indicates start of $\ell$-th episode.

**4**      $\mathcal{A}_{\text{global}} \leftarrow [K]$ ;                                                       // Global active arm set.

**5**      For each $m = 2, 4, \ldots, 2^{\lceil \log(T) \rceil}$ and $s = t_\ell + 1, \ldots, T$:

**6**         Sample and store $B_{s,m} \sim \text{Bernoulli}\left(\frac{1}{m^{1/3} \cdot (s-t_\ell)^{2/3}}\right)$. ;                   // Set replay schedule.

**7** Run $\text{BOSSE}(t_\ell, T + 1 - t_\ell)$.

**8 if** $t < T$ **then** restart from Line 2 (i.e. start a new episode). ;

---

For the non-stationary setting, we use a hierarchical algorithm METABOSSE (Algorithm 2) to schedule multiples copies of the base algorithm $\text{BOSSE}(t_{\text{start}}, m)$ (Algorithm 1) at random times $t_{\text{start}}$ and durations $m$.

Going into more detail, METABOSSE proceeds in *episodes*, starting each episode by running a starter instance of BOSSE. A running base algorithm may further *activates* its own base algorithms of varying durations (Line 12 of Algorithm 1), called *replays* according to a random schedule decided by the Bernoulli's $B_{s,m}$ (see Line 6 of Algorithm 2). We refer to the base algorithm playing at round $t$ as the *active base algorithm*.

The *candidate arm set* $\mathcal{A}_t$ is pruned by the active base algorithm at round $t$, and globally shared between all running base algorithms. In addition, all other variables, i.e. the $\ell$-th episode start time $t_\ell$, round count $t$, and replay schedule $\{B_{s,m}\}_{s,m}$, are shared between base algorithms.

In sharing these global variables, any replay can trigger a new episode: every time an arm is evicted by a replay, it is also evicted from the *global arm set* $\mathcal{A}_{\text{global}}$, essentially the active arm set for the entire episode. A new episode is triggered when $\mathcal{A}_{\text{global}}$ becomes empty, i.e., there is no *safe* arm left to play.

For further intuition on the hierarchical schedule and management of base algorithms, we defer the reader to Section 4 of Suk & Kpotufe (2022). We focus here on highlighting the novelties of this algorithm.

**Each Active Base Alg. Chooses Own Learning Rate**     Each base algorithm $\text{BOSSE}(t_{\text{start}}, m)$ determines its own time-varying learning rate using its starting round $t_{\text{start}}$. Then, the *global learning rate* $\eta_t$ at round $t$ is set by the base algorithm active at round $t$ (Line 3 of Algorithm 1). Furthermore, the history of global learning rates $\{\eta_t\}_t$ (globally accessible by any base algorithm) is used to determine the eviction thresholds (16) and (17) over intervals $[s_1, s_2]$ of rounds where multiple base algorithms may be active.

This ensures reliable detection of critical segments $[s_1, s_2]$ of time where an arm has large (weighted) Borda regret. For such a critical segment, a core argument of the analysis is that an ideal replay, i.e., an instance of $\text{BOSSE}(s_1, m)$ for $m = s_2 - s_1$, is scheduled with high probability. However, such an ideal replay must be scheduled for a duration commensurate with $s_2 - s_1$, and hence must also use a commensurate learning rate.

**A Different Replay Scheduling Rate.**     In order to properly detect critical segments while also safeguarding $T^{2/3}$ regret, a different replay scheduling rate (Line 12 of Algorithm 1) is required. For this, we find that instantiating a base algorithm of length $m$ at time $t$ in episode $[t_\ell, t_{\ell+1})$ with probability $m^{-1/3} \cdot (s - t_\ell)^{-2/3}$ balances regret minimization and detection of changes in winner.

**Remark 2.** *For the Borda setting, the total complexity of our algorithm is $O(KT^2)$ versus $O(KT)$ for DEX3 (Saha & Gupta, 2022), but the latter algorithm requires knowledge of non-stationarity. For the Condorcet setting, we have a complexity of $O(K^2 T^2)$ which matches that of ANACONDA (Buening & Saha, 2023), the only other adaptive algorithm with regret guarantees for Condorcet winner. We emphasize, regardless of setting, it remains open whether higher runtime can be avoided while getting adaptive and optimal regret.*

# 5 Dynamic Regret Upper Bounds

## 5.1 Borda Dueling Bandits

Our goal then is to establish a dynamic regret upper bound which depends optimally on the number of SBS, and does not require knowledge of the underlying non-stationarity.

**Adaptive and Optimal Regret Upper Bound.** Intuitively, if one knows the SBS $\tau_i$, then, in each SBS phase $[\tau_i, \tau_{i+1})$, one can achieve a tight regret bound of order $K^{1/3} \cdot (\tau_{i+1} - \tau_i)^{2/3}$ by learning the *last safe arm*, or the last arm to incur significant Borda regret in said phase. We show that in fact such a rate can be attained, over all SBS phases, without any knowledge of the non-stationarity.

**Theorem 2.** *Algorithm 2 with the fixed weight specification (see Definition 4) satisfies:*

$$\mathbb{E}[\mathrm{Regret}^{\mathrm{B}}] \leq \tilde{O}\left(\sum_{i=0}^{\tilde{S}_{\mathrm{B}}} K^{1/3} \cdot (\tau_{i+1} - \tau_i)^{2/3}\right).$$

**Corollary 3.** *By Jensen's inequality, the regret bound of Theorem 2 is upper bounded by $K^{1/3} \cdot T^{2/3} \cdot \tilde{S}_{\mathrm{B}}^{1/3}$. Furthermore, relating SBS to total variation $V_T$, a regret bound of order $V_T^{1/4} \cdot T^{3/4} \cdot K^{1/4} + K^{1/3} \cdot T^{2/3}$ also holds (see Appendix H.5).*

In particular, the upper bound of Theorem 2 matches the lower bound of Section 3 (see Theorem 7). The $K^{1/3}T^{2/3}\tilde{S}_{\mathrm{B}}^{1/3}$ rate also improves the previously best-known upper bound (Saha & Gupta, 2022, Theorem 6.1), which was suboptimal, non-adaptive, and relied on a coarser notion of non-stationary (i.e., counting all changes in Borda winner).

## 5.2 Condorcet Dueling Bandits

**Significant Notions of Change Outside of** SST∩STI**.** Following the discussion of Section 1, it was previously shown in Suk & Agarwal (2023) that an analogue of Definition 1 for Condorcet winner switches can't be learned adaptively outside the SST and STI preference model assumptions.

**Assumption 1.** *There is a total ordering on arms, $\succ_t$, such that $\forall i \succeq_t j \succeq_t k$:*

- $i \succeq_t j \iff P_t(i, j) \geq 1/2$.

- $P_t(i, k) \geq \max\{P_t(i, j), P_t(j, k)\}$ *(SST).*

- $P_t(i, k) \leq P_t(i, j) + P_t(j, k)$ *(STI).*

In particular, such conditions are convenient for relating uncompared arms through an inferred ordering, and turn out to be fundamental to detecting unknown changes in CW.

However, this impossibility result leaves open whether *other tighter notions of non-stationarity* can be learned outside of SST∩STI, and at rates faster than the state-of-the-art $K\sqrt{S_{\mathrm{C}}T}$ Condorcet dynamic regret, in terms of $S_{\mathrm{C}}$ changes in winner, achieved by Buening & Saha (2023).

We show this is indeed possible in a broad class of preference models, called *GIC*, outside of SST∩STI.

**Condition 1.** (**G***eneral* **I***dentifiability* **C***ondition*) *At each round $t$, there exists an arm $a_t^*$ such that $a_t^* \in \mathrm{argmax}_{a \in [K]} P_t(a, a')$ for all arms $a' \in [K]$.*

While GIC requires the CW to beat every other arm with the largest margin, it is far broader than SST∩STI in not requiring any ordering on non-winner arms, allowing for cycles or arbitrary preference relations. The GIC was previously studied in utility dueling bandits by Zimmert & Seldin (2018) (see also Bengs et al., 2021, Section 3.1). Unlike these prior works, we do not require the winner arm $a_t^*$ to be unique.

We also note that Condition 1 implies the Condorcet condition, but is weaker than SST while incomparable with STI. See Figure 1 for a full comparison.

We next show that, under GIC, a tighter notion of non-stationarity than the count $S_C$ of winner switches can be learned with improved regret rates. In particular, we count changes in *approximate winner*, which are arms exhibiting similar preferences to other arms as the true winner. The threshold for similarity is determined by the minimax regret rate which in this case is $K^{2/3}T^{2/3}$ (Theorem 11). We count an approximate winner change only when there is no longer a stable approximate winner.

**Definition 4** (Approximate Winner Changes). *Define $\zeta_i$ recursively as follows: let $\zeta_0 = 1$ and let $\zeta_{i+1}$ be the smallest $t > \zeta_i$ such that there does not exist an* **approximate winner arm** $\tilde{a}$ *such that*

$$\forall s \in [\zeta_i, t], a \in [K] : |P_s(a_s^*, a) - P_s(\tilde{a}, a)| \leq \left( \frac{K^2}{s - \zeta_i} \right)^{1/3}. \tag{6}$$

*Let $S_{\text{approx}} \doteq 1 + \max\{i : \zeta_i < T\}$ be the total count of approximate winner phases.*

Note that (6) always holds so long as the winner $a_t^*$ does not change. Thus, we have $S_{\text{approx}} \leq S$.

**Example 1** (Environment where $S_{\text{approx}} = 1$, but $S = \Omega(T)$). *Consider a non-stationary environment with three arms $1, 2, 3$ whose preference structure alternates between the two matrices:*

$$\boldsymbol{P}^+ := \begin{pmatrix} 1/2 & 1/2 + T^{-1/3} & 1 \\ 1/2 - T^{-1/3} & 1/2 & 1/2 + T^{-1/3} \\ 0 & 1/2 - T^{-1/3} & 1/2 \end{pmatrix}, \boldsymbol{P}^- := \begin{pmatrix} 1/2 & 1/2 - T^{-1/3} & 0 \\ 1/2 + T^{-1/3} & 1/2 & 1/2 - T^{-1/3} \\ 1 & 1/2 + T^{-1/3} & 1/2 \end{pmatrix}.$$

*Then, under environment $\boldsymbol{P}^+$ (resp. $\boldsymbol{P}^-$), the arms are ordered as $1 \succ 2 \succ 3$ (resp. $3 \succ 2 \succ 1$). In either environment, arm 2 is an approximate winner arm meaning the $T$-round non-stationary environment has $S = \Omega(T)$ winner switches but no approximate winner switches.*

**Condorcet Dynamic Regret Bound.** Our regret upper bound is then as follows.

**Theorem 5.** *Under Condition 1, Algorithm 2 with unknown weight specification (see Definition 4) satisfies:*

$$\mathbb{E}[\text{Regret}^C] \leq \tilde{O}(\min\{K^{2/3} \cdot T^{2/3} \cdot S_{\text{approx}}^{1/3}, K^{1/2} \cdot V_T^{1/4} \cdot T^{3/4} + K^{2/3} \cdot T^{2/3}\}).$$

We now compare this regret bound to the $K\sqrt{ST}$ rate of Buening & Saha (2023). In particular, we have:

$$\min\{K^{\frac{2}{3}}T^{\frac{2}{3}}S_{\text{approx}}^{\frac{1}{3}}, K^{\frac{1}{2}}V_T^{\frac{1}{4}}T^{\frac{3}{4}} + K^{\frac{2}{3}}T^{\frac{2}{3}}\} \leq K\sqrt{ST} \iff K \geq \min\left\{ \frac{T^{\frac{1}{2}}S_{\text{approx}}}{S^{\frac{3}{2}}}, \max\left\{ \frac{T^{\frac{3}{2}}V_T^{\frac{1}{2}}}{S}, \frac{T^{\frac{1}{2}}}{S^{\frac{3}{2}}} \right\} \right\}. \tag{7}$$

We highlight two regimes, under GIC, where this comparison is favorable to our new rate:

- **Many spurious winner changes** $S = T$: if there are $S = T$ changes in winner, then (7) always holds regardless of the values of $V_T, S_{\text{approx}}, K, T$ and thus captures regimes where sublinear regret is possible while the $K\sqrt{ST}$ rate is vacuous. The superiority of our regret rate is most evident when $S_{\text{approx}}$ or $V_T$ are small, which is possible if the majority of $S = T$ winner changes are spurious to performance.

- **Large number of arms**: viewed another way, (7) states that if the number of arms $K$ is larger than some threshold determined by the discrepancy in the non-stationarity measures $S$ vs. $S_{\text{approx}}$ or $S$ vs $V_T$, then Theorem 5's regret rate is superior to the $K\sqrt{ST}$ rate.

We also note that the $K^{1/2} \cdot V_T^{1/4} \cdot T^{3/4}$ rate of Theorem 5 is the first adaptive Condorcet dynamic regret bound in terms of total variation, as prior works only achieved regret upper bounds under SST∩STI (Buening & Saha, 2023; Suk & Agarwal, 2023).

## 6 A New Unified View of Condorcet and Borda Regret

Continuing the discussion of Subsection 1.1, our regret upper bounds (Theorems 2 and 5) are shown by appealing to a new framework which generalizes the Borda and Condorcet regret minimization tasks.

Key to this is the new idea of a *weighted Borda score* (WBS), which measures the preference of an arm $a$ over a reference distribution or *weight* of arms $\mathbf{w} \in \Delta^K$, where $\Delta^K$ denotes the probability simplex on $[K]$. More precisely, we define a WBS with respect to weight $\mathbf{w} = (w^1, \ldots, w^K)$ as

$$b_t(a, \mathbf{w}) \doteq \sum_{a' \in [K]} P_t(a, a') \cdot w^{a'}.$$

Notions of *weighted Borda winner* (maximizer of WBS) and *weighted Borda dynamic regret* (analogous to (1)) follow suit. Taking $\mathbf{w}$ to be $\text{Unif}\{[K]\}$, this recovers the Borda score and dynamic regret.

If a Condorcet winner $a_t^C$ exists, then taking $\mathbf{w}$ to be the point-mass weight $\mathbf{w}(a_t^C)$ on $a_t^C$ also allows us to capture the Condorcet regret. Indeed, one observes that the CW $a_t^C$ maximizes the score $b_t(a, \mathbf{w}(a_t^C))$, and the regret in terms of weighted Borda scores of arm $a$ w.r.t. weight $\mathbf{w}(a_t^C)$ becomes

$$b_t(a_t^C, \mathbf{w}(a_t^C)) - b_t(a, \mathbf{w}(a_t^C)) = P_t(a_t^C, a) - \frac{1}{2},$$

which is precisely the Condorcet regret at round $t$.

Recasting the Condorcet regret objective as a Borda-like regret quantity will be key to bypassing the need for SST∩STI, while still capturing an enhanced measure of non-stationarity.

**Changing and Unknown Weights (Key Difficulty).** The challenge with this reformulation of the Condorcet problem is that the reference weight $\mathbf{w}(a_t^C)$ is unknown since the CW $a_t^C$ is. Furthermore, in the more difficult non-stationary problem, the identity of $a_t^C$ may change at unknown times meaning so can the weight $\mathbf{w}(a_t^C)$. This makes it difficult to even estimate the weighted Borda score $b_t(a, \mathbf{w}_t(a_t^C))$ compared to the usual Borda task where the weight $\mathbf{w}$ is fixed over time.

We will show that such difficulties are resolved by tracking all point-mass weights and carefully amortizing the regret analysis to periods of time where estimating a fixed point-mass weight suffices.

## 7 Proof Outlines and Novelties

To simplify discussion, we'll first outline the proof for the regret upper bound for Borda dueling bandits (Theorem 2) and then subsequently discuss the necessary modifications for the Condorcet setting (Theorem 5).

### 7.1 Proof Outline for Theorem 2

We follow a general proof strategy seen in prior works on non-stationary (dueling) bandits (Suk & Kpotufe, 2022; Buening & Saha, 2023; Suk & Agarwal, 2023). The full proof can be found in Appendix H for the generalized weigted Borda problem with a fixed weight (introduced in Section 6).

**Estimators Concentrate around True Gaps.** Let $\hat{\delta}_t^B(a', a)$ be the estimated weighted Borda score induced by (3) for the uniform weight $\mathbf{w} \sim \text{Unif}\{[K]\}$. We first assert using Freedman's inequality (Lemma 15) that our estimated (cumulative) weighted Borda scores $\sum_{s=s_1}^{s_2} \hat{\delta}_t^B(a', a)$ concentrate about the true values $\sum_{s=s_1}^{s_2} \delta_s^B(a', a)$. This means our arm elimination criterion (4) is valid in the sense that arm $a$ being evicted means it has significant regret (2).

Thus, with high probability, each episode $[t_\ell, t_{\ell+1})$ aligns with the SBS phases (Definition 1) in the sense that a restart only occurs if an SBS has occurred within the episode (Lemma 20). Thus, it suffices to bound the regret in each episode $[t_\ell, t_{\ell+1})$ by the minimax regret rate over each overlapping SBS phase:

$$\sum_{i \in [\tilde{L}_{\text{Known}}]: [\tau_i, \tau_{i+1}) \cap [t_\ell, t_{\ell+1}) \neq \emptyset} K^{1/3} \cdot (\tau_{i+1} - \tau_i)^{2/3}.$$

**Decomposing the per-Episode Regret** Let $a_t^\sharp$ denote the *last safe arm* at round $t$, or the last arm to incur significant Borda regret in the unique SBS phase $[\tau_i, \tau_{i+1})$ containing round $t$, per Definition 1. Furthermore, let $a_\ell$ denote the *last global arm* of episode $[t_\ell, t_{\ell+1})$ or the last arm to be evicted from $\mathcal{A}_{\text{global}}$ in said episode. Then, we can decompose the per-episode regret:

$$\sum_{t=t_\ell}^{t_{\ell+1}-1} \delta_t^B(i_t) + \delta_t^B(j_t) = \sum_{t=t_\ell}^{t_{\ell+1}-1} \delta_t^B(a_t^\sharp) + \sum_{t=t_\ell}^{t_{\ell+1}-1} \delta_t^B(a_\ell, i_t) + \delta_t^B(a_\ell, j_t) + \sum_{t=t_\ell}^{t_{\ell+1}-1} \delta_t^B(a_t^\sharp, a_\ell).$$

We next handle each of the three sums on the RHS above. They are respectively:

- **Per-Episode Regret of the Last Safe Arm:** by definition of the safe arm and SBS (Definition 1) this arm has safe regret $K^{1/3}(\tau_{i+1} - \tau_i)^{2/3}$ on each SBS phase $[\tau_i, \tau_{i+1})$.

- **Per-episode Regret of the Played Arms to Last Global Arm:** this part of the analysis is analogous to Appendix B.1 of Suk & Kpotufe (2022), Appendix A.2 of Buening & Saha (2023), or Appendix C.3 of Suk & Agarwal (2023). First, note the play distribution $q_t$ (5) defaults to uniform exploration at a frequency which is safe for $K^{1/3} \cdot (t_{\ell+1} - t_\ell)^{2/3}$ regret. Thus, it suffices to bound regret when $q_t$ explores the arms in the candidate arm set $\mathcal{A}_t$. Now, $a_\ell$, by definition, is any arm $a'$ retained at all rounds in episode $\ell$, and $\delta_t^B(a', i_t) = \sum_{a \in [K]} \delta_t^B(a', a) \cdot \mathbf{1}\{i_t = a\}$. Thus, it suffices to bound the relative regrets $\sum_t \delta_t^B(a', a) \cdot \mathbf{1}\{i_t = a\}$, by carefully considering intervals of time where $a$ is also retained. In particular, we first bound this sum on an interval from $t_\ell$ to $t_\ell(a)$, defined as the round at which $a$ is evicted from $\mathcal{A}_{\text{global}}$, and then intervals within replays of BOSSE which bring $a$ back into $\mathcal{A}_t$. Regret on the first kind of interval is bounded using our elimination criterion (4) and concentration. Bounding the regret on the second kind of interval uses the key fact that sufficiently few replays are scheduled.

- **Per-Episode Regret of the Last Global Arm to the Last Safe Arm:** here, we'll use a *bad segment analysis* similar to Appendix B.2 of Suk & Kpotufe (2022) or Appendix C.4 of Suk & Agarwal (2023). Roughly, a bad segment $[s_1, s_2]$ will be such that $\sum_{s=s_1}^{s_2} \delta_s^B(a_t^\sharp, a_\ell) \gtrsim (s_2 - s_1)^{2/3} \cdot K^{1/3}$. On such a bad segment $[s_1, s_2]$, a *perfect replay* is roughly defined as a replay whose scheduled duration overlaps most of $[s_1, s_2]$. Then, the key fact is that the randomized scheduling of replays (Line 12 of Algorithm 1) ensures that a perfect replay is scheduled for some bad segment (thus evicting $a_\ell$ from $\mathcal{A}_{\text{global}}$ before too many bad segments elapse, giving us a way of bounding $\sum_{s=s_1}^{s_2} \delta_s^B(a_t^\sharp, a_\ell)$ with high probability.

## 7.2 Challenges of Condorcet Regret Analysis for Proving Theorem 5

Leaving the details of the analysis to Appendix I, we instead focus here on highlighting the key challenges in showing our Condorcet dynamic regret upper bound.

**Recasting Condorcet Regret as a weighted Borda Regret.** As discussed in Section 1, we emphasize our main analysis novelty is in reformulating the Condorcet regret as a weighted Borda-like regret. This allows us to take the analysis route of prior works on non-stationary MAB (Suk & Kpotufe, 2022) without suffering from difficulties endemic to preference feedback, as seen in Buening & Saha (2023); Suk & Agarwal (2023). To our knowledge, this is the first work on dueling bandits to make use of such a trick.

**Keeping Track of All Unknown Weights.** As mentioned earlier in Section 6, a crucial difficulty with making use of the weighted Borda framework for Condorcet dueling bandits is that the reference weights $\mathbf{w}$ are unknown. Furthermore, the space of all possible weights $\Delta^K$ is combinatorially large and thus one cannot hope to maintain estimates for all weights in $\Delta^K$ without an intractable union bound. We show in fact that such accurate estimation can be bypassed because the worst-case regret is always attained at a point-mass weight on some arm $a \in [K]$ due to regret being linear in the weight vector.

**Keeping Track of Unknown Changes in Weights.** An added difficulty in the non-stationary setting is that the unknowns weights may change at unknown times. This makes it challenging to bound the regret over intervals of rounds $[s_1, s_2]$ which elapse multiple approximate winner changes $\rho_i$ and hence for which we require score estimation w.r.t. a changing weight sequence $\{\mathbf{w}_t\}_t$. However, even following the above proposal for tracking point-mass weights, we note the space of all *sequences of point-mass weights* is combinatorially large. This prohibits estimating all $\sum_{s=s_1}^{s_2} b_s(a, \mathbf{w}_s)$ for changing point-mass weights $\mathbf{w}_s$.

Instead, we carefully divide up the regret analysis into segments of time $[s_1, s_2] \subseteq [\rho_i, \rho_{i+1})$ lying within an approximate winner phase $[\rho_i, \rho_{i+1})$ where it suffices to bound the regret using a single weight (as it turns out, that of the current approximate winner $a^\sharp$). This involves doing a separate analysis of when an arm is detected as having significant regret for each arm $a \in [K]$ and each base algorithm.

## 8 Conclusion and Future Questions

We've achieved the first optimal and adaptive dynamic Borda regret upper bound. Additionally, we've introduced the weighted Borda framework which revealed new preference models where faster Condorcet regret rates are attainable, adaptively, in terms of new tighter measures of non-stationarity. In the Condorcet setting, it's still unclear if the bounds of Theorem 5 or the $K\sqrt{S_\mathrm{C}T}$ rate of Buening & Saha (2023) are tight for the GIC class and, more challengingly, whether the $K\sqrt{S_\mathrm{C}T}$ rate can be improved outside of the GIC class. Outside of the Condorcet Winner assumption, tracking changing von-Neumann winners in non-stationary dueling bandits appears a challenging, but quite interesting direction.

More broadly, on the theoretical side, future work can also independently study this weighted Borda framework (with generic known or unknown weights) and characterize the instance-dependent regret rates, which remain unclear even in stationary settings.

It'd also be interesting to explore instance-dependent, or logarithmic, regret bounds for the non-stationary dueling bandit problem. We note that even in the non-stationary MAB problem, it's been shown that logarithmic instance-dependent dynamic regret bounds are unattainable in general without knowledge of non-stationarity (Lattimore & Szepesvári, 2020, Theorem 31.2). Thus, it remains open what is an appropriate instance-dependent dynamic regret rate for this problem.

Future work can also implement Algorithm 2 on real datasets and compare with other algorithms for non-stationary dueling bandits. One challenge here is the high computational cost ($O(KT^2)$ computations over $T$ rounds) which makes our procedure somewhat impractical. We note a recent work (Li et al., 2024) explores ways to improve the computational complexity while preserving optimal regret guarantees in non-stationary bandits.

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

# A  Setting up the Weighted Borda Problem

Recall from Section 6 that the weighted Borda score (WBS) $b_t(a, \mathbf{w})$ is defined as

$$b_t(a, \mathbf{w}) \doteq \sum_{a' \in [K]} P_t(a, a') \cdot w^{a'},$$

with respect to weight $\mathbf{w} \doteq (w^1, \dots, w^K) \in \Delta^K$. Our goal is to minimize the analogous notion of dynamic regret with respect to the quantities $b_t(a, \mathbf{w})$ in two situations: (1) the setting of *known weights* where the vector $\mathbf{w}$ is known to the learner and (2) the setting of *unknown and changing weights* where $\mathbf{w}_t$ changes over time $t$ and are unknown to the learner. These two scenarios will respectively capture the Borda and Condorcet regret problems. To avoid confusion, throughout this appendix we'll sometimes refer to the usual Borda score $b_t(a, \mathrm{Unif}\{[K]\})$ as the *uniform Borda* score (resp. regret, winner).

Throughout this appendix, we'll rewrite the definitions, theorems, and proofs of the body in terms of this general framework (with arbitrary weights $\mathbf{w}_t$) in order to unify presentation across the Borda and Condorcet settings. We first summarize the key results.

Note that our investigation into characterizing the minimax regret rates for this problem go beyond what is needed for showing the main results presented in the body of the paper.

## A.1  Summary of Results in Appendix for Weighted Borda Problem

| Environment | Model Assumption | Reference Weights | Regret Lower Bound | Upper Bound |
|---|---|---|---|---|
| Fixed Winner | None | Known $\mathbf{w}_t = \mathbf{w}$ | $K^{1/3}T^{2/3}$ | $K^{1/3}T^{2/3}$ |
| | | Unknown $\mathbf{w}_t = \mathbf{w}$ | $\Omega(T)$ | |
| | GIC (Condition 1) | Unknown $\mathbf{w}_t = \mathbf{w}$ | $K^{2/3}T^{2/3}$ | $K^{2/3}T^{2/3}$ |
| Non-Stationary | None | Known $\{\mathbf{w}_t\}_t$ | $K^{1/3}\tilde{L}_{\mathrm{Known}}^{1/3}T^{2/3}$ | $K^{1/3}\tilde{L}_{\mathrm{Known}}^{1/3}T^{2/3}$ |
| | GIC (Condition 1) | Unknown, aligned $\{\mathbf{w}_t\}_t$ | $K^{2/3}\tilde{L}_{\mathrm{Unknown}}^{1/3}T^{2/3}$ | $K^{2/3}\tilde{L}_{\mathrm{Unknown}}^{1/3}T^{2/3}$ |

Table 2: Summary of Results

(a) In Appendix B.1, we first establish a minimax regret rate of $K^{1/3}T^{2/3}$ for a known reference weight under the stationary setting. Interestingly, the minimax regret is the same as that of the usual Borda regret, thus showing the generic known weight problem is no harder than the Borda problem.

(b) Then, in Appendix B.2, we establish the minimax regret rate of $K^{1/3}\tilde{L}_{\mathrm{Known}}^{1/3}T^{2/3}$ in the non-stationary dueling bandit problem where the underlying preference model and (known) reference weights are allowed to change, and $\tilde{L}_{\mathrm{Known}}$ counts the *significant changes in weighted winner* (Definition 1).

(c) We next consider the unknown weight setting and show in Appendix C.1 that it's impossible for a single algorithm to simultaneously obtain sublinear regret for both the uniform and Condorcet winner reference weights, thus ruling out getting both $T^{2/3}$ Borda regret and $\sqrt{T}$ Condorcet winner regret.

(d) Then, in Appendix C.2, we show via a lower bound that, even under GIC, a higher dependence on $K$ of $K^{2/3}$ is unavoidable. This is in comparison to the $K^{1/3}$ dependence in the known weight setting).

This implies a lower dynamic regret lower bound of $K^{2/3}\tilde{L}_{\mathrm{Unknown}}^{1/3}T^{2/3}$ in terms of $\tilde{L}_{\mathrm{Unknown}}$ *significant winner switches w.r.t. unknown weights* (Definition 2), which can be found in Appendix C.3.

(e) Finally, we establish matching upper bounds on dynamic regret for both the known and unknown weight settings in, respectively, Appendices H and I. As a reminder, our algorithm is *adaptive* in that it does not require any knowledge of the non-stationarity. As a result, we show Theorem 2 and Theorem 5.

### A.2 Generalized Notions Related to WBS

**Weighted Borda Scores and Regret.** For ease of presentation, in the remainder of the appendix, we reparametrize the WBS $b_t(a, \mathbf{w})$ to be in terms of the *gaps in preferences* $\delta_t(a, a') \doteq P_t(a, a') - \frac{1}{2}$. Or,

$$b_t(a, \mathbf{w}) \doteq \sum_{a' \in [K]} \delta_t(a, a') \cdot w^{a'}.$$

Note the resulting regret notions and rates will not change under this new formulation. Now, define the *WBS winner* as $a_t^*(\mathbf{w}) \doteq \mathrm{argmax}_{a \in [K]} b_t(a, \mathbf{w})$ and the *WBS dynamic regret* as

$$\mathrm{Regret}(\{\mathbf{w}_t\}_{t=1}^T) \doteq \sum_{t=1}^T \frac{1}{2} \left( 2 \cdot b_t(a_t^*(\mathbf{w}_t), \mathbf{w}_t) - b_t(i_t, \mathbf{w}_t) - b_t(j_t, \mathbf{w}_t) \right).$$

In the case of a known weight $\mathbf{w}_t \equiv \mathbf{w}$, we'll simplify the notation as $\mathrm{Regret}(\mathbf{w})$.

Let $\delta_t^{\mathbf{w}}(a', a) \doteq b_t(a', \mathbf{w}) - b_t(a, \mathbf{w})$ be the gap in WBS between arms $a$ and $a'$ at time $t$. Let $\delta_t^{\mathbf{w}}(a) \doteq \max_{a' \in [K]} \delta_t^{\mathbf{w}}(a', a)$ be the absolute gap in WBS of arm $a$.

**Non-Stationarity Measures.** We now introduce generalizations of significant Borda winner switches (Definition 1) for arbitrary weights.

**Definition 1** (**S**ignificant Winner Switches w.r.t. **K**nown **W**eightings (SKW)). *Fix a weight* $\mathbf{w}$*. Define an arm* $a$ *as having* **significant weighted Borda regret** *over* $[s_1, s_2]$ *if*

$$\sum_{s=s_1}^{s_2} \delta_s^{\mathbf{w}}(a) \geq K^{1/3} \cdot (s_2 - s_1)^{2/3}. \tag{8}$$

*Define* **significant winner switches w.r.t. the known weight w** *(abbreviated as SKW) as follows: the 0-th sig. shift is defined as* $\tau_0 = 1$ *and the* $(i+1)$-*th sig. shift* $\tau_{i+1}$ *is recursively defined as the smallest* $t > \tau_i$ *such that for each arm* $a \in [K]$*,* $\exists [s_1, s_2] \subseteq [\tau_i, t]$ *such that arm* $a$ *has significant weighted Borda regret over* $[s_1, s_2]$*. We refer to the interval of rounds* $[\tau_i, \tau_{i+1})$ *as an* SKW *phase. Let* $\tilde{L}_{\mathrm{Known}}$ *be the number of SKW phases elapsed in* $T$ *rounds*[5].

**Remark 3.** *Definition 1 and subsequent results (Theorem 7 and Theorem 12) for a known weight* $\mathbf{w}$ *can be trivially generalized to the setting where there's a fixed and known sequence of weights* $\{\mathbf{w}_t\}_{t=1}^T$*. For simplicity of presentation, we focus on the fixed weight* $\mathbf{w}_t \equiv \mathbf{w}$*, which preserves the essence of the theory.*

We introduce a similar generalization of Definition 4 for the weighted Borda problem with unknown weights. In particular, we define a notion of significant shifts which tracks when an SKW occurs for any weight $\mathbf{w}$.

**Definition 2** (**S**ignificant Winner Switches w.r.t. **U**nknown **W**eightings (SUW)). *Define an arm* $a$ *as having* **significant worst-case weighted Borda regret** *over* $[s_1, s_2]$ *if*

$$\max_{\mathbf{w} \in \Delta^K} \sum_{s=s_1}^{s_2} \delta_s^{\mathbf{w}}(a) \geq K^{2/3} \cdot (s_2 - s_1)^{2/3}. \tag{9}$$

*We then define weighted significant winner switches* $\rho_0, \rho_1, \dots$ *in an analogous manner to Definition 1. Let* $\tilde{L}_{\mathrm{Unknown}}$ *be the number of weighted SUW phases.*

**Remark 1.** *Under GIC,* $\tilde{L}_{\mathrm{Unknown}}$ *is less than the number of changes in the winner* $a_t^*$ *(Condition 1).*

Although SUW are a stronger notion of shift than SKW (i.e., $\tilde{L}_{\mathrm{Unknown}} \geq \tilde{L}_{\mathrm{Known}}$), the SUW can be learned in a more general setting where one aims to minimize dynamic regret w.r.t. any sequence of changing weights $\{\mathbf{w}_t\}_{t=1}^T$ which are *aligned* with the SUW, defined as follows.

**Definition 3** (Aligned Sequence of Weights). *We say a sequence* $\{\mathbf{w}_t\}_{t=1}^T$ *of weights is aligned with SUW if, for each SUW phase* $[\rho_i, \rho_{i+1})$*, the weight* $\mathbf{w}_t$ *does not change for rounds* $t$ *lying in* $[\rho_i, \rho_{i+1})$*.*

In particular, as mentioned in Section 6, setting $\mathbf{w}_t$ as the Condorcet winner weight $\mathbf{w}(a_t^C)$ recovers the Condorcet regret problem. Quite importantly, in this setting, the weights $\mathbf{w}_t$ are unknown and so their changepoints are also unknown since the SUW are unknown.

---

[5]while $\tau_i, \tilde{L}_{\mathrm{Known}}$ depend on the fixed weight $\mathbf{w}$, we'll drop the dependence as needed for sake of presentation

# B Regret Lower Bounds for WBS with Known Weights

## B.1 Stationary Regret Lower Bound

We first show that, in the stationary setting, the problem of minimizing weighted Borda regret w.r.t. any fixed and known weight $\mathbf{w}$ is as hard as the uniform Borda problem (i.e., $K^{1/3} \cdot T^{2/3}$ minimax regret).

**Theorem 6** (Lower Bound on Regret in Stochastic Setting). *For any algorithm and any reference weight $\mathbf{w}_t \equiv \mathbf{w}$, there exists a preference matrix $\mathbf{P}$, with $K \geq 3$, such that the expected regret w.r.t. $\mathbf{w}_t$ is:*

$$\mathbb{E}[\text{Regret}(\mathbf{w})] \geq \Omega(K^{1/3} \cdot T^{2/3}).$$

*Proof.* We construct an environment similar to the lower bound construction for the uniform Borda score (Saha et al., 2021, Lemma 14). As in said result, it will in fact suffice to construct an environment forcing a regret lower bound (w.r.t. weight $\mathbf{w}$) of order $\Omega(\min\{T \cdot \varepsilon, K/\varepsilon^2\})$ for $\varepsilon \in (0, 0.05)$. Taking $\varepsilon := (K/T)^{1/3}$ will then give the desired result, since $(K/T)^{1/3} < 0.05$ for large enough value of $T/K$. For $T/K$ smaller than a constant, we may take $\varepsilon$ small enough so that $\frac{K}{\varepsilon^2} \geq \left(\frac{T}{K}\right)^{2/3} \cdot K = T^{2/3}K^{1/3}$ to conclude.

Suppose $K$ is even; if $K$ is odd, we may reduce to a dueling bandit problem with $K-1$ arms by setting the gaps $\delta(a, a')$ to zero for some fixed arm $a \in [K]$ and any other arm $a' \in [K]$. First observe that for any reference weight $\mathbf{w}$, there must exist a set of arms $\mathcal{I}$ of size at most $K/2$ such that its mass under $\mathbf{w}$ is $w(\mathcal{I}) \geq 1/2$. This follows since either the set of arms $\{1, \ldots, K/2\}$ or $\{K/2 + 1, \ldots, K\}$ has mass at least $1/2$ under the distribution $w(\cdot)$. We then partition the $K$ arms into a *good set* $\mathcal{G} := [K] \backslash \mathcal{I}$ and *bad set* $\mathcal{B} := \mathcal{I}$. Without loss of generality, suppose $\mathcal{G} = \{1, \ldots, K/2\}$. We'll consider $K/2 + 1$ different environments $\mathcal{E}_0, \mathcal{E}_1, \ldots, \mathcal{E}_{K/2}$ where in $\mathcal{E}_a$ for $a \in [K/2]$, arm $a$ will maximize the weighted Borda score $b(\cdot, \mathbf{w})$.

Specifically, we set the preference matrix $\mathbf{P} \equiv \mathbf{P}_t$ as follows:

**Environment $\mathcal{E}_0$:** Let the preference matrix $\mathbf{P}$ have entries

$$P(a, a') = \begin{cases} 0.5 & a, a' \in \mathcal{G} \text{ or } a, a' \in \mathcal{B} \\ 0.9 & a \in \mathcal{G}, a' \in \mathcal{B} \end{cases}.$$

In this environment, the weighted Borda score is

$$b(a, \mathbf{w}) = \begin{cases} 0.4 \cdot w(\mathcal{B}) & a \in \mathcal{G} \\ -0.4 \cdot w(\mathcal{G}) & a \in \mathcal{B} \end{cases},$$

where $w(\cdot)$ denotes the distribution on $\Delta^{[K]}$ induced by weight $\mathbf{w}$. We remind the reader here that the WBS $b(a, \mathbf{w})$ are defined in terms of the preference gaps $\delta_t(a', a)$ and hence can take negative values.

The alternative environments $\mathcal{E}_1, \ldots, \mathcal{E}_{K/2}$ will be small perturbations of $\mathcal{E}_0$ where an arm $a \in \mathcal{G}$ will have the highest weighted Borda score by a margin of $\varepsilon$.

**Environment $\mathcal{E}_a$ for $a \in [K/2]$:** Let the preference matrix $\mathbf{P}$ be identical to that of environment $\mathcal{E}_0$ except in the entries $P(a, a') = 0.9 + \varepsilon$ for $a' \in \mathcal{B}$. Thus, in this environment the weighted Borda scores are identical to that of $\mathcal{E}_0$ except for $b(a, \mathbf{w}) = (0.4 + \varepsilon) \cdot w(\mathcal{B})$, meaning arm $a$ is the winner w.r.t. weight $\mathbf{w}$.

Let $\mathbb{E}_{\mathcal{E}_a}[\cdot], \mathbb{P}_{\mathcal{E}_a}(\cdot)$ denote the expectation and probability measure of the induced distributions on observations and decisions under environment $\mathcal{E}_a$. Now, the expected regret on environment $\mathcal{E}_a$ is lower bounded by

$$\mathbb{E}_{\mathcal{E}_a}[\text{Regret}(\mathbf{w})] \geq \sum_{t=1}^{T} \varepsilon \cdot \mathbb{E}_{\mathcal{E}_a}[\mathbf{1}\{(i_t, j_t) \neq (a, a)\}] = \varepsilon \cdot \left( T - \mathbb{E}_{\mathcal{E}_a}\left[ \sum_{t=1}^{T} \mathbf{1}\{(i_t, j_t) = (a, a)\} \right] \right).$$

Letting $\mathcal{U}$ be a uniform prior over $\{1, \ldots, K/2\}$, we have the expected regret over a random environment drawn from $\mathcal{U}$ is lower bounded by:

$$\mathbb{E}_{a \sim \mathcal{U}}[\mathbb{E}_{\mathcal{E}_a}[\text{Regret}(\mathbf{w})]] \geq \varepsilon \cdot \left( T - \mathbb{E}_{a \in \mathcal{U}}\left[ \mathbb{E}_{\mathcal{E}_a}\left[ \sum_{t=1}^{T} \mathbf{1}\{(i_t, j_t) = (a, a)\} \right] \right] \right). \tag{10}$$

Let $N_T(a) := \sum_{t=1}^T \mathbf{1}\{(i_t, j_t) = (a, a)\}$ be the (random) *arm-pull count* of arm $a$. Then, since $N_T(a) \leq T$ for all values of $a$, by Pinsker's inequality (see Saha & Gaillard, 2022, proof of Lemma C.1), we have

$$\mathbb{E}_{\mathcal{E}_a}[N_T(a)] - \mathbb{E}_{\mathcal{E}_0}[N_T(a)] \leq T\sqrt{\frac{\mathrm{KL}(\mathcal{P}_0, \mathcal{P}_a)}{2}},$$

where $\mathcal{P}_{a'}$ is the induced distribution on the history of observations and decisions under environment $\mathcal{E}_{a'}$. Taking a further expectation over $a \in \mathcal{U}$ and using Jensen's inequality, we obtain:

$$\mathbb{E}_{a \sim \mathcal{U}}[\mathbb{E}_{\mathcal{E}_a}[N_T(a)]] \leq \frac{1}{K/2} \sum_{a \in \{1,\dots,K/2\}} \mathbb{E}_{\mathcal{E}_0}[N_T(a)] + T\sqrt{\frac{2}{K} \sum_{a \in \{1,\dots,K/2\}} \mathrm{KL}(\mathcal{E}_0, \mathcal{E}_a)}. \tag{11}$$

We now aim to bound this last KL term. Letting $\mathcal{H}_t$ be the history of randomness, observations, and decisions till round $t$: $\mathcal{H}_t := \{a\} \cup \{(i_s, j_s, O_s(i_s, j_s))\}_{s \leq t}$ for $t \geq 1$ and $\mathcal{H}_0 := \{a\}$. Let $\mathcal{P}_a^t$ denote the marginal distribution over the round $t$ data $(i_t, j_t, O_t(i_t, j_t))$ under the realization of environment $\mathcal{E}_a$. By the chain rule for KL, we have

$$\mathrm{KL}(\mathcal{P}_0, \mathcal{P}_a) = \sum_{t=1}^T \mathrm{KL}(\mathcal{P}_0^t \mid \mathcal{H}_{t-1}, \mathcal{P}_a^t \mid \mathcal{H}_{t-1}) = \sum_{t=1}^T \sum_{i=K/2+1}^K \mathbb{P}_{\mathcal{E}_0}(\{i_t, j_t\} = \{a, i\}) \cdot \mathrm{KL}(\mathrm{Ber}(0.9), \mathrm{Ber}(0.9 + \varepsilon)). \tag{12}$$

Next, observe the following bound on the KL between $\mathrm{Ber}(0.9)$ and $\mathrm{Ber}(0.9 + \varepsilon)$:

$$\mathrm{KL}(\mathrm{Ber}(0.9), \mathrm{Ber}(0.9 + \varepsilon)) \leq 0.9 \cdot \log\left(\frac{0.9}{0.9 + \varepsilon}\right) + 0.1 \cdot \log\left(\frac{0.1}{0.1 - \varepsilon}\right).$$

By elementary calculations, the above is less than $10 \cdot \varepsilon^2$ for any $\varepsilon \in (0, 0.05)$.

Now, suppose the algorithm incurs regret at most $\epsilon \cdot T$ on all environments $\mathcal{E}_a$, lest we be done. Then, the total number of times a bad arm in $\mathcal{B}$ is played must be at most $\epsilon \cdot T/0.4$ or for all $a \in \{0, 1, \dots, K/2 + 1\}$:

$$\mathbb{E}_{\mathcal{E}_a}\left[\sum_{t=1}^T \mathbf{1}\{\{i_t, j_t\} \cap \mathcal{B} \neq \emptyset\}\right] \leq \frac{\varepsilon \cdot T}{0.4}.$$

Plugging the above two steps into (12) yields

$$\frac{2}{K} \sum_{a \in \{1,\dots,K/2\}} \mathrm{KL}(\mathcal{P}_0, \mathcal{P}_a) \leq \frac{20\varepsilon^2}{K} \sum_{a \in \{1,\dots,K/2\}} \sum_{i=K/2+1}^K \mathbb{E}_{\mathcal{E}_0}\left[\sum_{t=1}^T \mathbf{1}\{\{i_t, j_t\} = \{a, i\}\}\right] \leq \frac{50 \cdot \varepsilon^3 \cdot T}{K}.$$

Then, plugging the above into (11) yields

$$\mathbb{E}_{a \sim \mathcal{U}}[\mathbb{E}_{\mathcal{E}_a}[N_T(a)]] \leq \frac{1}{K/2} \sum_{a \in \{1,\dots,K/2\}} \mathbb{E}_{\mathcal{E}_0}[N_T(a)] + T\sqrt{\frac{50\varepsilon^3}{K} \cdot T}.$$

Thus, plugging the above steps into (10) gives

$$\mathbb{E}_{a \in \mathcal{U}, \mathcal{E}_a}[\mathrm{Regret}(\mathbf{w})] \geq \varepsilon \cdot \left(T - \left(\frac{T}{K/2} + T\sqrt{\frac{50\varepsilon^3}{K} \cdot T}\right)\right).$$

Now, suppose $\frac{50\varepsilon^3 \cdot T}{K} \leq 1/50$. Then, the above RHS is lower bounded by $\varepsilon \cdot T/2500$ for $K \geq 3$.

It remains to handle the case of $\frac{50\varepsilon^3 \cdot T}{K} > 1/50 \implies T \geq \frac{K}{2500\varepsilon^3}$. Suppose, for contradiction, that for all such $T$ we have regret at most $\frac{K}{2500^2\varepsilon^2}$. Then, the regret over just the first $T_0 := \frac{K}{2500\varepsilon^3}$ rounds must also be at most $\frac{K}{2500^2\varepsilon^2} = \varepsilon \cdot T_0/2500$. However, this contradicts the previous case's lower bound for $T = T_0$. $\qquad\square$

## B.2 Dynamic Regret Lower Bound

Now, for the known weight setting, the $K^{1/3}T^{2/3}$ lower bound of Theorem 6 can naturally be extended to a dynamic regret lower bound of order $\tilde{L}_{\text{Known}}^{1/3}T^{2/3}K^{1/3}$ over $\tilde{L}_{\text{Known}}$ SKW phases. The proof techniques are routine, similar to arguments already used for showing dynamic regret lower bounds for Condorcet regret (Saha & Gupta, 2022, Section 5), and will not be done in detail here to avoid redundancy.

**Theorem 7** (Fixed Weight Dynamic Regret Lower Bound in Terms of SKW). *For $L \in [T]$, let $\mathcal{P}(L)$ be the class of environments with SKW $\tilde{L}_{\text{Known}}(\mathbf{w}) \leq L$. For any algorithm, we have that the minimax dynamic regret w.r.t. known weight $\mathbf{w}$ over class $\mathcal{P}(L)$ is lower bounded by*

$$\sup_{\mathcal{E} \in \mathcal{P}(L)} \mathbb{E}_{\mathcal{E}}[\text{Regret}(\mathbf{w})] \geq \Omega(L^{1/3}T^{2/3}K^{1/3}).$$

*Proof.* (Sketch) We can take the lower bound construction of Theorem 6 over a horizon of length $T/L$ to force a regret of $K^{1/3}(T/L)^{2/3}$. Repeating this construction with a new randomly chosen winner arm every $T/L$ rounds forces a total regret of $K^{1/3}L^{1/3}T^{2/3}$ while satisfying $\tilde{L}_{\text{Known}}(\mathbf{w}) \leq L$. □

**Theorem 8** (Fixed Weight Dynamic Regret Lower Bound in Terms of Total Variation). *For $V \in [0, T] \cap \mathbb{R}$, let $\mathcal{P}(V)$ be the class of environments with total variation $V_T$ at most $V$. For any algorithm, we have the minimax dynamic regret w.r.t. known weight $\mathbf{w}$ is lower bounded by*

$$\sup_{\mathcal{E} \in \mathcal{P}(V)} \mathbb{E}_{\mathcal{E}}[\text{Regret}(\mathbf{w})] \geq \Omega(V^{1/4}T^{3/4}K^{1/4} + K^{1/3}T^{2/3}).$$

*Proof.* (Sketch) The argument is analogous to the proof of Theorem 5.2 in Saha & Gupta (2022). First, assume $T^{1/4} \cdot V^{3/4} \cdot K^{-1/4} \geq 1$. Then, using the previous lower bound construction forcing regret of order $L^{1/3} \cdot T^{2/3} \cdot K^{1/3}$, we can use the specialization $L \propto T^{1/4} \cdot V^{3/4} \cdot K^{-1/4}$ which ensures the total variation $V_T$ at most

$$L \cdot (K \cdot L/T)^{1/3} = L^{4/3} \cdot K^{1/3} \cdot T^{-1/3} \leq V$$

and forces a regret lower bound of order

$$L^{1/3} \cdot T^{2/3} \cdot K^{1/3} \propto T^{3/4} \cdot K^{1/4} \cdot V^{1/4}.$$

If $T^{1/4} \cdot V^{3/4} \cdot K^{-1/4} < 1$, then $V^{1/4} \cdot T^{3/4} \cdot K^{1/4} < K^{1/3} \cdot T^{2/3}$ which means it suffices to establish a regret lower bound of order $K^{1/3} \cdot T^{2/3}$ which is already evident from Theorem 6. □

**Remark 2.** *Note the optimal dependence on $V_T$, $T$, and $K$ in Theorem 8 differ from the minimax Condorcet dynamic regret rate of $V_T^{1/3}T^{2/3}K^{1/3}$. This arises from the different stationary minimax regret rates ($T^{2/3}$ versus $T^{1/2}$) of Condorcet versus Borda regret.*

# C Regret Lower Bounds for WBS with Unknown Weights

## C.1 Hardness of Learning all Weighted Winners

We next turn our attention to minimizing stationary regret w.r.t an unknown, but fixed, weight $\mathbf{w}$, which the following theorem asserts is hard.

**Theorem 9** (Lower Bound on Regret with Unknown Weight). *There exists a stochastic problem $\mathbf{P} \in [0,1]^{K \times K}$ such that for any algorithm, there is a reference weight $\mathbf{w} \in \Delta^K$ such that:*

$$\mathbb{E}[\text{Regret}(\mathbf{w})] \geq T/45.$$

*Proof.* Consider the following stationary preference matrix:

$$\mathbf{P} := \begin{pmatrix} 1/2 & 3/5 & 3/5 \\ 2/5 & 1/2 & 1 \\ 2/5 & 0 & 1/2 \end{pmatrix}$$

In this environment arm 1 is the Condorcet winner while arm 2 is the Borda winner w.r.t. the uniform reference weight. For the Condorcet weight $\mathbf{w}(1)$, both arms 2 and 3 have a gap of $1/10$. On the other hand, the Condorcet winner has a gap in Borda scores of $1/15$ to arm 2. However, any algorithm must play one of arms 1, 2, or 3 at least $T/3$ times in expectation. No matter which arm is played $T/3$ times, there is a weight for which its Weighted Borda regret is at least $T/45$. $\qquad\square$

**Remark 3.** *In fact, the lower bound of Theorem 9 holds even for an algorithm knowing the preference matrix $\mathbf{P}$. The task of minimizing weighted Borda regret for an unknown reference weight $\mathbf{w}$ can also be cast as a* **partial monitoring** *problem with fixed feedback $\mathbf{P}$ but unknown matrix of losses decided by the adversary's weight. It's straightforward to verify that this game is not* **globally observable***, meaning the minimax expected regret is $\Omega(T)$ by the classification of finite partial monitoring games (Bartók et al., 2014).*

### C.2 Stationary Regret Lower Bound under GIC

Under GIC (Condition 1), we next focus on deriving regret lower bounds in stochastic environments where there is a fixed winner arm $a^* \equiv a_t^*$ across all rounds. Even under GIC, the best rate we can achieve for all weights is $T^{2/3}$. In particular, so long as an algorithm attains optimal uniform Borda regret it cannot achieve $\sqrt{T}$ Condorcet regret.

**Theorem 10** (Lower Bound on Adaptive Regret with GIC). *Fix any algorithm satisfying*

$$\mathbb{E}[\mathrm{Regret}(\mathrm{Unif}\{[K]\})] \leq C \cdot T^{2/3},$$

*for all $K = 3$ armed dueling bandit instances. Then, there exists a stochastic problem $\mathbf{P} \in [0,1]^{3\times3}$ satisfying Condition 1 such that for the specialization $\mathbf{w} = \mathbf{w}_{a^*}$ and sufficiently large $T$, we have:*

$$\mathbb{E}[\mathrm{Regret}(\mathbf{w})] \geq \Omega(T^{2/3}/C^2).$$

*Proof.* Consider the preference matrix for $\varepsilon := 4C \cdot T^{-1/3}$:

$$\mathbf{P}^1 = \begin{pmatrix} 1/2 & 1/2 & 0.9+\varepsilon \\ 1/2 & 1/2 & 0.9 \\ 0.1-\varepsilon & 0.1 & 1/2 \end{pmatrix}, \mathbf{P}^2 = \begin{pmatrix} 1/2 & 1/2 & 0.9 \\ 1/2 & 1/2 & 0.9+\varepsilon \\ 0.1 & 0.1-\varepsilon & 1/2 \end{pmatrix}.$$

Set $T$ large enough so that $\varepsilon < 0.05$.

We first sketch out the intuition behind the proof. In $\mathbf{P}^1$, arm 1 is both the Condorcet winner and the Borda winner, while in $\mathbf{P}^2$, arm 2 is. The argument will go as follows: first, the algorithm must identify the winner arm and whether we're in $\mathbf{P}^1$ or $\mathbf{P}^2$ since otherwise it pays more than $C \cdot T^{2/3}$ uniform Borda regret. The only way to identify the winner arm, however, is to play arm 3 at least $\Omega(T^{2/3})$ times. This forces a $\Omega(T^{2/3})$ Condorcet winner regret.

Specifically, for $a \in \{1,2\}$ let $a^- := \{1,2\}\backslash\{a\}$ denote the other potential winner arm. Then, we have

$$\mathbb{E}_{\mathbf{P}^a}\left[\mathrm{Regret}(\mathrm{Unif}\{[3]\})\right] \leq C \cdot T^{2/3} \implies \mathbb{E}_{\mathbf{P}^a}\left[\sum_{t=1}^{T}\mathbf{1}\{\{a^-,3\}\cap\{i_t,j_t\}\neq\emptyset\}\right] \leq \frac{3T}{4}.$$

Then, using a Pinsker inequality argument similar to the proof of Theorem 6,

$$\mathbb{E}_{\mathbf{P}^a}\left[\mathrm{Regret}(\mathbf{w}_a)\right] \geq \varepsilon \cdot \mathbb{E}_{\mathbf{P}^a}\left[\sum_{t=1}^{T}\mathbf{1}\{\{3,a^-\}\cap\{i_t,j_t\}\neq\emptyset\}\right]$$

$$\geq \varepsilon \cdot \left(T - \mathbb{E}_{\mathbf{P}^a}\left[\sum_{t=1}^{T}\mathbf{1}\{i_t=j_t=a\}\right]\right)$$

$$\geq \varepsilon \cdot \left(T - \left(\mathbb{E}_{\mathbf{P}^{a^-}}\left[\sum_{t=1}^{T}\mathbf{1}\{i_t=j_t=a\}\right] + T\sqrt{\frac{\mathrm{KL}(\mathcal{P}_{a^-},\mathcal{P}_a)}{2}}\right)\right),$$

where once again $\mathcal{P}_a, \mathcal{P}_{a^-}$ denote the corresponding induced distributions on all random variables. We next bound the above KL divergence by chain rule and our earlier bound on $\mathrm{KL}(\mathrm{Ber}(0.9), \mathrm{Ber}(0.9 + \varepsilon))$:

$$\mathrm{KL}(\mathcal{P}_{a^-}, \mathcal{P}_a) \leq 10\varepsilon^2 \cdot \mathbb{E}_{\mathcal{P}_{a^-}} \left[ \sum_{t=1}^T \mathbf{1}\{\{i_t, j_t\} = \{a, 3\}\} + \mathbf{1}\{\{i_t, j_t\} = \{a^-, 3\}\} \right]$$

Next, note that arm 3 has a uniform Borda gap of at least $0.4 \cdot 2/3$ in either $\mathbf{P}^1$ or $\mathbf{P}^2$. This means we must have

$$\mathbb{E}_{\mathcal{P}_{a^-}} \left[ \sum_{t=1}^T \mathbf{1}\{\{i_t, j_t\} = \{a, 3\}\} + \mathbf{1}\{\{i_t, j_t\} = \{a^-, 3\}\} \right] \leq \frac{C \cdot T^{2/3}}{0.4 \cdot 2/3} = \frac{\varepsilon \cdot T}{0.4 \cdot 8/3}.$$

Thus, our KL bound from earlier becomes $\frac{75}{8} \cdot \varepsilon^3 \cdot T$.

Plugging this into our earlier inequality, we have a regret lower bound of

$$\varepsilon \cdot \left( T - \left( \frac{3T}{4} + T \sqrt{\frac{75}{8} \cdot \varepsilon^3 \cdot T} \right) \right).$$

By a similar argument to the proof of Theorem 6, the above is order $\Omega(\min\{\varepsilon \cdot T, 1/\varepsilon^2\})$ by analyzing the two cases $\varepsilon^3 \cdot T \cdot (75/8) > 8/75$ and $\varepsilon^3 \cdot T \cdot (75/8) \leq 8/75$.

Now, plugging in our value of $\varepsilon$, this last rate is of order $T^{2/3} \cdot \min\{C, C^{-2}\}$. $\qquad\square$

As a consequence, Theorem 10 prohibits using Condorcet regret minimizing algorithms in this setting, and hence new algorithmic techniques are required for efficiently learning winners under GIC.

We next investigate the dependence on $K$ in the minimax regret rate. We show that, to achieve optimal regret for all (unknown) weights adaptively, we need to suffer a higher dependence on $K$ of $K^{2/3}$, compared to the known and fixed weight setting with dependence $K^{1/3}$ (see Theorem 6).

**Theorem 11** (Lower Bound on Adaptive Regret under GIC). *Fix any algorithm. Then, for $K > 3$, there exists a dueling bandit problem $\mathbf{P} \in [0,1]^{K \times K}$ satisfying Condition 1 such that there is a fixed weight $\mathbf{w} \in \Delta^K$ with:*

$$\mathbb{E}[\mathrm{Regret}(\mathbf{w})] \geq \Omega(\min\{K^{2/3}T^{2/3}, T\}).$$

*Proof.* We first sketch the argument. We'll consider a variant of the environments $\mathcal{E}_0, \mathcal{E}_1, \dots, \mathcal{E}_{K/2}$ introduced in the proof of Theorem 6. Recall there is a good set $\mathcal{G} := \{1, \dots, K/2\}$ and a bad set $\mathcal{B} := \{K/2 + 1, \dots, K\}$ of arms. In every environment, the good arms (resp. bad arms) have a gap of zero when compared to each other. In $\mathcal{E}_0$, each good arm has a gap of 0.4 to each bad arm. Now, we introduce a further refinement of this construction and define the environment $\mathcal{E}_{a,a'}$ for $a \in \mathcal{G}$ and $a' \in \mathcal{B}$ as identical to the environment $\mathcal{E}_0$ except $P(a, a') = 0.9 + \varepsilon$. Now, the sample complexity of finding the winner arm $a$ in environment $\mathcal{E}_{a,a'}$ will be $\frac{K^2}{\varepsilon^2}$ whence we'll pay a regret of $\min\{\varepsilon \cdot T, K^2 \cdot \varepsilon^{-2}\}$. Setting $\varepsilon \propto K^{2/3}T^{-1/3}$ then forces $K^{2/3} \cdot T^{2/3}$ regret.

Let $\mathcal{U}$ be a uniform prior over pairs of arms $(a, a')$ for $a \in \mathcal{G}$ and $a' \in \mathcal{B}$. Then, we have

$$\mathbb{E}_{(a,a') \sim \mathcal{U}} \mathbb{E}_{\mathcal{E}_{a,a'}}[\mathrm{Regret}(\mathbf{w}(a'))] \geq \varepsilon \cdot \left( T - \mathbb{E}_{(a,a') \sim \mathcal{U}} \left[ \mathbb{E}_{\mathcal{E}_{a,a'}} \left[ \sum_{t=1}^T \mathbf{1}\{(i_t, j_t) = (a, a)\} \right] \right] \right).$$

Now, by Pinsker's inequality we have

$$\mathbb{E}_{(a,a') \sim \mathcal{U}} \left[ \mathbb{E}_{\mathcal{E}_{a,a'}}[N_T(a)] \right] \leq \frac{1}{(K/2)^2} \sum_{a \in \mathcal{G}, a' \in \mathcal{B}} \mathbb{E}_{\mathcal{E}_0}[N_T(a)] + T \sqrt{\frac{1}{(K/2^2)} \sum_{a \in \mathcal{G}, a' \in \mathcal{B}} \mathrm{KL}(\mathcal{E}_0, \mathcal{E}_{a,a'})},$$

where recall $N_T(a) := \sum_{t=1}^T \mathbf{1}\{(i_t, j_t) = (a, a)\}$. Next, we bound this KL in an identical way to the calculations of the proof of Theorem 6:

$$\mathrm{KL}(\mathcal{E}_0, \mathcal{E}_{a,a'}) \leq 10 \cdot \varepsilon^2 \cdot \mathbb{E}_{\mathcal{E}_0} \left[ \sum_{t=1}^T \mathbf{1}\{\{i_t, j_t\} = \{a, a'\}\} \right].$$

Now, suppose the algorithm incurs regret at most $\varepsilon \cdot T$ on environment $\mathcal{E}_0$ lest we be done. This means it cannot pull arms $a' \in \mathcal{B}$ more than $T \cdot \varepsilon / 0.4$ times in total, or

$$\mathbb{E}_{\mathcal{E}_0}\left[\sum_{a \in \mathcal{G}, a' \in \mathcal{B}}\sum_{t=1}^{T}\mathbf{1}\{\{i_t, j_t\} = \{a, a'\}\}\right] \leq \frac{T \cdot \varepsilon}{0.4}.$$

Thus,

$$\frac{4}{K^2}\sum_{a \in \mathcal{G}, a' \in \mathcal{B}}\mathrm{KL}(\mathcal{E}_0, \mathcal{E}_{a,a'}) \leq \frac{100 \cdot \varepsilon^3 \cdot T}{K^2}.$$

Plugging this KL bound into our earlier Pinsker bound, we obtain a regret lower bound of

$$\mathbb{E}_{(a,a')\sim\mathcal{U}}\mathbb{E}_{\mathcal{E}_{a,a'}}[\mathrm{Regret}(\mathbf{w}(a'))] \geq \varepsilon \cdot \left(T - \left(\frac{4T}{K^2} + T\sqrt{\frac{100 \cdot \varepsilon^3 \cdot T}{K^2}}\right)\right).$$

By an analogous argument to the proof of Theorem 6, the above is lower bounded by $\Omega(\min\{\varepsilon \cdot T, K^2\varepsilon^{-2}\})$.

$\square$

## C.3 Dynamic Regret Lower Bounds

By analogous arguments as in the previous section, except now using the stationary lower bound construction of Theorem 11 as a base template, we have the dynamic regret is lower bounded by $L^{1/3} \cdot T^{2/3} \cdot K^{2/3}$ for environments SUW count $\tilde{L}_{\mathrm{Unknown}} \leq L$.

For total variation budget $V$, we claim the regret lower bound is of order $V^{1/4} \cdot T^{3/4} \cdot K^{1/2} + K^{2/3} \cdot T^{2/3}$. In particular, we note that if $V = O(K^{2/3}T^{-1/3})$, then $V^{1/4}K^{3/4}K^{1/2} = O(K^{2/3}T^{2/3})$ so that it suffices to use the stationary regret bound of Theorem 11. Otherwise, we have $V = \Omega(K^{2/3}T^{-1/3})$ and $L \doteq V^{3/4}T^{1/4}K^{-1/2} = \Omega(1)$ so that a regret lower bound of order $L^{1/3}T^{2/3}K^{2/3} = \Omega(V^{1/4}T^{3/4}K^{1/2})$ holds. At the same time $L \cdot (K^2/T)^{1/3} \leq V$ so that the constructed environment has total variation at most $V$.

# D Formal Statement of Regret Upper Bounds

## D.1 Known Weights.

We first state formally the upper bound on dynamic regret w.r.t. known weight $\mathbf{w}$ in terms of the SKW (Definition 1). In particular, we show matching upper bounds, up to log terms of the lower bounds of Appendix B.2. As a reminder, taking $\mathbf{w}$ to be the uniform weight, we recover the Borda dynamic regret and so the below results generalize Theorem 2 and Corollary 3.

**Theorem 12.** *Algorithm 2 with the fixed weight specification (see Definition 4) w.r.t. fixed weight $\mathbf{w}$ satisfies:*

$$\mathbb{E}[\mathrm{Regret}(\mathbf{w})] \leq \tilde{O}\left(\sum_{i=0}^{\tilde{L}_{\mathrm{Known}}(\mathbf{w})} K^{1/3} \cdot (\tau_{i+1}(\mathbf{w}) - \tau_i(\mathbf{w}))^{2/3}\right).$$

**Corollary 13.** *By Jensen's inequality, the regret bound of Theorem 12 is further upper bounded by $K^{1/3}T^{2/3}\tilde{L}_{\mathrm{Known}}^{1/3}$. Furthermore, relating SKW to total variation, a regret bound of order $V_T^{1/4} \cdot T^{3/4} \cdot K^{1/4} + K^{1/3} \cdot T^{2/3}$ also holds in terms of total variation quantity $V_T$ (see Appendix H.5 for proof).*

## D.2 Unknown Weights.

We next state the analogous result for the unknown weight setting.

**Theorem 14.** *Algorithm 2 with the unknown weight specification (see Definition 4) satisfies for all sequences of aligned weights $\{\mathbf{w}_t\}_t$ (see Definition 3):*

$$\mathbb{E}[\mathrm{Regret}(\{\mathbf{w}_t\}_{t=1}^T)] \leq \tilde{O}(\min\{K^{2/3}T^{2/3}\tilde{L}_{\mathrm{Unknown}}^{1/3}, K^{1/2}V_T^{1/4}T^{3/4} + K^{2/3}T^{2/3}\}).$$

**Remark 4.** *Interestingly, Theorem 14 holds even without GIC (Condition 1). However, without GIC, Definition 2 may not be a meaningful notion of non-stationarity as (8) may be triggered even in stationary environments.*

As a warmup to proving the above dynamic regret upper bounds, we'll first show a regret upper bound in easier *fixed winner* environments where there is a fixed winner arm either (1) with respect to a known weight or (2) with respect to all unknown weights under Condition 1.

These "simpler" analyses will reappear and serve as a core template for the more complicated dynamic regret analyses of Appendix H (known weight) and Appendix I (unknown weight).

Even before doing this, however, we'll establish some preliminary notation and concentration bounds on estimation error, which will be used in all our regret analyses.

## E  Regret Analysis Preliminaries and Estimation Bounds

Throughout the regret upper bound analyses $c_1, c_2, \ldots$ will denote positive constants not depending on $T$ or any distributional parameters. We first recall a version of Freedman's inequality, which has become standard in adaptive non-stationary bandit analyses (Suk & Agarwal, 2023; Buening & Saha, 2023; Suk & Kpotufe, 2022).

**Lemma 15** (Theorem 1 of Beygelzimer et al. (2011)). *Let $X_1, \ldots, X_n \in \mathbb{R}$ be a martingale difference sequence with respect to some filtration $\{\mathcal{F}_0, \mathcal{F}_1, \ldots\}$. Assume for all $t$ that $X_t \leq R$ a.s. and that $\sum_{i=1}^n \mathbb{E}[X_i^2 | \mathcal{F}_{i-1}] \leq V_n$ a.s. for some constant $V_n$ only depending on $n$. Then for any $\delta \in (0,1)$ and $\lambda \in [0, 1/R]$, with probability at least $1 - \delta$, we have:*

$$\sum_{i=1}^n X_i \leq (e-1)\left(\sqrt{V_n \log(1/\delta)} + R\log(1/\delta)\right).$$

We next apply Lemma 15 to bound the estimation error of our estimates $\hat{b}_t(a, \mathbf{w})$ (see (3) in Subsection 4.1)

**Proposition 16.** *Let $\mathcal{E}_1$ be the event that for all rounds $s_1 < s_2$ and all arms $a \in \mathcal{A}_t$ for all $t \in [s_1, s_2]$, and weight $\mathbf{w} \in \mathcal{W}$:*

$$\left|\sum_{t=s_1}^{s_2} \hat{b}_t(a, \mathbf{w}) - \sum_{t=s_1}^{s_2} \mathbb{E}\left[\hat{b}_t(a, \mathbf{w}) \mid \mathcal{F}_{t-1}\right]\right| \leq 10(e-1)\log(T|\mathcal{W}|)\left(\sqrt{\left(\sum_{s=s_1}^{s_2} K \sum_{a'} \frac{w_{a'}^2}{q_s(a')}\right)} + K \cdot \max_{t \in [s_1, s_2]} \eta_t^{-1}\right).$$
(13)

*where $\mathcal{F} := \{\mathcal{F}_t\}_{t=1}^T$ is the canonical filtration generated by observations and randomness of elapsed rounds. Then, $\mathcal{E}_1$ occurs with probability at least $1 - 1/T^2$.*

*Proof.* First, note for any arm $a \in [K]$ and weight $\mathbf{w}$, the random variable $\hat{b}_t(a, \mathbf{w}) - \mathbb{E}[\hat{b}_t(a, \mathbf{w})|\mathcal{F}_{t-1}]$ is a martingale difference bounded above by $\max_{t \in [s_1, s_2]} K \cdot \eta_t^{-1}$. Then, in light of Lemma 15, it suffices to compute the variance of $\hat{b}_t(a)$. We have for arm $a$ active at round $t$ (i.e., $a \in \mathcal{A}_t$):

$$\mathbb{E}[\hat{b}_t^2(a)|\mathcal{H}_{t-1}] \leq \mathbb{E}\left[\left(\sum_{a' \in [K]} \frac{\mathbf{1}\{i_t = a, j_t = a'\} \cdot (O_t(a,a') - 1/2)}{q_t(a) \cdot q_t(a')} \cdot w_{a'}\right)^2\right]$$

$$= \mathbb{E}\left[\sum_{a' \in [K]} \frac{\mathbf{1}\{(i_t, j_t) = (a, a')\}}{q_t^2(a) \cdot q_t^2(a')} \cdot w_{a'}^2\right]$$

$$= \sum_{a' \in [K]} \frac{w_{a'}^2}{q_t(a) \cdot q_t(a')}$$

$$\leq K \sum_{a'} \frac{w_{a'}^2}{q_t(a')},$$

where the last inequality follows from the fact that $q_t(a) \geq 1/K$ by virtue of $a \in \mathcal{A}_t$. Then, the result follows from Lemma 15 and taking union bounds over arms $a$, weight $\mathbf{w} \in \mathcal{W}$, and rounds $s_1, s_2$. $\qquad\square$

We next establish separate applications of this concentration bound for the setting of known weights and unknown weights. For these settings, we respectively set $\mathcal{W} = \{\mathbf{w}\}$ for known weight $\mathbf{w}$ and $\mathcal{W} = \{\mathbf{w}_a, a \in [K]\}$ for unknown weights.

**Known Reference Weight.** First, in the case of a known and fixed reference weight $\mathcal{W} = \{\mathbf{w}\}$, we can observe:

$$q_s(a) \geq \eta_s \cdot w_a \implies K \sum_{a'} \frac{w_{a'}^2}{q_s(a')} \leq K \sum_{a'} w_{a'} \cdot \eta_s^{-1} = K \cdot \eta_s^{-1}. \tag{14}$$

**Unknown Reference Weight.** For an unknown reference weight, and when $\mathcal{W} = \{\mathbf{w}_a, a \in [K]\}$, we have the algorithm plays, w.p. $\eta_t$ at round $t$, from $\mathrm{Unif}\{[K]\}$ so that:

$$q_t(a) \geq \eta_t/K \implies K \sum_{a'} \frac{w_{a'}^2}{q_s(a')} \leq \frac{K^2}{\eta_t}. \tag{15}$$

Now, combining (14) and (15) with (13) yields the following *parameter specifications* for use in the known vs. unknown weight setting.

**Definition 4** (Parameter Specifications). *We define two parameter specifications for use in Algorithm 1 (for known vs. unknown weights). We define the* **known weight specification** *w.r.t. weight $\mathbf{w} \in \Delta^K$ via $\mathcal{W} \doteq \{\mathbf{w}\}$, $\gamma_t \doteq \min\{K^{1/3} \cdot t^{-1/3}, 1\}$,*

$$F([s_1, s_2]) \doteq K^{1/3} \cdot (s_2 - s_1)^{2/3} \vee \left( \sqrt{K \sum_{s=s_1}^{s_2} \eta_s^{-1}} + K \max_{s \in [s_1, s_2]} \eta_s^{-1} \right). \tag{16}$$

*The* **unknown weight specification** *is defined via $\mathcal{W} \doteq \{\mathbf{w}_a, a \in [K]\}$ (i.e., the point-mass weights), $\gamma_t \doteq \min\{K^{2/3} \cdot t^{-1/3}, 1\}$, and*

$$F([s_1, s_2]) \doteq K^{2/3} \cdot (s_2 - s_1)^{2/3} \vee \left( \sqrt{K^2 \sum_{s=s_1}^{s_2} \eta_s^{-1}} + K \max_{s \in [s_1, s_2]} \eta_s^{-1} \right). \tag{17}$$

We next establish an elementary helper lemma which asserts that the intervals of rounds $[s_1, s_2]$ over which we evict an arm $a$ in (4) must be at least $\Omega(K)$ rounds in length for the fixed weight specification and at least $\Omega(K^2)$ rounds in length for the unknown weight specification. This will serve useful in further simplifying the concentration bound of (13) throughout the regret analysis, as needed.

**Lemma 17.** *On event $\mathcal{E}_1$, letting $F([s_1, s_2])$ be as in (16) with the fixed weight specification, we have that the eviction criterion (4) holding over interval $[s_1, s_2]$ implies*

$$s_2 - s_1 \geq K/8.$$

*Letting $F([s_1, s_2])$ be as in (17) with the unknown weight specification, we have*

$$s_2 - s_1 \geq K^2/8.$$

*Proof.* By concentration (Proposition 16) and since the weighted Borda gaps $\delta_t^{\mathbf{w}}(a', a)$ are bounded above by 1, eviction of an arm $a$ over $[s_1, s_2]$ under the fixed weight specification implies

$$2 \cdot (s_2 - s_1) \geq s_2 - s_1 + 1 \geq \sum_{s=s_1}^{s_2} \delta_s^{\mathbf{w}}(a) \geq K^{1/3} \cdot (s_2 - s_1)^{2/3}.$$

The above implies $s_2 - s_1 \geq K/8$. For the unknown weight specification, we repeat the above argument with $K^{1/3}$ replaced by $K^{2/3}$. $\qquad\square$

## F    Stationary Regret Upper Bound for Known Weight

**Theorem 18** (Regret Upper Bound for Known Weight)**.** *Algorithm 1 with the fixed weight specification (see Definition 4) w.r.t. fixed weight $\mathbf{w}$, satisfies in any environment with a fixed winner $a^* \equiv a_t^*(\mathbf{w})$:*

$$\mathbb{E}[\text{Regret}(\mathbf{w})] \leq \tilde{O}(K^{1/3} \cdot T^{2/3}).$$

*Proof.* We first note the regret bound is vacuous for $T < K$; so, assume $T \geq K$.

WLOG suppose arms are evicted in the order $1, 2, \ldots, K$ at respective times $t_1 \leq t_2 \leq \cdots \leq t_K$ (if arm $a$ is not evicted, let $t_a := T + 1$. Then, we can first decompose the regret depending on whether we play an active arm or explore other arms with probability $\eta_t$:

$$\mathbb{E}\left[\sum_{t=1}^{T} 2b_t(a^*, \mathbf{w}) - b_t(i_t, \mathbf{w}) - b_t(j_t, \mathbf{w})\right] \leq \sum_{t=1}^{T} \eta_t + \mathbb{E}\left[\sum_{t=1}^{T} \sum_{a,a' \in \mathcal{A}_t} (2 \cdot b_t(a^*, \mathbf{w}) - b_t(a, \mathbf{w}) - b_t(a', \mathbf{w})) \cdot q_t(a)\right].$$

Using the fixed specification for $\eta_t = (K/t)^{1/3}$ from Definition 4, we have first term on the RHS above (our exploration cost) is of order $K^{1/3} \cdot T^{2/3}$.

Now, in light of our concentration bound Proposition 16, it suffices to bound the regret on the high–probability good event $\mathcal{E}_1$ since the total regret is negligible outside of this event. For the second expectation, using our fixed-weight concentration bound (14) we get that the regret of playing arm $a$ as a candidate is at most (on the good event $\mathcal{E}_1$):

$$\mathbf{1}\{\mathcal{E}_1\} \sum_{t=1}^{t_a-1} \frac{b_t(a^*, \mathbf{w}) - b_t(a, \mathbf{w})}{|\mathcal{A}_t|} \leq 10 \cdot (e-1) \log(T) \left(\sqrt{K \sum_{t=1}^{t_a-1} \eta_t^{-1}} + K \cdot \max_{t \in [1, t_a-1]} \eta_t^{-1}\right). \tag{18}$$

Note that $|\mathcal{A}_t| \geq K + 1 - a$ since we assumed WLOG that arm $a$ is the $a$-th arm to be evicted. Additionally, plugging in the specialization for $\eta_t$ according to the fixed weight specification of Definition 4, we have that

$$K \sum_{t=1}^{t_a-1} \eta_t^{-1} \leq K^2 + K \sum_{t=1}^{t_a-1} \left(\frac{t}{K}\right)^{1/3} \leq K^2 + c_1 K^{2/3} \cdot (t_a - 1)^{4/3},$$

$$K \cdot \max_{t \in [1, t_a-1]} \eta_t^{-1} = K^{2/3} \cdot (t_a - 1)^{1/3}.$$

Next, by Lemma 17, we must have $t_a - 1 \geq K/8$ so that $K^{2/3} \cdot (t_a - 1)^{1/3}$ is of order $K^{1/3} \cdot t_a^{2/3}$. By the same reasoning, $K^2$ is of order $K^{2/3} \cdot (t_a - 1)^{4/3}$.

Then, plugging the above two displays into (18) and summing the inequality over arms $a$ gives:

$$2 \cdot \mathbb{E}\left[\mathbf{1}\{\mathcal{E}_1\} \sum_{a=1}^{K} \sum_{t=1}^{t_a-1} \frac{b_t(a^*, \mathbf{w}) - b_t(a, \mathbf{w})}{|\mathcal{A}|}\right] \leq \sum_{a=1}^{K} \frac{c_2 \log(T) \cdot K^{1/3} \cdot t_a^{2/3}}{K + 1 - a}.$$

Upper bounding each $t_a$ by $T$, we obtain a regret bound of order $\log(K) \cdot \log(T) \cdot K^{1/3} T^{2/3}$.    □

## G    Regret Upper Bound under Fixed Winner for Unknown Weight

**Theorem 19** (Adaptive Regret Upper Bound for Unknown Weights)**.** *Suppose Condition 1 holds and there is a winner arm $a^* \equiv a_t^*$ fixed across time. Then, Algorithm 1 with the unknown weight specification (see Definition 4) satisfies for all weights $\mathbf{w} \in \Delta^K$:*

$$\mathbb{E}[\text{Regret}(\mathbf{w})] \leq \tilde{O}(K^{2/3} \cdot T^{2/3}).$$

*Proof.* We first show that it suffices to bound regret w.r.t. the point-mass weights $\mathbf{w}_a \in \mathcal{W}$ for $a \in [K]$. This is true since the regret is linear in the reference weight $\mathbf{w}$:

$$\sum_{t=1}^{T} b_t(a^*, \mathbf{w}) - b_t(q_t, \mathbf{w}) = \mathbb{E}_{a' \sim \mathbf{w}} \left[ \sum_{t=1}^{T} \delta_t(a^*, a') - \delta_t(q_t, a') \right] \leq \max_{a \in [K]} \sum_{t=1}^{T} b_t(a^*, \mathbf{w}_a) - b_t(q_t, \mathbf{w}_a).$$

Now, in light of our concentration bound (14) from Proposition 16, it suffices to bound the regret on the high–probability good event $\mathcal{E}_1$ since the total regret is negligible outside of this event.

We then follow the recipe of the proof of Theorem 19 (see Appendix F) except now using the concentration bound of Proposition 16 with (15) and the unknown weight specification $\eta_t \doteq K^{2/3}/t^{1/3}$. First, note that it's clear the exploration cost $\sum_{t=1}^{T} \eta_t$ for the unknown weight specification is order $K^{2/3} \cdot T^{2/3}$.

Now, suppose WLOG that the arms are evicted in the order $1, 2, \ldots, K$ at times $t_1 \leq t_2 \leq \cdots \leq t_K$. Then, Proposition 16 gives us a bound with respect to any point-mass weight in $\mathcal{W}$:

$$\mathbf{1}\{\mathcal{E}_1\} \max_{a' \in [K]} \sum_{t=1}^{t_a-1} \frac{b_t(a^*, \mathbf{w}(a')) - b_t(a, \mathbf{w}(a'))}{|\mathcal{A}_t|} \leq 10 \cdot (e-1) \log(T) \left( \sqrt{K^2 \sum_{t=1}^{t_a-1} \eta_t^{-1}} + K \cdot \max_{t \in [1, t_a-1]} \eta_t^{-1} \right). \quad (19)$$

Crucially, note that the fixed winner arm $a^*$ is never evicted by Condition 1. Now, we have

$$K^2 \sum_{t=1}^{t_a-1} \eta_t^{-1} \leq K^4 + K^2 \sum_{t=1}^{t_a-1} \frac{t^{1/3}}{K^{2/3}} \leq K^4 + c_3 K^{4/3} \cdot (t_a - 1)^{4/3},$$

$$K \cdot \max_{t \in [1, t_a-1]} \eta_t^{-1} = K^{1/3} \cdot (t_a - 1)^{1/3}.$$

Now, by Lemma 17, we have since $s_2 - s_1 \geq K^2/8$, the $K^4$ term in the first display above is of order $K^{4/3} \cdot (t_a - 1)^{4/3}$.

Then, plugging the above two displays into (19) and summing over arms $a \in [K]$, we have:

$$2\mathbb{E}\left[ \mathbf{1}\{\mathcal{E}_1\} \sum_{a=1}^{K} \sum_{t=1}^{t_a} \frac{b_t(a^*, \mathbf{w}) - b_t(a, \mathbf{w})}{|\mathcal{A}|} \right] \leq \sum_{a=1}^{K} \frac{c_4 \log(T) \cdot K^{2/3} \cdot t_a^{2/3}}{K + 1 - a}.$$

As before, upper bounding each $t_a$ by $T$, we obtain a regret bound of order $\log(K) \cdot \log(T) \cdot K^{2/3} \cdot T^{2/3}$. $\square$

## H   Dynamic Regret Analysis for Known Weights

**Proof of Theorem 12**   The broad outline of our regret analysis will be similar to prior works on adaptive non-stationary (dueling) bandits (Suk & Agarwal, 2023; Buening & Saha, 2023; Suk & Kpotufe, 2022). Our first goal is to show that episodes $[t_\ell, t_{\ell+1})$ align with the SKW phases, in the sense that a new episode is triggered only when an SKW has occurred.

Recall from Algorithm 2 that $[t_\ell, t_{\ell+1})$ represents the $\ell$-th episode. As a notation, we suppose there are $T$ total episodes and, by convention, we let $t_\ell := T + 1$ if only $\ell - 1$ episodes occurred by round $T$.

**Episodes Align with SKW Phases**

**Lemma 20.** *On event $\mathcal{E}_1$, for each episode $[t_\ell, t_{\ell+1})$ with $t_{\ell+1} \leq T$ (i.e., an episode which concludes with a restart), there exists an SKW $\tau_i \in [t_\ell, t_{\ell+1})$.*

*Proof.* We first note that $\hat{b}_t(a, \mathbf{w})$ is an unbiased estimator for $b_t(a, \mathbf{w})$ in the sense that $\mathbb{E}[\hat{b}_t(a, \mathbf{w})|\mathcal{F}_{t-1}] = b_t(a, \mathbf{w})$ if $a \in \mathcal{A}_t$. Then, by concentration (Proposition 16) and our eviction criteria (4), we have that arm $a$ being evicted from $\mathcal{A}_t$ over the interval $[s_1, s_2]$ using the eviction threshold (16) implies

$$\sum_{s=s_1}^{s_2} b_s(a_s^*, \mathbf{w}) - b_s(a, \mathbf{w}) \geq c_5 (s_2 - s_1)^{2/3} \cdot K^{1/3}.$$

This means arm $a$ incurs significant weighted Borda regret w.r.t. $\mathbf{w}$ over $[s_1, s_2]$. Since a new episode is triggered only if all arms are evicted from $\mathcal{A}_{\text{global}}$, there must exist an SKW in each episode $[t_\ell, t_{\ell+1})$. $\qquad\square$

## H.1 Decomposing the Regret

Let $a_t^\sharp$ denote the *last safe arm* at round $t$, or the last arm to incur significant weighted Borda regret w.r.t. $\mathbf{w}$ in the unique phase $[\tau_i, \tau_{i+1})$ containing round $t$, per Definition 1. Furthermore, let $a_\ell$ denote the *last global arm* of episode $[t_\ell, t_{\ell+1})$ or the last arm to be evicted from $\mathcal{A}_{\text{global}}$ in said episode. Then, we can decompose the per-episode regret:

$$\mathbb{E}\left[\sum_{t=t_\ell}^{t_{\ell+1}-1} \delta_t^{\mathbf{w}}(i_t) + \delta_t^{\mathbf{w}}(j_t)\right] = \mathbb{E}\left[\sum_{t=t_\ell}^{t_{\ell+1}-1} \delta_t^{\mathbf{w}}(a_t^\sharp)\right] + \mathbb{E}\left[\sum_{t=t_\ell}^{t_{\ell+1}-1} \delta_t^{\mathbf{w}}(a_\ell, i_t) + \delta_t^{\mathbf{w}}(a_\ell, j_t)\right] + \mathbb{E}\left[\sum_{t=t_\ell}^{t_{\ell+1}-1} \delta_t^{\mathbf{w}}(a_t^\sharp, a_\ell)\right]. \tag{20}$$

We next handle each of the expectations on the RHS separately. Our goal will be to show that each of the expectations above is of order

$$\mathbb{E}\left[\mathbf{1}\{\mathcal{E}_1\} \sum_{i \in [\tilde{L}_{\text{Known}}]: [\tau_i, \tau_{i+1}) \cap [t_\ell, t_{\ell+1}) \neq \emptyset} K^{1/3} \cdot (\tau_{i+1} - \tau_i)^{2/3}\right] + \frac{1}{T}. \tag{21}$$

Admitting this goal, summing the regret over episodes while using Lemma 20 to ensure each SKW phase $[\tau_i, \tau_{i+1})$ only intersects at most two episodes will yield the desired total regret bound. This will follow in a nearly identical manner to Section 5.5 of Suk & Kpotufe (2022).

Now, the three expectations on the RHS of (20) are respectively:

- The per-episode dynamic regret of the last safe arm (analyzed in Appendix H.2)

- The per-episode regret of the played arms $\{i_t, j_t\}$ to the last global arm (analyzed in Appendix H.3)

- The per-episode regret of the last global arm to the last safe arm (analyzed in Appendix H.4).

## H.2 Bounding the per-Episode Regret of the Safe Arm

By the definition of SKW (Definition 1), the safe arm $a_t^\sharp$ is fixed for $t \in [\tau_i, \tau_{i+1})$ and has dynamic regret upper bounded by $K^{1/3} \cdot (s_2 - s_1)^{2/3}$ on any subinterval $[s_1, s_2] \subseteq [\tau_i, \tau_{i+1})$. In particular, letting $[s_1, s_2]$ be the intersection $[\tau_i, \tau_{i+1}) \cap [t_\ell, t_{\ell+1})$ (which is necessarily an interval), we have that the dynamic regret of $a_t^\sharp$ on episode $[t_\ell, t_{\ell+1})$ is at most order (21).

## H.3 Bounding per-Episode Regret of Active Arms to Last Global Arm

This will follow a similar argument as our regret analysis in fixed winner environments (Theorem 18). Recall that the global learning rate $\eta_t$ is the learning rate $\gamma_{t-t_{\text{start}}}$ set by the base algorithm which is active at round $t$. Note that $\eta_t$ is a random variable which depends on the scheduling of base algorithms in episode $[t_\ell, t_{\ell+1})$.

We first observe that the play distribution $q_t$ plays an arm according to weight $\mathbf{w}$ with probability $\eta_t$ and plays an arm chosen from $\mathcal{A}_t$ uniformly at random with probability $1 - \eta_t$. Let $a$ be a random draw from the distribution $q_t$ be the play distribution at round $t$, which is measurable w.r.t. $\mathcal{F}_{t-1}$. Then, we may rewrite the regret as

$$\mathbb{E}_{t_\ell}\left[\sum_{t=t_\ell}^{T} \mathbb{E}_{q_t}[\mathbb{E}[\mathbf{1}\{t < t_{\ell+1}\} \cdot b_t(a, \mathbf{w}) \mid t_\ell, q_t] \mid t_\ell]\right].$$

Note that $\mathbf{1}\{t < t_{\ell+1}\}$ and $b_t(a, \mathbf{w})$ are independent conditional on $t_\ell$ and $q_t$. Next, observe that:

$$\mathbb{E}[b_t(a, \mathbf{w})|t_\ell, q_t] = \eta_t \cdot \mathbb{E}_{a \sim \mathbf{w}}[b_t(a, \mathbf{w})|t_\ell, q_t] + (1 - \eta_t) \cdot \mathbb{E}_{a \sim \text{Unif}\{\mathcal{A}_t\}}[b_t(a, \mathbf{w})|t_\ell, q_t].$$

Plugging in the results of the above to our regret formula, we obtain:

$$\mathbb{E}\left[\sum_{t=t_\ell}^{t_{\ell+1}-1} \delta_t^{\mathbf{w}}(a_\ell, i_t) + \delta_t^{\mathbf{w}}(a_\ell, j_t)\right] \le \mathbb{E}\left[\sum_{t=t_\ell}^{t_{\ell+1}-1} \eta_t\right] + \mathbb{E}\left[\sum_{t=t_\ell}^{t_{\ell+1}-1} \mathbb{E}_{a\sim\text{Unif}\{\mathcal{A}_t\}}\left[\delta_t^{\mathbf{w}}(a_\ell, a)\right]\right]. \tag{22}$$

**Bounding the Regret of Extra Exploration.** We bound the first expectation on the above RHS. Recall that $\gamma_{t-t_{\text{start}}}$ denotes the learning rate $\gamma_t$ set by a base algorithm initiated at round $t_{\text{start}}$. Now, we can coarsely bound the sum of the $\eta_t$'s by the sum of all possible $\gamma_{t-t_{\text{start}}}$, weighted by the probabilities of the scheduling of each base algorithm. So, we have

$$\mathbb{E}\left[\sum_{t=t_\ell}^{t_{\ell+1}-1} \eta_t\right] \le \mathbb{E}\left[\sum_{t=t_\ell}^{t_{\ell+1}-1} \gamma_{t-t_\ell}\right] + \mathbb{E}\left[\sum_{\text{BOSSE}(t_{\text{start}},m)} B_{t_{\text{start}},m} \sum_{t=t_{\text{start}}}^{t_{\text{start}}+m} \gamma_{t-t_{\text{start}}}\right].$$

Recall in the above that the Bernoulli $B_{s,m}$ (see Line 6 of Algorithm 2) decides whether $\text{BOSSE}(s,m)$ is scheduled.

The first sum on the RHS above is order $K^{1/3} \cdot (t_{\ell+1} - t_\ell)^{2/3}$. The analogous sum in the second expectation on the RHS above is order $K^{1/3} \cdot m^{2/3}$. Then, it suffices to bound

$$\mathbb{E}\left[\sum_{t_{\text{start}}=t_\ell+1}^{T} \sum_m B_{t_{\text{start}},m} \cdot \mathbf{1}\{t_{\text{start}} < t_{\ell+1}\} \cdot K^{1/3} \cdot m^{2/3}\right].$$

Conditioning on $t_\ell$, and noting that $\mathbf{1}\{B_{t_{\text{start}},m}\}$ and $\mathbf{1}\{t_{\text{start}} < t_{\ell+1}\}$ are independent conditional on $t_\ell$, we have that by tower rule:

$$\mathbb{E}_{t_\ell}\left[\sum_{t_{\text{start}}=t_\ell+1}^{T} \sum_m \mathbb{E}[B_{t_{\text{start}},m}|t_\ell] \cdot \mathbb{E}[\mathbf{1}\{t_{\text{start}} < t_{\ell+1}\}|t_\ell] \cdot K^{1/3} \cdot m^{2/3}\right].$$

Plugging in $\mathbb{E}[B_{t_{\text{start}},m}|t_\ell] = \frac{1}{m^{1/3} \cdot (t_{\text{start}}-t_\ell)^{2/3}}$ (from Line 12 of Algorithm 2) and taking sums over $m$ and $s$, the above becomes order $\log(T) \cdot K^{1/3} \cdot (t_{\ell+1} - t_\ell)^{2/3}$. This is of the right order with respect to (21) by the sub-additivity of the function $x \mapsto x^{2/3}$.

**Bounding the Regret of Active Arms.** Next, we bound the second expectation on the RHS of (22). This follows a similar argument to Appendix F (the proof of Theorem 18). Supposing WLOG that the arms are evicted from $\mathcal{A}_{\text{global}}$ in the order $1, 2, \ldots, K$ at respective times $t_\ell^1 \le \cdots \le t_\ell^K$, then we have the second expectation on the RHS of (22) can be written as

$$\mathbb{E}\left[\sum_{a=1}^{K} \sum_{t=t_\ell}^{t_\ell^a-1} \frac{\delta_t^{\mathbf{w}}(a_\ell, a)}{|\mathcal{A}_t|} + \sum_{a=1}^{K} \sum_{t=t_\ell^a}^{t_{\ell+1}-1} \frac{\delta_t^{\mathbf{w}}(a_\ell, a)}{|\mathcal{A}_t|} \cdot \mathbf{1}\{a \in \mathcal{A}_t\}\right]. \tag{23}$$

For the first double sum above, we repeat the arguments of Appendix F using the episode-specific learning rates $\{\eta_t\}_{t=t_\ell}^{t_{\ell+1}-1}$. Crucially, we note that, no matter which base algorithm $\text{BOSSE}(t_{\text{start}}, m)$ is active at round $t \in [t_\ell, t_{\ell+1}), \eta_t \ge K^{1/3} \cdot (t - t_\ell)^{-1/3}$. Thus, we have

$$K \sum_{t=t_\ell}^{t_\ell^a-1} \eta_t^{-1} \le K^2 + c_6 \cdot K^{2/3} \cdot (t_\ell^a - t_\ell)^{4/3},$$

$$K \max_{t\in[t_\ell, t_\ell^a-1]} \eta_t^{-1} \le K^{2/3} \cdot (t_\ell^a - t_\ell)^{1/3}.$$

Then, using Lemma 17 and following the arguments of Appendix F:

$$\mathbf{1}\{\mathcal{E}_1\} \sum_{a=1}^{K} \sum_{t=t_\ell}^{t_\ell^a-1} \frac{\delta_t^{\mathbf{w}}(a_\ell, a)}{|\mathcal{A}_t|} \le c_7 \log(T) \cdot K^{1/3} \cdot (t_{\ell+1} - t_\ell)^{2/3}.$$

For the second double sum in (23), our aim is to bound the regret of playing arm $a$ on the rounds $t \in [t_\ell^a, t_{\ell+1})$ where a replay is active and, thus, reintroduces arm $a$ to $\mathcal{A}_t$ after it has been evicted from $\mathcal{A}_{\text{global}}$. For this, we rely on a careful decomposition of the rounds in $[t_\ell^a, t_{\ell+1})$ when $a \in \mathcal{A}_t$ based on which replay reintroduces the arm $a$ to the active set. We'll first require some definitions, repeated from the analyses of Suk & Agarwal (2023); Suk & Kpotufe (2022).

We first note that the various base algorithms scheduled in the process of running METABOSSE have an ancestor-parent-child structure determined by which instance of BOSSE calls another. Keeping this in mind, we now set up the following terminology (which is all w.r.t. a fixed arm $a$):

**Definition 5.**

    *(i) For each scheduled and activated* BOSSE$(s, m)$*, let the round* $M(s, m)$ *be the minimum of two quantities: (a) the last round in* $[s, s+m]$ *when arm $a$ is retained by* BOSSE$(s, m)$ *and all of its children, and (b) the last round that* BOSSE$(s, m)$ *is active and not permanently interrupted. Call the interval* $[s, M(s, m)]$ *the* **active interval** *of* BOSSE$(s, m)$.

    *(ii) Call a replay* BOSSE$(s, m)$ **proper** *if there is no other scheduled replay* BOSSE$(s', m')$ *such that* $[s, s+m] \subset (s', s'+m')$ *where* BOSSE$(s', m')$ *will become active again after round $s+m$. In other words, a proper replay is not scheduled inside the scheduled range of rounds of another replay. Let* PROPER$(t_\ell, t_{\ell+1})$ *be the set of proper replays scheduled to start before round $t_{\ell+1}$.*

    *(iii) Call a scheduled replay* BOSSE$(s, m)$ **subproper** *if it is non-proper and if each of its ancestor replays (i.e., previously scheduled replays whose durations have not concluded)* BOSSE$(s', m')$ *satisfies* $M(s', m') < s$. *In other words, a subproper replay either permanently interrupts its parent or does not, but is scheduled after its parent (and all its ancestors) stops playing arm $a$. Let* SUBPROPER$(t_\ell, t_{\ell+1})$ *be the set of all subproper replays scheduled before round $t_{\ell+1}$.*

Equipped with this language, we now show some basic claims which essentially reduce analyzing the complicated hierarchy of replays to analyzing the active intervals of replays in PROPER$(t_\ell, t_{\ell+1}) \cup$ SUBPROPER$(t_\ell, t_{\ell+1})$.

**Proposition 21.** *The active intervals*

$$\{[s, M(s, m)] : \text{BOSSE}(s, m) \in \text{PROPER}(t_\ell, t_{\ell+1}) \cup \text{SUBPROPER}(t_\ell, t_{\ell+1})\},$$

*are mutually disjoint.*

*Proof.* Clearly, the classes of replays PROPER$(t_\ell, t_{\ell+1})$ and SUBPROPER$(t_\ell, t_{\ell+1})$ are disjoint. Next, we show the respective active intervals $[s, M(s, m)]$ and $[s', M(s', m')]$ of any two BOSSE$(s, m)$, BOSSE$(s', m') \in$ PROPER$(t_\ell, t_{\ell+1}) \cup$ SUBPROPER$(t_\ell, t_{\ell+1})$ are disjoint. There are three cases here:

1. Proper replay vs. subproper replay: a subproper replay can only be scheduled after the round $M(s, m)$ of the most recent proper replay BOSSE$(s, m)$ (which is necessarily an ancestor). Thus, the active intervals of proper replays and subproper replays are disjoint.

2. Two distinct proper replays: two such replays can only permanently interrupt each other, and since $M(s, m)$ always occurs before the permanent interruption of BOSSE$(s, m)$, we have the active intervals of two such replays are disjoint.

3. Two distinct subproper replays: consider two non-proper replays

$$\text{BOSSE}(s, m), \text{BOSSE}(s', m') \in \text{SUBPROPER}(t_\ell, t_{\ell+1}),$$

with $s' > s$. The only way their active intervals intersect is if BOSSE$(s, m)$ is an ancestor of BOSSE$(s', m')$. Then, if BOSSE$(s', m')$ is subproper, we must have $s' > M(s, m)$, which means that $[s', M(s', m')]$ and $[s, M(s, m)]$ are disjoint.

$\square$

Next, we claim that the active intervals $[s, M(s, m)]$ for $\mathsf{BOSSE}(s, m) \in \text{PROPER}(t_\ell, t_{\ell+1}) \cup \text{SUBPROPER}(t_\ell, t_{\ell+1})$ contain all the rounds where $a$ is played after being evicted from $\mathcal{A}_{\text{global}}$. To show this, we first observe that for each round $t$ when a replay is active, there is a unique proper replay associated to $t$, namely the proper replay scheduled most recently. Next, note that any round $t > t_\ell^a$ where arm $a \in \mathcal{A}_t$ must either belong to the active interval $[s, M(s, m)]$ of the unique proper replay $\mathsf{BOSSE}(s, m)$ associated to round $t$, or else satisfies $t > M(s, m)$ in which case a unique subproper replay $\mathsf{BOSSE}(s', m') \in \text{SUBPROPER}(t_\ell, t_{\ell+1})$ is active at round $t$ and not yet permanently interrupted. Thus, it must be the case that $t \in [s', M(s', m')]$.

At the same time, every round $t \in [s, M(s, m)]$ for a proper or subproper $\mathsf{BOSSE}(s, m)$ is clearly a round where arm $a$ is once again active, i.e. $a \in \mathcal{A}_t$, and no such round is accounted for twice by Proposition 21. Thus,

$$\{t \in [t_\ell^a, t_{\ell+1}) : a \in \mathcal{A}_t\} = \bigsqcup_{\mathsf{BOSSE}(s,m) \in \text{PROPER}(t_\ell, t_{\ell+1}) \cup \text{SUBPROPER}(t_\ell, t_{\ell+1})} [s, M(s, m)].$$

Then, we can rewrite the second double sum in (23) as:

$$\sum_{a=1}^{K} \sum_{\mathsf{BOSSE}(s,m) \in \text{PROPER}(t_\ell, t_{\ell+1}) \cup \text{SUBPROPER}(t_\ell, t_{\ell+1})} B_{s,m} \sum_{t=s \vee t_\ell^a}^{M(s,m)} \frac{\delta_t^{\mathbf{w}}(a_\ell, a)}{|\mathcal{A}_t|}.$$

Further bounding the sum over $t$ above by its positive part, we can expand the sum over $\mathsf{BOSSE}(s, m) \in \text{PROPER}(t_\ell, t_{\ell+1}) \cup \text{SUBPROPER}(t_\ell, t_{\ell+1})$ to be over all scheduled $\mathsf{BOSSE}(s, m)$, or obtain:

$$\sum_{a=1}^{K} \sum_{\mathsf{BOSSE}(s,m)} B_{s,m} \left( \sum_{t=s \vee t_\ell^a}^{M(s,m)} \frac{\delta_t(a_\ell, a)}{|\mathcal{A}_t|} \cdot \mathbf{1}\{a \in \mathcal{A}_t\} \right)_+, \tag{24}$$

where the sum is over all replays $\mathsf{BOSSE}(s, m)$, i.e. $s \in \{t_\ell + 1, \ldots, t_{\ell+1} - 1\}$ and $m \in \{2, 4, \ldots, 2^{\lceil \log(T) \rceil}\}$. It then remains to bound the contributed relative regret of each $\mathsf{BOSSE}(s, m)$ in the interval $[s \vee t_\ell^a, M(s, m)]$, which will follow similarly to the previous steps. Fix $s, m$ and suppose $t_\ell^a + 1 \leq M(s, m)$ since otherwise $\mathsf{BOSSE}(s, m)$ contributes no regret in (24).

Note that the global learning rate $\eta_t$ for $t \in [s \vee t_\ell^a, M(s, m))$ must satisfy $\eta_t \geq K^{1/3} \cdot m^{-1/3}$ since any child base algorithm $\mathsf{BOSSE}(s', m')$ of $\mathsf{BOSSE}(s, m)$ will use a larger learning rate $\gamma_{t-s'} \geq \gamma_{t-s}$.

Then, following similar reasoning as before, i.e. combining our concentration bound (13) with the eviction criterion (4), we have for a fixed arm $a$:

$$\mathbf{1}\{\mathcal{E}_1\} \sum_{t=s \vee t_\ell^a}^{M(s,m)} \frac{\delta_t^{\mathbf{w}}(a_\ell, a)}{|\mathcal{A}_t|} \leq \frac{c_8 \log(T) \cdot (K^{1/3} \cdot m^{2/3} \wedge m)}{\min_{t \in [s, M(s,m)]} |\mathcal{A}_t|},$$

Plugging this into (24) and switching the ordering of the outer double sum, we obtain (we also overload the notation $M(s, m, a)$ to avoid ambiguity on which arm $a$ we're bounding the regret of):

$$\sum_{\mathsf{BOSSE}(s,m)} B_{s,m} \cdot c_8 \log(T) \cdot (K^{1/3} \cdot m^{2/3} \wedge m) \sum_{a=1}^{K} \frac{1}{\min_{t \in [s, M(s,m.a)]} |\mathcal{A}_t|}.$$

Following previous arguments, the above innermost sum over $a$ is at most $\log(K)$ since the $k$-th arm $a_k$ in $[K]$ to be evicted by $\mathsf{BOSSE}(s, m)$ satisfies $\min_{t \in [s, M(s,m,a_k)]} |\mathcal{A}_t| \geq K + 1 - k$.

Now, let $R(m) := c_8 \log(K) \cdot \log(T) \cdot (K^{1/3} \cdot m^{2/3} \wedge m)$ which is the bound we've obtained so far on the relative regret for a single $\mathsf{BOSSE}(s, m)$. Now, plugging $R(m)$ into (24) gives:

$$\mathbb{E}\left[ \mathbf{1}\{\mathcal{E}_1\} \sum_{a=1}^{K} \sum_{t=t_\ell^a}^{t_{\ell+1}-1} \frac{\delta_t^{\mathbf{w}}(a_\ell, a)}{|\mathcal{A}_t|} \cdot \mathbf{1}\{a \in \mathcal{A}_t\} \right] \leq \mathbb{E}_{t_\ell}\left[ \mathbb{E}\left[ \sum_{\mathsf{BOSSE}(s,m)} B_{s,m} \cdot R(m) \mid t_\ell \right] \right]$$

$$= \mathbb{E}_{t_\ell}\left[ \sum_{s=t_\ell}^{T} \sum_{m} \mathbb{E}[B_{s,m} \cdot \mathbf{1}\{s < t_{\ell+1}\} \mid t_\ell] \cdot R(m) \right].$$

Next, we observe that $B_{s,m}$ and $\mathbf{1}\{s < t_{\ell+1}\}$ are independent conditional on $t_\ell$ since $\mathbf{1}\{s < t_{\ell+1}\}$ only depends on the scheduling and observations of base algorithms scheduled before round $s$. Thus, recalling that $\mathbb{P}(B_{s,m} = 1) = m^{-1/3} \cdot (s - t_\ell)^{-2/3}$,

$$\mathbb{E}[B_{s,m} \cdot \mathbf{1}\{s < t_{\ell+1}\} \mid t_\ell] = \mathbb{E}[B_{s,m} \mid t_\ell] \cdot \mathbb{E}[\mathbf{1}\{s < t_{\ell+1}\} \mid t_\ell]$$
$$= \frac{1}{m^{1/3} \cdot (s - t_\ell)^{2/3}} \cdot \mathbb{E}[\mathbf{1}\{s < t_{\ell+1}\} \mid t_\ell].$$

Then, plugging the above display into our expectation from before and unconditioning, we obtain:

$$\mathbb{E}\left[\sum_{s=t_\ell+1}^{t_{\ell+1}-1} \sum_{n=1}^{\lceil \log(T) \rceil} \frac{1}{2^{n/3} \cdot (s - t_\ell)^{2/3}} \cdot R(2^n)\right] \leq c_9 \log(K) \cdot \log^2(T) \cdot \mathbb{E}_{t_\ell, t_{\ell+1}} \left[K^{1/3} \cdot (t_{\ell+1} - t_\ell)^{2/3}\right]. \quad (25)$$

Note that in the above, we used the fact that the episode length $t_{\ell+1} - t_\ell$ dominates $K$ by Lemma 17 to bound a term of order $K^{2/3} \cdot (t_{\ell+1} - t_\ell)^{1/3}$ by $K^{1/3} \cdot (t_{\ell+1} - t_\ell)^{2/3}$.

### H.4 Bounding per-Episode Regret of the Last Global Arm to the Last Safe Arm

Now, it remains to bound $\mathbb{E}\left[\sum_{t=t_\ell}^{t_{\ell+1}-1} \delta_t^{\mathbf{w}}(a_t^\sharp, a_\ell)\right]$. For this, we'll use a *bad segment analysis* similar to Suk & Agarwal (2023); Buening & Saha (2023); Suk & Kpotufe (2022). Roughly a bad segment $[s_1, s_2]$ will be such that $\sum_{s=s_1}^{s_2} \delta_s^{\mathbf{w}}(a_t^\sharp, a_\ell) \gtrsim (s_2 - s_1)^{2/3} \cdot K^{1/3}$. On such a bad segment $[s_1, s_2]$, a *perfect replay* is roughly defined as a replay whose scheduled duration overlaps most of $[s_1, s_2]$. Then, the key fact is that the randomized scheduling of replays (Line 12 of Algorithm 1) ensures that a perfect replay is scheduled for some bad segment (thus evicting $a_\ell$ from $\mathcal{A}_{\text{global}}$ before too many bad segments elapse, giving us a way of bounding $\sum_{s=s_1}^{s_2} \delta_s^{\mathbf{w}}(a_t^\sharp, a_\ell)$ with high probability.

We next formally define such notions. In what follows, bad segments will be defined with respect to a fixed arm $a$ and conditional on the episode start time $t_\ell$. In particular, this will hold for $a = a_\ell$ which will ultimately be used to bound $\delta_t^{\mathbf{w}}(a_t^\sharp, a_\ell)$ across the episode $[t_\ell, t_{\ell+1})$. The following definition only depends on a fixed arm $a$ and the episode start time $t_\ell$ and, conditional on these quantities, are deterministic given the environment.

**Definition 6.** *Fix the episode start time $t_\ell$, and let $[\tau_i, \tau_{i+1})$ be any phase intersecting $[t_\ell, T)$. For any arm $a$, define rounds $s_{i,0}(a), s_{i,1}(a), s_{i,2}(a) \ldots \in [t_\ell \vee \tau_i, \tau_{i+1})$ recursively as follows: let $s_{i,0}(a) := t_\ell \vee \tau_i$ and define $s_{i,j}(a)$ as the smallest round in $(s_{i,j-1}(a), \tau_{i+1})$ such that arm $a$ satisfies for some fixed $c_{10} > 0$:*

$$\sum_{t=s_{i,j-1}(a)}^{s_{i,j}(a)} \delta_t^{\mathbf{w}}(a_t^\sharp, a) \geq c_{10} \log(T) \cdot K^{1/3} \cdot (s_{i,j}(a) - s_{i,j-1}(a))^{2/3}, \quad (26)$$

*if such a round $s_{i,j}(a)$ exists. Otherwise, we let the $s_{i,j}(a) := \tau_{i+1} - 1$. We refer to any interval $[s_{i,j-1}(a), s_{i,j}(a))$ as a **critical segment**, and as a **bad segment** (w.r.t. arm a) if (26) above holds.*

**Remark 4.** *Arm $a_t^\sharp$ is fixed within any critical segment $[s_{i,j-1}(a), s_{i,j}(a)) \subseteq [\tau_i, \tau_{i+1})$ since an SKW does not occur inside $[\tau_i, \tau_{i+1})$.*

The following fact, which may be considered an analogue of Lemma 17, will serve useful.

**Fact 22.** *A bad segment $[s_{i,j}(a), s_{i,j+1}(a))$ satisfies $s_{i,j+1}(a) - s_{i,j}(a) \geq K/8$.*

Now, note a bad segment $[s_{i,j}(a), s_{i,j+1}(a))$ only contributes order $K^{1/3} \cdot (s_{i,j+1}(a) - s_{i,j}(a))^{2/3}$ regret of $a$ to $a_t^\sharp$. At the same time, we claim that a well-timed replay (see Definition 7 below) running from $s_{i,j}(a)$ to $s_{i,j+1}(a)$ will in fact be capable of evicting arm $a$ using (4). This will allow us to reduce the problem to studying the number and lengths of bad segments which elapse before one is detected by such a replay.

We next define a *perfect replay*.

**Definition 7.** *Let $\tilde{s}_{i,j}(a) := \lceil \frac{s_{i,j}(a) + s_{i,j+1}(a)}{2} \rceil$ denote the approximate midpoint of $[s_{i,j}(a), s_{i,j+1}(a))$. Given a bad segment $[s_{i,j}(a), s_{i,j+1}(a))$, define a **perfect replay** w.r.t. $[s_{i,j}(a), s_{i,j+1}(a))$ as a call of $\mathsf{BOSSE}(t_{\text{start}}, m)$ where $t_{\text{start}} \in [s_{i,j}(a), \tilde{s}_{i,j}(a)]$ and $m \geq s_{i,j+1}(a) - s_{i,j}(a)$*

Next, we analyze the behavior of a perfect replay on the bad segment $[s_{i,j}(a), s_{i,j+1}(a))$. We first invoke an elementary lemma:

**Lemma 23.** $(x+y)^{2/3} - x^{2/3} \geq y^{2/3}/2$ for real numbers $y \geq x > 0$.

*Proof.* This follows from noting the derivative of the function $y \mapsto (x+y)^{2/3} - x^{2/3} - y^{2/3}/2$ in the domain $y \geq x$ is positive since

$$\frac{\partial}{\partial y}(x+y)^{2/3} - x^{2/3} - y^{2/3}/2 = \frac{2}{3 \cdot (x+y)^{1/3}} - \frac{1}{3 \cdot y^{1/3}}.$$

The above is positive since

$$\frac{1}{(x/y+1)^{1/3}} \geq \frac{1}{2^{1/3}} > \frac{1}{2} \implies \frac{2}{3 \cdot (x+y)^{1/3}} > \frac{1}{3 \cdot y^{1/3}}.$$

Thus, $y \mapsto (x+y)^{2/3} - x^{2/3} - y^{2/3}/2$ is an increasing function in $y$. Next, since $(2x)^{2/3} - x^{2/3} - x^{2/3}/2 = x^{2/3} \cdot (2^{2/3} - 1 - 1/2) > 0$ if $x > 0$, we have that the desired inequality must always be true for $y \geq x > 0$. $\square$

**Proposition 24.** *Suppose the good event $\mathcal{E}_1$ holds (cf. Proposition 16). Let $[s_{i,j}(a), s_{i,j+1}(a))$ be a bad segment with respect to arm $a$. Then, if a perfect replay with respect to $[s_{i,j}(a), s_{i,j+1}(a))$ is scheduled, arm $a$ will be evicted from $\mathcal{A}_{\mathrm{global}}$ by round $s_{i,j+1}(a)$.*

*Proof.* We first observe that by Lemma 23 and Definition 6:

$$\sum_{t=\tilde{s}_{i,j}(a)}^{s_{i,j+1}(a)} \delta_t^{\mathbf{w}}(a_t^{\sharp}, a) = \sum_{t=s_{i,j}(a)}^{s_{i,j+1}(a)} \delta_t^{\mathbf{w}}(a_t^{\sharp}, a) - \sum_{t=s_{i,j}(a)}^{\tilde{s}_{i,j}(a)-1} \delta_t^{\mathbf{w}}(a_t^{\sharp}, a) \geq \frac{c_{10}}{2}\log(T) \cdot K^{1/3} \cdot (s_{i,j+1}(a) - \tilde{s}_{i,j}(a))^{2/3}.$$

Next, we note that any perfect replay $\mathsf{BOSSE}(t_{\mathrm{start}}, m)$ will not evict $a_t^{\sharp}$ since otherwise it incurs significant regret within SKW phase $[\tau_i, \tau_{i+1})$ (see also the proof of Lemma 20). The same applies for any child base algorithm of a perfect replay.

Next, we argue that arm $a$ must be evicted from $\mathcal{A}_{\mathrm{global}}$ at some round in $[\tilde{s}_{i,j}(a), s_{i,j+1}(a)]$. If $a \notin \mathcal{A}_t$ for some round $t \in [\tilde{s}_{i,j}(a), s_{i,j+1}(a)]$ we are already done. Otherwise, suppose $a \in \mathcal{A}_t$ for all $t \in [\tilde{s}_{i,j}(a), s_{i,j+1}(a)]$ and so we must have $\mathbb{E}[\hat{\delta}_t^{\mathbf{w}}(a_t^{\sharp}, a)|\mathcal{F}_{t-1}] = \delta_t^{\mathbf{w}}(a_t^{\sharp}, a)$ for all such rounds $t$. Now, if a perfect replay is scheduled, then we must have a global learning rate $\eta_t \geq K^{1/3} \cdot (s_{i,j+1}(a) - s_{i,j}(a))^{-1/3}$ for all $t \in [\tilde{s}_{i,j}(a), s_{i,j+1}(a)]$ since any child base algorithm of a perfect replay can only set a larger learning rate by Definition 4.

Now, noting $[s_{i,j}(a), s_{i,j+1}(a))$ and the second half of the bad segment $[s_{i,j}(a), s_{i,j+1}(a))$ have commensurate lengths up to constants, this means, if a perfect replay w.r.t. $[s_{i,j}(a), s_{i,j+1}(a))$ is scheduled, by similar calculations to earlier (and using Fact 22) we must have

$$\sqrt{K \sum_{s=\tilde{s}_{i,j}(a)}^{s_{i,j+1}(a)} \eta_s^{-1}} + K \cdot \max_{s \in [\tilde{s}_{i,j}(a), s_{i,j+1}(a)]} \eta_s^{-1} \leq c_{11} K^{1/3} \cdot (s_{i,j+1}(a) - s_{i,j}(a))^{2/3}.$$

Then, by our eviction criterion (4) and concentration, we have that arm $a$ will be evicted over $[\tilde{s}_{i,j}(a), s_{i,j+1}(a)]$ for large enough constant $c_{10}$ in Definition 6. $\square$

It remains to show that, for any arm $a$, a perfect replay is scheduled w.h.p. before too much regret is incurred on the elapsed bad segments w.r.t. $a$. In particular, this will hold for the last global arm $a_\ell$, allowing us to bound the remaining expectation $\mathbb{E}[\sum_{t=t_\ell}^{t_{\ell+1}-1} \delta_t^{\mathbf{w}}(a_t^{\sharp}, a_\ell)]$.

To show this, we'll define a *bad round* $s(a) > t_\ell$ which will roughly be the latest time that arm $a$ can be evicted by a perfect replay before there is too much regret. We'll then show that a perfect replay with respect to some bad segment is indeed scheduled before the bad round is reached.

First, fix an arm $a$ and an episode start time $t_\ell$.

**Definition 8.** *(Bad Round) For a fixed round $t_\ell$ and arm $a$, the* **bad round** $s(a) > t_\ell$ *is defined as the smallest round which satisfies, for some fixed $c_{12} > 0$:*

$$\sum_{(i,j)} (s_{i,j+1}(a) - s_{i,j}(a))^{2/3} > c_{12} \log(T) \cdot (s(a) - t_\ell)^{2/3}, \tag{27}$$

*where the above sum is over all pairs of indices $(i,j) \in \mathbb{N} \times \mathbb{N}$ such that $[s_{i,j}(a), s_{i,j+1}(a))$ is a bad segment (see Definition 6) with $s_{i,j+1}(a) < s(a)$.*

Our goal is then to then to show that arm $a$ is evicted by some perfect replay scheduled within episode $[t_\ell, t_{\ell+1})$ with high probability before the bad round $s(a)$ occurs.

For each bad segment $[s_{i,j}(a), s_{i,j+1}(a))$, recall that $\tilde{s}_{i,j}(a)$ is the approximate midpoint between $s_{i,j}(a)$ and $s_{i,j+1}(a)$ (see Definition 7). Next, let $m_{i,j} := 2^n$ where $n \in \mathbb{N}$ satisfies:

$$2^n \geq s_{i,j+1}(a) - s_{i,j}(a) > 2^{n-1}.$$

Plainly, $m_{i,j}$ is a dyadic approximation of the bad segment length. Next, recall that the Bernoulli $B_{t,m}$ decides whether $\mathsf{BOSSE}(t,m)$ is scheduled at round $t$ (see Line 6 of Algorithm 2). If for some $t \in [s_{i,j}(a), \tilde{s}_{i,j}(a)]$, $B_{t,m_{i,j}} = 1$, i.e. a perfect replay is scheduled, then $a$ will be evicted from $\mathcal{A}_{\text{global}}$ by round $s_{i,j+1}(a)$ (Proposition 24). We will show this happens with high probability via concentration on the sum

$$X(a, t_\ell) := \sum_{(i,j):s_{i,j+1}(a) < s(a)} \sum_{t=s_{i,j}(a)}^{\tilde{s}_{i,j}(a)} B_{t,m_{i,j}},$$

Note that the random variable $X(a, t_\ell)$ only depends on the replay scheduling probabilities $\{B_{s,m}\}_{s,m}$ given a fixed arm $a$ and episode start time $t_\ell$, since the bad round $s(a)$ is also fixed given these quantities. This means that $X(a, t_\ell)$ is an independent sum of Bernoulli random variables $B_{t,m_{i,j}}$, conditional on $t_\ell$. Then, a multiplicative Chernoff bound over the randomness of $X(a, t_\ell)$, conditional on $t_\ell$ yields

$$\mathbb{P}\left( X(a, t_\ell) \leq \frac{\mathbb{E}[X(a, t_\ell) \mid t_\ell]}{2} \mid t_\ell \right) \leq \exp\left( -\frac{\mathbb{E}[X(a, t_\ell) \mid t_\ell]}{8} \right).$$

The above RHS error probability is bounded above above by $1/T^3$ by observing:

$$\mathbb{E}\left[ X(a, t_\ell) \mid t_\ell \right] \geq \sum_{(i,j)} \sum_{t=s_{i,j}(a)}^{\tilde{s}_{i,j}(a)} \frac{1}{m_{i,j}^{1/3} \cdot (t - t_\ell)^{2/3}} \geq \frac{1}{2} \sum_{(i,j)} \frac{(s_{i,j+1}(a) - s_{i,j}(a))^{2/3}}{(s(a) - t_\ell)^{2/3}} \geq \frac{c_{12}}{2} \log(T),$$

for $c_{12} > 0$ large enough, where the last inequality follows from (27) in the definition of the bad round $s(a)$ (Definition 8). Taking a further union bound over the choice of arm $a \in [K]$ gives us that $X(a, t_\ell) > 1$ for all choices of arm $a$ (define this as the good event $\mathcal{E}_2(t_\ell)$) with probability at least $1 - K/T^3$. Thus, under event $\mathcal{E}_2(t_\ell)$, all arms $a$ will be evicted before round $s(a)$ with high probability.

Recall on the event $\mathcal{E}_1$ the concentration bounds of Proposition 16 hold. Then, on $\mathcal{E}_1 \cap \mathcal{E}_2(t_\ell)$, letting $a = a_\ell$ in the preceding arguments we must have $t_{\ell+1} - 1 \leq s(a_\ell)$ Thus, by the definition of the bad round $s(a_\ell)$ (Definition 8), we must have:

$$\sum_{[s_{i,j}(a_\ell), s_{i,j+1}(a_\ell)):s_{i,j+1}(a_\ell) < t_{\ell+1}-1} (s_{i,j+1}(a_\ell) - s_{i,j}(a_\ell))^{2/3} \leq c_{12} \log(T) \cdot (t_{\ell+1} - t_\ell)^{2/3}. \tag{28}$$

Thus, by (26) in the definition of bad segments (Definition 6), over the bad segments $[s_{i,j}(a_\ell), s_{i,j+1}(a_\ell))$ which elapse before the end of the episode $t_{\ell+1} - 1$, the regret of $a_\ell$ to $a_t^\sharp$ is at most order $\log^2(T)K^{1/3} \cdot (t_{\ell+1} - t_\ell)^{2/3}$.

Over each non-bad critical segment $[s_{i,j}(a_\ell), s_{i,j+1}(a_\ell))$, the regret of playing arm $a_\ell$ to $a_t^\sharp$ is at most $\log(T) \cdot K^{1/3} \cdot (\tau_{i+1} - \tau_i)^{2/3}$ and there is at most one non-bad critical segment per phase $[\tau_i, \tau_{i+1})$ (follows from Definition 6).

So, we conclude that on event $\mathcal{E}_1 \cap \mathcal{E}_2(t_\ell)$:

$$\sum_{t=t_\ell}^{t_{\ell+1}-1} \delta_t^{\mathbf{w}}(a_t^\sharp, a_\ell) \leq c_{13} \log^2(T) \sum_{i \in \text{PHASES}(t_\ell, t_{\ell+1})} K^{1/3}(\tau_{i+1} - \tau_i)^{2/3}.$$

Taking expectation, we have by conditioning first on $t_\ell$ and then on event $\mathcal{E}_1 \cap \mathcal{E}_2(t_\ell)$:

$$\mathbb{E}\left[\sum_{t=t_\ell}^{t_{\ell+1}-1} \delta_t^{\mathbf{w}}(a_t^\sharp, a_\ell)\right] \leq \mathbb{E}_{t_\ell}\left[\mathbb{E}\left[\mathbf{1}\{\mathcal{E}_1 \cap \mathcal{E}_2(t_\ell)\} \sum_{t=t_\ell}^{t_{\ell+1}-1} \delta_t^{\mathbf{w}}(a_t^\sharp, a_\ell) \mid t_\ell\right]\right] + T \cdot \mathbb{E}_{t_\ell}\left[\mathbb{E}\left[\mathbf{1}\{\mathcal{E}_1^c \cup \mathcal{E}_2^c(t_\ell)\} \mid t_\ell\right]\right]$$

$$\leq c_{13} \log^2(T)\mathbb{E}_{t_\ell}\left[\mathbb{E}\left[\mathbf{1}\{\mathcal{E}_1 \cap \mathcal{E}_2(t_\ell)\} \sum_{i \in \text{PHASES}(t_\ell, t_{\ell+1})} K^{1/3} \cdot (\tau_{i+1} - \tau_i)^{2/3} \mid t_\ell\right]\right] + \frac{2K}{T^2}$$

$$\leq c_{13} \log^2(T)\mathbb{E}\left[\mathbf{1}\{\mathcal{E}_1\} \sum_{i \in \text{PHASES}(t_\ell, t_{\ell+1})} K^{1/3} \cdot (\tau_{i+1} - \tau_i)^{2/3}\right] + \frac{2}{T},$$

where in the last step we bound $\mathbf{1}\{\mathcal{E}_1 \cap \mathcal{E}_2(t_\ell)\} \leq \mathbf{1}\{\mathcal{E}_1\}$ and apply tower law again. This concludes the proof.
∎

## H.5 Total Variation Regret Rates for Known Weights (Proof of Corollary 13)

**Re-Defining Total Variation in Terms of Weighted Borda Scores.** Recall the total variation quantity is defined as:

$$V_T := \sum_{t=2}^{T} \max_{a,a'} |\delta_t(a, a') - \delta_{t-1}(a, a')|,$$

where $\delta_t(a, a') \doteq P_t(a, a') - \frac{1}{2}$ is the gap in dueling preferences. We first note that this can be rewritten as $\sum_{t=2}^{T} \max_{a \in [K], \mathbf{w} \in \Delta^K} |b_t(a, \mathbf{w}) - b_{t-1}(a, \mathbf{w})|$ since by Jensen:

$$\max_{\mathbf{w} \in \Delta^K} |b_t(a, \mathbf{w}) - b_{t-1}(a, \mathbf{w})| \leq \max_{\mathbf{w} \in \Delta^K} \mathbb{E}_{a' \sim \mathbf{w}}[|\delta_t(a, a') - \delta_{t-1}(a, a')|] \leq \max_{a,a'} |\delta_t(a, a') - \delta_{t-1}(a, a')|.$$

Note the other directions of the above inequalities are clear by taking $\mathbf{w} = \mathbf{w}(a')$. Thus, we may redefine the total variation in terms of weighted Borda scores:

$$V_T := \sum_{t=2}^{T} \max_{\mathbf{w} \in \Delta^K} |b_t(a, \mathbf{w}) - b_{t-1}(a, \mathbf{w})|.$$

**Proof of Corollary 13.** First, we bound the total variation $V_{[\tau_i, \tau_{i+1})}$ over an SKW phase $[\tau_i, \tau_{i+1})$. Consider the arm $a_{\tau_{i+1}}^*(\mathbf{w})$. By Definition 1, there must exist a round $t \in [\tau_i, \tau_{i+1})$ such that

$$b_t(a_t^*(\mathbf{w}), \mathbf{w}) - b_t(a_{\tau_{i+1}}^*(\mathbf{w}), \mathbf{w}) > \left(\frac{K}{\tau_{i+1} - \tau_i}\right)^{1/3}.$$

Adding $b_{\tau_{i+1}}(a_{\tau_{i+1}}^*(\mathbf{w}), \mathbf{w}) - b_{\tau_{i+1}}(a_t^*(\mathbf{w}), \mathbf{w}) \geq 0$ to the RHS we obtain that

$$V_{[\tau_i, \tau_{i+1})} \geq \left(\frac{K}{\tau_{i+1} - \tau_i}\right)^{1/3}.$$

Now, summing over SKW phases, we have by Hölder's inequality:

$$\sum_{i=0}^{\tilde{L}_{\text{Known}}} K^{1/3} \cdot (\tau_{i+1} - \tau_i)^{2/3} \leq \left(\sum_i \left(\frac{K}{\tau_{i+1} - \tau_i}\right)^{1/3}\right)^{1/4} \left(\sum_i (\tau_{i+1} - \tau_i) \cdot K^{1/3}\right)^{3/4} + K^{1/3} \cdot T^{2/3}$$

Thus, we obtain a total dynamic regret bound of $V_T^{1/4} \cdot T^{3/4} \cdot K^{1/4} + K^{1/3} \cdot T^{2/3}$.

# I Dynamic Regret Analysis for Unknown Weights

## I.1 Analysis Overview for Theorem 14

The proof of Theorem 14 will broadly follow the same outline as the regret analysis for known weights, but with replacements of the eviction threshold by $(s_2 - s_1)^{2/3} \cdot K^{2/3}$ (see the unknown weight specification in Definition 4), and bounding the variance of estimation by quantities of this order. The key difficulty for unknown weights is that the evaluation weights $\mathbf{w}_t$ may change at the unknown SUW shifts $\rho_i$.

Importantly, Algorithm 2 can only estimate aggregate gaps $\sum_{s=s_1}^{s_2} \delta_t^{\mathbf{w}}(a', a)$ over intervals $[s_1, s_2]$ with respect to a fixed and unchanging weight $\mathbf{w}$. Thus, the main novelty in this analysis is to carefully partition the regret analysis along intervals $[s_1, s_2]$ lying within a phase $[\rho_i, \rho_{i+1})$. In fact, such a strategy is already inherent to the bad segment argument done in Appendix H.4 to bound $\sum_{t=t_\ell}^{t_{\ell+1}-1} \delta_t^{\mathbf{w}}(a_t^\sharp, a_\ell)$ for a known and known weight $\mathbf{w}$. So, we'll repeat a similar such bad segment analysis, but for different arms and even for different base algorithms.

This strategy is similar to the approach taken in the Condorcet winner dynamic regret analysis of Buening & Saha (2023, see Appendix A.3 therein), who rely on such a tactic to avoid decomposing the regret using triangle inequalities as done in the original non-stationary MAB analysis of Suk & Kpotufe (2022). Our need for this strategy is different, as we must constrain ourselves to only being able to detect that an arm $a$ is bad over segments $[s_1, s_2]$ of rounds where we can use a single reference weight $\mathbf{w}$.

We first establish the analogue of Lemma 20 from Appendix H for the unknown weight setting.

## I.2 Episodes Align with SUW Phases

**Lemma 25.** *On event $\mathcal{E}_1$, for each episode $[t_\ell, t_{\ell+1})$ with $t_{\ell+1} \leq T$ (i.e., an episode which concludes with a restart), there exists an SUW shift $\rho_i \in [t_\ell, t_{\ell+1})$.*

*Proof.* This follows in an analogous manner as Lemma 20, where we note that significant SUW regret occurs if, for some weight $\mathbf{w} \in \Delta^K$, we have $\sum_{s=s_1}^{s_2} \delta_s^{\mathbf{w}}(a) \geq K^{2/3} \cdot (s_2 - s_1)^{2/3}$. Thus, an arm being evicted from $\mathcal{A}_{\text{global}}$ implies it has significant SUW regret meaning a new episode is triggered only when an SUW has occurred. □

In what follows, we redefine the *last safe arm* $a_t^\sharp$ as the last arm to become unsafe in the sense of Definition 2 in the unique SUW phase $[\rho_i, \rho_{i+1})$ containing round $t$.

## I.3 Generic Bad Segment Analysis

We'll first define a generic good event $\mathcal{E}_3$ over which the bad segment-type argument (as seen in Appendix H.4) holds for any arm $a$ and any episode start time $t_\ell$. Before we can define such an event, we'll generically redefine the necessary mathematical objects of Appendix H.4 for the unknown weight setting. We note that everything that follows in this subsection is independent of any observations or decisions made by Algorithm 2 and depend only on the possible random choices of replay schedules (see Line 12 of Algorithm 2), which may be instantiated independently and obliviously to the algorithm's actual behavior.

In what follows, fix a *starting round* $t_{\text{init}} \in [T]$ and an arm $a \in [K]$. Define the random variables $B_{s,m}^{t_{\text{init}}} \sim \text{Ber}(m^{-1/3} \cdot (s - t_{\text{init}})^{-2/3})$ for $m = 2, 4, \ldots, 2^{\lceil \log(T) \rceil}$ and $s = t_{\text{init}} + 1, \ldots, T$, which are the replay schedulers of Line 12 in Algorithm 2 if $t_{\text{init}}$ was the start of an episode. Also, fix a round $t_{\text{start}}$ from which we will begin defining bad segments; the variable $t_{\text{start}}$ will serve useful when we analyze bad segments for different base algorithms $\text{BOSSE}(t_{\text{start}}, m)$. The following definitions will then be relative to a fixed $t_{\text{init}}$, $t_{\text{start}}$, and arm $a$.

**Definition 9.** *(Generic Bad Segment) Fix an SUW phase $[\rho_i, \rho_{i+1})$ intersecting $[t_{\text{start}}, T]$. Define rounds $s_{i,0}, s_{i,1}, s_{i,2} \ldots \in [t_{\text{start}} \vee \rho_i, \rho_{i+1})$ recursively as follows: let $s_{i,0} := t_{\text{start}} \vee \rho_i$ and define $s_{i,j}$ as the smallest round in $(s_{i,j-1}, \rho_{i+1})$ such that arm $a$ satisfies for some fixed $c_{14} > 0$:*

$$\max_{a' \in [K]} \sum_{t=s_{i,j-1}}^{s_{i,j}} \delta_t^{\mathbf{w}(a')}(a_t^\sharp, a) \geq c_{14} \log(T) \cdot K^{2/3} \cdot (s_{i,j} - s_{i,j-1})^{2/3}, \tag{29}$$

*if such a round $s_{i,j}$ exists. Otherwise, we let the $s_{i,j} := \rho_{i+1} - 1$. We refer to any interval $[s_{i,j-1}, s_{i,j})$ as a* **critical segment**, *and as a* **bad segment** *if* (26) *above holds.*

**Remark 5.** *Note that* (29) *mimics the notion* (9) *of significant worse-case weighted Borda regret in Definition* 2. *However, we need only concern ourselves with checking for bad regret w.r.t. point-mass weights $\mathbf{w}(a')$ for $a' \in [K]$, as that will suffice for the analysis.*

**Definition 10.** *(Generic Bad Round) Define the* **bad round** $s(a, t_{\mathrm{start}}, t_{\mathrm{init}}) > t_{\mathrm{start}}$ *as the smallest round which satisfies, for some fixed $c_{15} > 0$:*

$$\sum_{(i,j)} (s_{i,j+1} - s_{i,j})^{2/3} > c_{15} \log(T) \cdot (s(a, t_{\mathrm{start}}, t_{\mathrm{init}}) - t_{\mathrm{init}})^{2/3}, \tag{30}$$

*where the above sum is over all pairs of indices $(i,j) \in \mathbb{N} \times \mathbb{N}$ such that $[s_{i,j}, s_{i,j+1})$ is a bad segment (see Definition* 6) *with $s_{i,j+1} < s(a, t_{\mathrm{start}}, t_{\mathrm{init}})$.*

Now, define the independent sum of Bernoulli's

$$X(a, t_{\mathrm{start}}, t_{\mathrm{init}}) := \sum_{(i,j): s_{i,j+1} < s(a, t_{\mathrm{start}}, t_{\mathrm{init}})} \sum_{t=s_{i,j}}^{\tilde{s}_{i,j}} B_{t, m_{i,j}}^{t_{\mathrm{init}}},$$

where $m_{i,j}$ is the dyadic approximation w.r.t. $[s_{i,j}(a), s_{i,j+1}(a))$ in the same sense as defined in Appendix H.4.

We are now prepared to define the good event $\mathcal{E}_3$, where all bad segment arguments hold, based on the following lemma.

**Lemma 26.** *Let $\mathcal{E}_3$ be the event over which for all $a \in [K]$ and rounds $t_{\mathrm{start}}, t_{\mathrm{init}} \in [T]$ with $t_{\mathrm{start}} \geq t_{\mathrm{init}}$, $X(a, t_{\mathrm{start}}, t_{\mathrm{init}}) > 1$. Then, over the randomness of all possible replay schedules, $\mathbb{P}(\mathcal{E}_3) > 1 - T^{-3}$.*

*Proof.* This follows from repeating the multiplicative Chernoff bound as used in Appendix H.4 for each $X(a, t_{\mathrm{start}}, t_{\mathrm{init}})$ and then taking union bounds over arms $a \in [K]$ and rounds $t_{\mathrm{init}} \in [T]$. We note that crucially all potential replay schedules $\{B_{s,m}^{t_{\mathrm{init}}}\}_{s,m,t_{\mathrm{init}}}$ are independent across different $s, m, t_{\mathrm{init}}$. $\qquad\square$

Next, we show that, under event $\mathcal{E}_3$, the regret of a fixed arm $a$ to $a_t^\sharp$ on the active interval $[s, M(s, m, a)]$ (see Definition 5 in Appendix H.3) of any scheduled base algorithm $\mathsf{BOSSE}(s, m)$ will be boundable by running a customized bad segment analysis using the notions defined above.

**Notation 27.** *(Active Interval of First Ancestor Base Algorithm) For the first ancestor base algorithm $\mathsf{BOSSE}(t_\ell, T + 1 - t_\ell)$ (i.e., that which is instantiated first in episode $[t_\ell, t_{\ell+1})$ per Line 7 of Algorithm 2), we'll let $M(t_\ell, T + 1 - t_\ell, a)$ denote the last round when $a$ is retained by $\mathsf{BOSSE}(t_\ell, T + 1 - t_\ell)$ and all of its children.*

**Definition 11.** *For an interval of rounds $I$, Let $\mathrm{PHASES}(I) \doteq \{i \in [\tilde{L}_{\mathrm{Unknown}}] : [\rho_i, \rho_{i+1}) \cap I \neq \emptyset\}$, i.e., denote those phases intersecting $I$. Let $S(I) := |\mathrm{PHASES}(I)|$ be the number of intersecting phases and let $L(I, i) := |[\rho_i, \rho_{i+1}) \cap I|$ be the intersection's length for phase $[\rho_i, \rho_{i+1})$.*

**Lemma 28.** *(Generic Bad Segment Analysis for $\mathsf{BOSSE}(s, m)$) For any scheduled $\mathsf{BOSSE}(s, m)$, letting $I := [s, M(s, m, a)]$ be its active interval, we have on event $\mathcal{E}_3 \cap \mathcal{E}_1$:*

$$\sum_{t=s}^{M(s,m,a)} \delta_t^{\mathbf{w}_t}(a_t^\sharp, a) \leq c_{16} \log^2(T) \left( K^{2/3} \cdot (s - t_\ell)^{2/3} \cdot \mathbf{1}\{S(I) > 1\} + \sum_{i \in \mathrm{PHASES}(I)} K^{2/3} \cdot L(I, i)^{2/3} \right).$$

*Proof.* For $(t_{\mathrm{init}}, t_{\mathrm{start}}) := (t_\ell, s)$, we can define perfect replays with respect to the generic bad segments of Definition 9 analogously to Definition 7. We next show an analogue of Proposition 24 for such perfect replays. In particular, for each bad segment $[s_{i,j}, s_{i,j+1})$, we have for some point-mass weight $\mathbf{w}$,

$$\sum_{t=\tilde{s}_{i,j}}^{s_{i,j+1}} \delta_t^{\mathbf{w}}(a_t^\sharp, a) \geq \frac{c_{17}}{2} \log(T) \cdot K^{2/3} \cdot (s_{i,j+1} - \tilde{s}_{i,j})^{2/3}.$$

Next, the key points hold if a perfect replay w.r.t. $[s_{i,j}, s_{i,j+1})$ is scheduled:

- Arm $a_t^\sharp$ (which is constant for $t \in [s_{i,j}, s_{i,j+1}) \subseteq [\rho_i, \rho_{i+1}))$ is not evicted from $\mathcal{A}_t$ since it does not incur significant SUW regret.

- The global learning rates $\eta_t$ set for $t \in [\tilde{s}_{i,j}, s_{i,j+1}]$ satisfy $\eta_t \geq K^{2/3} \cdot (s_{i,j+1} - s_{i,j})^{-1/3}$ since any child base algorithm can only increase the learning rate of its parent base algorithm.

- By similar calculations to the proof of Theorem 19 (see Appendix G) and using the fact that a generic bad segment, as defined in (29) must have length at least $K^2/8$ (analogous to Lemma 17), we have

$$\sqrt{K^2 \sum_{s=\tilde{s}_{i,j}}^{s_{i,j+1}} \eta_s^{-1} + K \cdot \max_{s \in [\tilde{s}_{i,j}, s_{i,j+1}]} \eta_s^{-1}} \leq c_{18} \cdot K^{2/3} \cdot (s_{i,j+1} - s_{i,j})^{2/3}.$$

Combining the above points with our concentration bound (15) and eviction criterion (4) give us that arm $a$ will be evicted from $\mathcal{A}_t$ by round $s_{i,j+1}$. By Lemma 28, we have that a perfect replay with respect to arm $a$, $t_{\text{start}} = s$, and $t_{\text{init}} = t_\ell$ will be scheduled before the bad round $s(a, s, t_\ell)$ on event $\mathcal{E}_3$. By (30), we then must have:

$$\sum_{(i,j)} (s_{i,j+1} - s_{i,j})^{2/3} \leq c_{15} \log(T) \cdot (M(s, m, a) - t_\ell)^{2/3},$$

where the sum is over pairs of indices $(i, j)$ representing these generic bad segments. Thus, the desired regret bound follows by similar arguments to Appendix H.4. Note that bounding the regret over the bad segments of multiple phases $[\rho_i, \rho_{i+1})$ is needed only if the number of intersecting phases $S(I) > 1$. Otherwise, we can avoid a bad segment analysis and follow the proof steps of Appendix F to directly bound the regret as order $K^{2/3} \cdot m^{2/3}$. $\qquad\square$

### I.4 Decomposing the Regret Along Different Base Algorithms

Equipped with the generic bad segment analysis for any base algorithm $\mathsf{BOSSE}(s, m)$, we're now ready to bound the per-episode regret. It will suffice to decompose the regret along active intervals $[s, M(s, m, a)]$ of different base algorithms $\mathsf{BOSSE}(s, m)$ in a similar fashion to Appendix H.3, within each of which we can plug in the regret bound of Lemma 28 and then carefully integrate with respect to the randomness of replay scheduling.

We may first decompose the regret as

$$\mathbb{E}\left[\sum_{t=t_\ell}^{t_{\ell+1}-1} \delta_t^{\mathbf{w}_t}(i_t) + \delta_t^{\mathbf{w}_t}(j_t)\right] = \mathbb{E}\left[\sum_{t=t_\ell}^{t_{\ell+1}-1} \delta_t^{\mathbf{w}_t}(a_t^\sharp)\right] + \mathbb{E}\left[\sum_{t=t_\ell}^{t_{\ell+1}-1} \delta_t^{\mathbf{w}_t}(a_t^\sharp, i_t) + \delta_t^{\mathbf{w}_t}(a_t^\sharp, j_t)\right].$$

The first expectation on the above RHS is bounded of the right order by Definition 2 and earlier arguments.

For the second expectation on the above RHS, we further condition on whether we're in exploration mode or playing from the candidate set $\mathcal{A}_t$. Following the steps of Appendix H.3, we have that it suffices to bound

$$\mathbb{E}\left[\sum_{a \in [K]} \sum_{t=t_\ell}^{t_{\ell+1}-1} \frac{\delta_t^{\mathbf{w}_t}(a_t^\sharp, a)}{|\mathcal{A}_t|} \cdot \mathbf{1}\{a \in \mathcal{A}_t\}\right].$$

In fact, following the same decomposition of rounds into active intervals of different base algorithms, we can further upper bound the above by

$$\mathbb{E}\left[\sum_{a=1}^{K} \sum_{\mathsf{BOSSE}(s,m)} B_{s,m} \left(\sum_{t=s}^{M(s,m,a)} \frac{\delta_t^{\mathbf{w}_t}(a_t^\sharp, a)}{|\mathcal{A}_t|}\right)_+\right].$$

Note that in the above we sum over all base algorithms including the "first ancestor" base algorithm $\mathsf{BOSSE}(t_\ell, T + 1 - t_\ell)$ using Notation 27 and for which we use the convention $B_{t_\ell, T+1-t_\ell} = 1$.

Next, we plug in the guarantee of Lemma 28. Let $I(s, m, a) := [s, M(s, m, a)]$ be the active interval of BOSSE$(s, m)$. Then, it remains to bound (hiding the added log terms from Lemma 28 for ease of notation):

$$\mathbb{E}\left[\sum_{\text{BOSSE}(s,m)} B_{s,m} \sum_{a\in[K]} \frac{K^{2/3}\cdot(s-t_\ell)^{2/3}\cdot\mathbf{1}\{S(I(s,m,a))>1\} + \sum_{i\in\text{PHASES}(I(s,m,a))} K^{2/3}\cdot L(I(s,m,a),i)^{2/3}}{\min_{t\in I(s,m,a)}|\mathcal{A}_t|}\right].$$

We can further upper bound $L([s, M(s, m, a)], i)$ by $L([s, s + m], i)$, the sum over phases $[\rho_i, \rho_{i+1})$ in PHASES$([s, M(s, m, a)])$ by a sum over PHASES$([s, s+m])$, and the $\mathbf{1}\{S(I(s, m, a) > 1\}$ by a $\mathbf{1}\{S([s, s+m]) > 1\}$ to obtain:

$$\mathbb{E}\left[\sum_{\text{BOSSE}(s,m)} B_{s,m} \sum_{a\in[K]} \frac{K^{2/3}\cdot(s-t_\ell)^{2/3}\cdot\mathbf{1}\{S([s,s+m])>1\} + \sum_{i\in\text{PHASES}([s,s+m])} K^{2/3}\cdot L([s,s+m],i)^{2/3}}{\min_{t\in I(s,m,a)}|\mathcal{A}_t|}\right].$$

Next, we note that $\sum_{a\in[K]}\frac{1}{\min_{t\in[s,M(s,m,a)]}|\mathcal{A}_t|} \leq \log(K)$ by previous arguments which turns the sum over $a \in [K]$ in the above display to a $\log(K)$ factor.

Now, define the function

$$G(s, m) := K^{2/3}\cdot(s-t_\ell)^{2/3}\cdot\mathbf{1}\{S([s,s+m])>1\} + \sum_{i\in\text{PHASES}([s,s+m])} K^{2/3}\cdot L([s,s+m],i)^{2/3}.$$

Then, following the same chain of arguments as in Appendix H.3, we have that it suffices to bound

$$\mathbb{E}\left[\sum_{s=t_\ell+1}^{t_{\ell+1}-1} \sum_{m} \frac{1}{m^{1/3}\cdot(s-t_\ell)^{2/3}}\cdot G(s,m)\right].$$

We split this up into two expectations based on the two terms in the definition of $G(s, m)$:

$$\mathbb{E}\left[\sum_{i\in\text{PHASES}([t_\ell,t_{\ell+1}))} \sum_{s=\rho_i}^{\rho_{i+1}-1} \sum_{m\geq\rho_{i+1}-s} \frac{1}{m^{1/3}}\right] + \mathbb{E}\left[\sum_{s=t_\ell}^{t_{\ell+1}-1} \sum_{m} \sum_{i\in\text{PHASES}([s,s+m])} \frac{L([s,s+m],i)^{2/3}}{m^{1/3}\cdot(s-t_\ell)^{2/3}}\right].$$

For the first expectation above, we have $\sum_{m\geq\rho_{i+1}-s} m^{-1/3} \leq \log(T)\cdot(\rho_{i+1}-s)^{-1/3}$. Then, summing over $s$, this becomes $\sum_{s=\rho_i}^{\rho_{i+1}-1}(\rho_{i+1}-s)^{-1/3} \leq (\rho_{i+1}-\rho_i)^{2/3}$. Thus, the first expectation in the above display is at most order $\log(T)\sum_{i\in\text{PHASES}([t_\ell,t_{\ell+1}))}(\rho_{i+1}-\rho_i)^{2/3}$.

For the second expectation, we use Jensen's inequality to bound

$$\sum_{i\in\text{PHASES}([s,s+m])} L([s,s+m],i)^{2/3} \leq m^{2/3}\cdot S([s,s+m])^{1/3},$$

where recall from Definition 11 that $S(I)$ counts the number of phases $[\rho_i, \rho_{i+1})$ intersecting interval $I$.

Now, plugging this into our earlier expectation gives

$$\mathbb{E}\left[\sum_{s=t_\ell}^{t_{\ell+1}-1} \sum_{m} \frac{m^{1/3}\cdot S([s,s+m])^{1/3}}{(s-t_\ell)^{2/3}}\right].$$

Then, coarsely bounding $S([s, s + m])$ by $S([t_\ell, t_{\ell+1}))$ and summing over $m$ and $s$, we obtain the above is at most order $\log(T)\cdot S([t_\ell, t_{\ell+1}))^{1/3}\cdot(t_{\ell+1}-t_\ell)^{2/3}$.

Combining the above steps, we obtain a per-episode regret bound of

$$\mathbb{E}\left[\sum_{a\in[K]} \sum_{t=t_\ell}^{t_{\ell+1}-1} \frac{\delta_t^{\mathbf{w}_t}(a_t^\sharp, a)}{|\mathcal{A}_t|}\cdot\mathbf{1}\{a\in\mathcal{A}_t\}\right] \leq$$

$$c_{19}\log(K)\log^3(T)K^{2/3}\mathbb{E}\left[\mathbf{1}\{\mathcal{E}_1\}\left(S([t_\ell,t_{\ell+1}))^{1/3}(t_{\ell+1}-t_\ell)^{2/3} + \sum_{i\in\text{PHASES}([t_\ell,t_{\ell+1}))}(\rho_{i+1}-\rho_i)^{2/3}\right)\right] + \frac{1}{T}.$$

$$(31)$$

Now, we will sum the above over episodes $[t_\ell, t_{\ell+1})$.

### I.5 Summing Regret Over Episodes

**In Terms of SUW.** We first show the total dynamic regret bound of order $K^{2/3} \cdot T^{2/3} \cdot \tilde{L}_{\text{Unknown}}^{1/3}$ in Theorem 14. By Hölder's inequality, summing over episodes gives:

$$\sum_{\ell=1}^{T} S([t_\ell, t_{\ell+1}))^{1/3} \cdot (t_{\ell+1} - t_\ell)^{2/3} \leq \left( \sum_{\ell=1}^{T} S([t_\ell, t_{\ell+1})) \right)^{1/3} \cdot T^{2/3} \leq (\tilde{L}_{\text{Unknown}} + \hat{L})^{1/3} \cdot T^{2/3},$$

where $\hat{L}$ represents the number of realized episodes by Algorithm 2 (i.e., episodes $[t_\ell, t_{\ell+1})$ where $t_\ell < T + 1$). Since Lemma 25 gives us that $\hat{L} \leq \tilde{L}_{\text{Unknown}}$, the above is of order $\tilde{L}_{\text{Unknown}}^{1/3} \cdot T^{2/3}$.

Next, again since Lemma 25 implies each phase intersects at most two episodes, we have that summing $\sum_{i \in \text{PHASES}([t_\ell, t_{\ell+1}))} (\rho_{i+1} - \rho_i)^{2/3}$ over $\ell$ gives an upper bound of $\tilde{L}_{\text{Unknown}}^{1/3} \cdot T^{2/3}$ by Jensens' inequality.

**Total Variation Regret Bound.** Next, we show the total variation regret bound of order $V_T^{1/4} \cdot T^{3/4} \cdot K^{1/2} + K^{2/3} \cdot T^{2/3}$. We'll follow a similar argument to that of Appendix H.5 for known weight, except taking care to handle the extra $S([t_\ell, t_{\ell+1})^{1/3} \cdot (t_{\ell+1} - t_\ell)^{2/3}$ term in (31).

We first bound the total variation $V_{[\rho_i, \rho_{i+1})}$ over an SUW phase $[\rho_i, \rho_{i+1})$. Consider the winner arm $a_{\rho_{i+1}}^*$. By Definition 2, there must exist a round $t \in [\rho_i, \rho_{i+1})$ and a weight $\mathbf{w} \in \Delta^K$ such that:

$$b_t(a_t^*, \mathbf{w}) - b_t(a_{\rho_{i+1}}^*, \mathbf{w}) > \left( \frac{K^2}{\rho_{i+1} - \rho_i} \right)^{1/3}.$$

This implies $V_{[\rho_i, \rho_{i+1})} \geq (K^2/(\rho_{i+1} - \rho_i))^{1/3}$. By an analogous argument to the SKW total variation regret analysis (with the only modification being the power of $K$), we have:

$$\sum_{i=0}^{\tilde{L}_{\text{Unknown}}} K^{2/3} \cdot (\rho_{i+1} - \rho_i)^{2/3} \leq V_T^{1/4} \cdot T^{3/4} \cdot K^{1/2} + K^{2/3} \cdot T^{2/3}.$$

Next, we bound the total variation $V_{[t_\ell, t_{\ell+1})}$ over an episode $[t_\ell, t_{\ell+1})$. Let $\tilde{S}([t_\ell, t_{\ell+1}))$ be the number of phases $[\rho_i, \rho_{i+1})$ properly contained in episode $[t_\ell, t_{\ell+1})$ or such that $[\rho_i, \rho_{i+1}) \subseteq [t_\ell, t_{\ell+1})$.

Then, we have the bound

$$V_{[t_\ell, t_{\ell+1})} \geq \sum_{i : [\rho_i, \rho_{i+1}) \subseteq [t_\ell, t_{\ell+1})} V_{[\rho_i, \rho_{i+1})} \geq \sum_{i : [\rho_i, \rho_{i+1}) \subseteq [t_\ell, t_{\ell+1})} \left( \frac{K^2}{\rho_{i+1} - \rho_i} \right)^{1/3}.$$

Next, we have that since $x \mapsto x^{-1/3}$ is convex, we have the above RHS can be further lower bounded by Jensen's inequality:

$$\sum_{i : [\rho_i, \rho_{i+1}) \subseteq [t_\ell, t_{\ell+1})} \left( \frac{K^2}{\rho_{i+1} - \rho_i} \right)^{1/3} \geq K^{2/3} \cdot \frac{\tilde{S}([t_\ell, t_{\ell+1}))^{4/3}}{(t_{\ell+1} - t_\ell)^{1/3}}.$$

Alternatively, we may also lower bound $V_{[t_\ell, t_{\ell+1})}$ by the same argument that we made for an SUW phase $[\rho_i, \rho_{i+1})$ since $[t_\ell, t_{\ell+1})$ is a period of rounds where every arm incurs significant regret in some period $[s_1, s_2] \subseteq [t_\ell, t_{\ell+1}]$ w.r.t. some weight $\mathbf{w}$. Thus, we also have

$$V_{[t_\ell, t_{\ell+1})} \geq \left( \frac{K^2}{t_{\ell+1} - t_\ell} \right)^{1/3}.$$

Now, by Hölder's inequality we have

$$\sum_{\ell=1}^{\hat{L}} K^{2/3} \cdot S([t_\ell, t_{\ell+1}))^{1/3} \cdot (t_{\ell+1} - t_\ell)^{2/3} \leq K^{1/2} \cdot T^{3/4} \cdot \left( \sum_{\ell=1}^{\hat{L}} K^{2/3} \frac{S([t_\ell, t_{\ell+1}))^{4/3}}{(t_{\ell+1} - t_\ell)^{1/3}} \right)^{1/4} + K^{2/3} \cdot T^{2/3}.$$

Upper bounding $S([t_\ell, t_{\ell+1}))^{4/3} \leq c_{20} \cdot (\tilde{S}([t_\ell, t_{\ell+1}))^{4/3} + 2^{4/3})$ using the inequality $(a+b)^{4/3} \leq 2 \cdot (a^{4/3} + b^{4/3})$, we can upper bound the above RHS by order:

$$K^{1/2} \cdot T^{3/4} \cdot \left( \sum_\ell V_{[t_\ell, t_{\ell+1})} \right)^{1/4} + K^{2/3} \cdot T^{2/3} \leq K^{1/2} \cdot T^{3/4} \cdot V_T^{1/4} + K^{2/3} \cdot T^{2/3}.$$

### I.6 Proof of Theorem 5

Although our main regret upper bound for CW (Theorem 5) does not directly follow from Theorem 14, the analysis will follow a nearly identical structure while substituting the SUW phases $[\rho_i, \rho_{i+1})$ with the approximate winner phases $[\zeta_i, \zeta_{i+1})$ (Definition 4).

First, we transform the Condorcet dynamic regret to a weighted Borda dynamic regret. Let $\mathbf{w}_t = \mathbf{w}(\tilde{a}_t)$ where $\tilde{a}_t$ is the approximate winner arm of the unique approximate winner phase $[\zeta, \zeta_{i+1})$ containing round $t$. Let $\zeta_{S_{\text{approx}}+1} \doteq T+1$. Now, the Condorcet dynamic regret may be re-written using Definition 4:

$$\mathbb{E}[\text{Regret}^{\text{C}}] = \mathbb{E}\left[ \sum_{t=1}^T \frac{1}{2} (P_t(a_t^{\text{C}}, i_t) + P_t(a_t^{\text{C}}, j_t) - 1) \right]$$

$$\leq \mathbb{E}\left[ \sum_{t=1}^T \frac{1}{2} (P_t(\tilde{a}_t, i_t) + P_t(\tilde{a}_t, j_t) - 1) \right] + \sum_{i=0}^{S_{\text{approx}}} K^{2/3} \cdot (\zeta_{i+1} - \zeta_i)^{2/3}$$

$$= \mathbb{E}\left[ \sum_{t=1}^T \frac{1}{2} (\delta_t(\tilde{a}_t, \tilde{a}_t) - \delta_t(i_t, \tilde{a}_t) + \delta_t(\tilde{a}_t, \tilde{a}_t) - \delta_t(j_t, \tilde{a}_t)) \right] + \sum_{i=0}^{S_{\text{approx}}} K^{2/3} \cdot (\zeta_{i+1} - \zeta_i)^{2/3}$$

$$\leq \mathbb{E}\left[ \frac{1}{2} \sum_{t=1}^T \delta_t^{\mathbf{w}_t}(i_t) + \delta_t^{\mathbf{w}_t}(j_t) \right] + \sum_{i=0}^{S_{\text{approx}}} K^{2/3} \cdot (\zeta_{i+1} - \zeta_i)^{2/3}.$$

It remains to bound the expectation on the above RHS by the sum (up to log terms) in the same display and then show

$$\sum_{i=0}^{S_{\text{approx}}} K^{2/3} \cdot (\zeta_{i+1} - \zeta_i)^{2/3} \leq \min \left\{ K^{2/3} T^{2/3} S_{\text{approx}}^{1/3}, K^{1/2} V_T^{1/4} T^{3/4} + K^{2/3} T^{2/3} \right\}.$$

The key facts will be crucial in showing these claims.

**Fact 29.** *An SUW cannot occur within an approximate winner phase. In other words, supposing $\tilde{a}_i$ is the approximate winner arm of phase $[\zeta_i, \zeta_{i+1})$ (such that (6) holds for $a = \tilde{a}_i$ and for all $s \in [\zeta_i, \zeta_{i+1}), a \in [K]$) then we must have that $\tilde{a}_i$ satisfies for all $[s_1, s_2] \subseteq [\zeta_i, \zeta_{i+1})$:*

$$\max_{\mathbf{w} \in \Delta^K} \sum_{s=s_1}^{s_2} \delta_s^{\mathbf{w}}(\tilde{a}_i) < K^{2/3} \cdot (s_2 - s_1)^{2/3}.$$

*Proof.* Note that (6), Jensen's inequality, and GIC (Condition 1) implies for any weight $\mathbf{w} \in \Delta^K$ and $s \in [\zeta_i, \zeta_{i+1})$:

$$\delta_s^{\mathbf{w}}(\tilde{a}_i) = |\delta_s^{\mathbf{w}}(a_s^*, \tilde{a}_i)| \leq \mathbb{E}_{a \sim \mathbf{w}}[|P_s(a_s^*, a) - P_s(\tilde{a}_i, a)|] \leq \left( \frac{K^2}{s - \zeta_i} \right)^{1/3}.$$

Thus, $\tilde{a}_i$ does not incur significant worst-case weighted Borda regret over any interval $[s_1, s_2] \subseteq [\zeta_i, \zeta_{i+1})$. $\square$

**Fact 30.** *The total variation in an approximate winner phase is at least*

$$V_{[\zeta_i, \zeta_{i+1})} \geq \left( \frac{K^2}{\zeta_{i+1} - \zeta_i} \right)^{1/3}.$$

*Proof.* Fix a phase $[\zeta_i, \zeta_{i+1})$ and consider the winner arm $a^*_{\zeta_{i+1}}$ at round $\zeta_{i+1}$. By Definition 4, there must exist a round $s \in [\zeta_i, \zeta_{i+1})$ such that for some arm $a \in [K]$:

$$\left(\frac{K^2}{\zeta_{i+1} - \zeta_i}\right)^{1/3} < \delta_s^{\mathbf{w}(a)}(a^*_s, a^*_{\zeta_{i+1}}) \leq \delta_s^{\mathbf{w}(a)}(a^*_s, a^*_{\zeta_{i+1}}) - \delta_{\zeta_{i+1}}^{\mathbf{w}(a)}(a^*_s, a_{\zeta_{i+1}}) \leq V_{[\zeta_i, \zeta_{i+1})},$$

where the second inequality follows from GIC. □

**Fact 31** (Analogue of Lemma 25)**.** *On event $\mathcal{E}_1$, for each episode $[t_\ell, t_{\ell+1})$ with $t_{\ell+1} \leq T$ (i.e., an episode which concludes with a restart), there exists an approximate winner change $\zeta_i \in [t_\ell, t_{\ell+1})$.*

*Proof.* This follows from Fact 29, as an SUW cannot occur within an approximate winner phase. This means that since a restart implies an SUW has occurred, an approximate winner change must have also occurred. □

Next, using Fact 31, we bound the regret $\mathbb{E}[\sum_{t=1}^{T} \delta_t^{\mathbf{w}_t}(i_t) + \delta_t^{\mathbf{w}_t}(j_t)]$ using the generic bad segment analysis of Appendix I.3 except replacing the SUW phases $[\rho_i, \rho_{i+1})$ with the approximate winner phases $[\zeta_i, \zeta_{i+1})$. Fact 29 ensures that the analogue of Lemma 28 will hold as the approximate winner arm $\tilde{a}_i$ of phase $\zeta_i$ cannot be evicted by a perfect replay corresponding to a generic bad segment in phase $[\zeta_i, \zeta_{i+1})$. Then, the proof steps of Appendix I.4 and Appendix I.5 follow mutatis mutandis while using Fact 30 to get the total variation bound.

## J  Additional Related Work

**Dueling bandits.**  The stochastic dueling bandit problem was first proposed by Yue & Joachims (2011); Yue et al. (2012), which provided an algorithm achieving an instance-dependent $O(K \log T)$ regret under the SST∩STI condition. Urvoy et al. (2013) studied this problem under the broader Condorcet winner condition and achieved an instance-dependent $O(K^2 \log T)$ regret bound, which was further improved by Zoghi et al. (2014) and Komiyama et al. (2015) to $O(K^2 + K \log T)$. Finally, Saha & Gaillard (2022) showed it is possible to achieve an optimal instance-dependent bound of $O(K \log T)$ and instance-independent bound of $O(\sqrt{KT})$ under the Condorcet condition. These works all assume a stationary environment.

Works on adversarial dueling bandits (Gajane et al., 2015; Saha et al., 2021) allow for changing preferences and thus are closer to this work. However, these works focus on the static regret objective against the 'best' arm in hindsight and whereas we consider the dynamic regret.

Other than the earlier mentioned works on dynamic regret minimization in dueling bandits, Kolpaczki et al. (2022) studies *weak dynamic regret* minimization but uses procedures requiring knowledge of non-stationarity.

**Borda Regret Minimization.**  The only works studying Borda regret (in stochastic or adversarial settings) are Saha et al. (2021); Saha & Gupta (2022); Wu et al. (2023). Of these, only Saha & Gupta (2022) establishes dynamic Borda regret bounds, which require knowledge of the underlying non-stationarity, and are suboptimal in light of our optimal regret bound (Theorem 2). Hilgendorf (2018) studies *weak Borda regret* where the learner only incurs the Borda regret of the better of the two arms paid.

**Other Notions of Winner.**  Other alternative notions of winner and objectives, beyond Condorcet and Borda, have been proposed, such as Copeland winner (Zoghi et al., 2015a; Komiyama et al., 2016; Wu & Liu, 2016) and von Neumann winner (Dudik et al., 2015; Balsubramani et al., 2016).

There have also been generalized notions of Borda (Brandt et al., 2022), and Condorcet winners (Agarwal et al., 2020; Haddenhorst et al., 2021) to combinatorial settings with subset comparisons.

Related notions to our *weighted Borda winner* (see Appendix A) appear in earlier social choice theory literature (Xia & Conitzer, 2008; Xia, 2013).

**Non-stationary multi-armed bandits.**  Switching multi-armed bandits with was first considered in the so-called adversarial setting by Auer et al. (2002), where a version of EXP3 was shown to attain optimal dynamic regret $\sqrt{LKT}$ when given knowledge of the number $L$ of changes in the rewards. Later works

showed similar guarantees in this problem for procedures inspired by stochastic bandit algorithms (Garivier & Moulines, 2011; Kocsis & Szepesvári, 2006). Recently, Auer et al. (2018; 2019); Chen et al. (2019) established the first adaptive and optimal dynamic regret guarantees, without requiring parameter knowledge of the number of changes.

Alternative characterizations of the change in rewards in terms of a total variation quantity was first introduced in Besbes et al. (2019) with minimax rates quantified therein and adaptive rates attained in Chen et al. (2019). There have also been characterizations of non-stationarity in terms of drift parameters (Jia et al., 2023; Krishnamurthy & Gopalan, 2021). Yet another characterization, in terms of the number of best arm switches $S$ was studied in Abbasi-Yadkori et al. (2023), establishing an adaptive regret rate of $\sqrt{SKT}$. Around the same time, Suk & Kpotufe (2022) introduced the aforementioned notion of *significant shifts* for the switching bandit problem and adaptively achieved rates of the form $\sqrt{\tilde{S}KT}$ in terms of $\tilde{S}$ significant shifts in the rewards, which serves as the inspiration for our Definition 1.

