# OpenReview forum: "Non-Stationary Dueling Bandits Under a Weighted Borda Criterion"
_TMLR — Accepted by TMLR_

### Review · Reviewer_VcZ5 · 2024-11-01

**Summary Of Contributions:**

The paper considers the problem of non-stationary dueling bandits where agent dealing with a bandit framework where the agent can play two arms and gets the feedback about which arm is better. An interesting contemporary setup where this framework could be useful is collecting feedback by presenting different LLM outputs to users and see which is the preferred output. For this framework the authors considered two parallel preference models of Borda and Condorcet and show their equivalence using a weighted Borda approach.

For the weighted Borda approach, the paper considers a non-stationary preference matrix and provide BOSSE algorithm which works by injecting uniform exploration controlled with time-dependent learning rate $\eta_t$. For the proposed algorithm, the authors derive regret bounds.

**Audience:**

Yes

**Broader Impact Concerns:**

The paper is theoretical and does not have broader impact concern.

**Claims And Evidence:**

Yes

**Requested Changes:**

Please address the weaknesses mentioned.

**Strengths And Weaknesses:**

Strengths:

1. I really appreciate the authors insight to combine the two notions of Borda regret and Condorcet regret. It is good works which unify different notions. With this the paper provides a new tool to analyze Condorcet models using Borda frameworks.

2. The paper provides an algorithm which works with non-stationary preference matrices by performing controlled exploration on all arms. The exploration is controlled using learning rate parameter $\eta_t$ which is time dependent. The regret analysis works by bounding the variation of the importance sampling estimator which gives insights into who to select the learning rate of the algorithm (Appendix E)


Weaknesses:
1. The paper is extremely difficult to parse.

   a . There are a lot of forward references, for example the regret of the algorithm described before defining the algorithm.

   b. The SST$\cap$STI condition is not clearly explained in the introduction again requiring to move forward. Even there the complete names should have been provided and some details/intuitions would have helped.

   c.  What is ordering of the arms, that is again not clearly defined.

2. Lack of motivation: It would be helpful if the authors explained why there exists two school of thoughts Borda/Condorcet best arms and where is one useful?

3. Some prototype algorithm implementation even on toy example would make the paper stronger.

---

> ### Author Response · Authors · 2024-11-14
>
> Thank you for the valuable feedback and careful review.
>
> > The paper is extremely difficult to parse.
>
> Thank you for pointing this out. In the rewrite, we will give intuition to define concepts and assumptions before they're formally introduced.
>
> > What is ordering of the arms, that is again not clearly defined.
>
> The ordering on arms is a common assumption in dueling bandits which says there exists a total ordering $\succ$ of arms such that $i \succ j$ implies $P(i,j) > 1/2$. See for instance Section 3.1 of "Preference-based Online Learning with Dueling Bandits: A Survey" (Bengs et al., 2020). We'll define this more thoroughly in the revision.
>
> > Lack of motivation: It would be helpful if the authors explained why there exists two school of thoughts Borda/Condorcet best arms and where is one useful?
>
> This goes back to social choice theory and different notions of voting winner in light of Arrow's impossibility theorem ("A Difficulty in the Concept of Social Welfare"; Arrow, 1950). The Borda winner may be preferred over the Condorcet winner because it always exists while the Condorcet winner may not exist (e.g., a cyclical ordering of three arms). On the other hand, the Borda winner may not always be a sensible choice either. For example, in ranking teams in a sports tournament using Borda scores, a team which achieves a large margin against lower-tier teams could be considered the Borda winner while having little chance of being a tournament winner due to another high-ranking team which has smaller average margins. Our novel notion of _weighted Borda winner_ (Section 5) is a generalized notion of winner which recovers both the Borda and Condorcet winners. Thus, our submission provides a new unified framework for addressing both the Borda and Condorcet situations. We'll include this discussion in the rewrite.
>
> > Some prototype algorithm implementation even on toy example would make the paper stronger.
>
> We view our contribution as being theoretical, answering open questions of adaptivity and achieving optimal regret for non-stationary Borda dueling bandits, rather than proposing a very practical algorithm.
>
> A practical state-of-the-art algorithm is left for future work. We mention here that a key challenge in doing a thorough experimentation is the high runtime complexity of detecting unknown changes (BOSSE requires $O(KT^2)$ computations) which is extensively discussed in the recent preprint [1]. We'll clarify this missing aspect as a limitation of our work meriting further exploration.
>
> [1] Li et al., 2024. "Diminishing Exploration: A Minimalist Approach to Piecewise Stationary Multi-Armed Bandits". ArXiv. https://arxiv.org/pdf/2410.05734

---

### Review · Reviewer_YQxQ · 2024-11-02

**Summary Of Contributions:**

This work studies the problem of non-stationary dueling bandits from the perspective of both Borda winner and Condorcet winner. For Borda winner, this work establishes the first optimal and adaptive dynamic regret upper bound. For the Condorect winner, this work introduces a new non-stationary measurement (less stringent than the previously studied one), under which an upper bound that improves the SOTA general performance is achieved. Both of these advances are originated from a unified weighted Borda framework and a novel soft elimination algorithm.

**Audience:**

Yes

**Claims And Evidence:**

Yes

**Requested Changes:**

Major (from weakness):
- Certain technical details in the main paper to facilitate readers to understand the key steps/main challenges of the proof. Currently, there are some verbal discussions, which I found useful, while I would encourage the authors to have a bit more technical details in the main paper.

- More discussions on the BOSSE algorithm. I would suggest the authors to provide more discussions on BOSSE to highlight its novelty/difference from the previously adopted schemes in non-stationary bandits.

Minor:
- Lower bounds can also be put in Table 1 for a complete view.

- I would love to hear the authors' comments on the possibility of doing some simulated experiments. While I understand the difficulty to do so due to the non-stationary environment and algorithmic complexity, I would lean to see some experiments. If not possible, please explicitly mention this as a limitation of this work worth further exploration.

**Strengths And Weaknesses:**

Strength:
- This work provides a comprehensive research on the problem of non-stationary dueling bandits, contributing to understanding on both Borda winner and Condorect winner, including both upper and lower bounds. The obtained results are of their importance based on my understanding: providing the first adaptive upper bound for Borda winner and a deep understanding on the non-stationary measurement for the Condorect winner.

- The writing of this work is well-organized. I had only basic knowledge of dueling bandits before, while I found the reading experience smooth and entertaining.

- The proposed unified view of Condorect and Borda regret, together with the BOSSE algorithmic framework, would be of independent interest based on my understanding. The unified view could contribute to a more strong connection between studies on these two targets in dueling bandits while the algorithm design can serve as an important base for further extensions.

Weakness:
- The main paper has omitted the majority of technical details, which is understandable. However, with that, the overall reading seems to lack of a deep dive into the key innovation/technical contributions. I conjecture that there would be some key steps (besides the unified view) for the upper and lower bound analysis.

- Similar to the first point, the algorithmic contributions of BOSSE is only touched very briefly in the main paper, which makes me hard to judge its contribution. While it does share similarity with the phased scheme in non-stationary and adversarial designs, I believe it holds its own merit that should be highlighted.

---

> ### Author Response · Authors · 2024-11-14
>
> Thank you for the encouraging comments and comprehensive review.
>
> > The main paper has omitted the majority of technical details, which is understandable. However, with that, the overall reading seems to lack of a deep dive into the key innovation/technical contributions. I conjecture that there would be some key steps (besides the unified view) for the upper and lower bound analysis.
>
> As the regret upper bound analysis is admittedly similar in structure to the prior works on non-stationary (dueling) bandits (Suk & Kpotufe, 2022; Buening & Saha, 2023; Suk & Agarwal, 2023), we dedicated much of the body of the paper to highlighting novelties of this submission. In particular, the last four paragraphs of Section 6 highlight the novelties of algorithm design and Section 7 discusses the differences/challenges in the regret analysis. In the revision, we'll dedicate more space to reviewing the requisite components of the non-stationary regret analysis from prior works so that readers can better digest the novelties.
>
> The lower bound regret analyses are more standard and follow the techniques of the prior works (Saha & Gaillard 2022; Saha et al., 2021). We'll clarify this as well.
>
> > Similar to the first point, the algorithmic contributions of BOSSE is only touched very briefly in the main paper, which makes me hard to judge its contribution. While it does share similarity with the phased scheme in non-stationary and adversarial designs, I believe it holds its own merit that should be highlighted.
>
> Again, we'll further elaborate on the algorithmic novelties at the end of Section 6. We now believe doing a more expansive review of the general strategy for tracking significant shifts from prior works will help clarify the novelties of this work for general readers.
>
> > Certain technical details in the main paper to facilitate readers to understand the key steps/main challenges of the proof. Currently, there are some verbal discussions, which I found useful, while I would encourage the authors to have a bit more technical details in the main paper.
>
> As in our previous answers, we'll also include a more expansive review of the standard non-stationary regret analysis originally developed in Suk & Kpotufe 2022 which we believe will better highlight the novelties/challenges required for this work.
>
> > I would love to hear the authors' comments on the possibility of doing some simulated experiments. While I understand the difficulty to do so due to the non-stationary environment and algorithmic complexity, I would lean to see some experiments. If not possible, please explicitly mention this as a limitation of this work worth further exploration.
>
> We view the main contribution as being theoretical, rather than proposing a very practical algorithm. The algorithm in our paper is of a theoretical nature that serves to answer the main theoretical questions: (1) for the Borda dueling bandit setting, it remained open whether the the optimal dynamic regret could be achieved (even with parameter knowledge) and (2) following the works of Buening & Saha, '23 and Suk & Agarwal, '23, it remained open whether tighter rates than $K\sqrt{S_C T}$ could be achieved in the Condorcet setting outside of $\text{SST} \cap \text{STI}$. We have fully resolved the first question and achieved non-trivial progress toward the second question.
>
> We agree with you and Reviewer VcZ5 that there's still important work to be done in designing practical procedures, and we admit our theoretical state-of-the-art is far from this. We note a key challenge in doing thorough experimentation is the high runtime complexity of detecting unknown changes (our algorithm requires $O(KT^2)$ computations) which is extensively discussed in a recent preprint [1]. We'll clarify this as a limitation of our work meriting further exploration.
>
> [1] Li et al., 2024. "Diminishing Exploration: A Minimalist Approach to Piecewise Stationary Multi-Armed Bandits". ArXiv. https://arxiv.org/pdf/2410.05734

---

### Review · Reviewer_meZc · 2024-11-02

**Summary Of Contributions:**

This paper considers the instance-independent bound for non-stationary dueling bandits under the Condorcet winner and Borda winner setup. It establishes the a dynamic regret upper bound, which depends on the (unknown) number of significant switches. Another contribution is a weighted version of the Borda score. The winner under the weighted Borda score can particularize to the Condorcet winner and Borda winner with the point-distribution and uniform distribution, respectively.

**Audience:**

Yes

**Broader Impact Concerns:**

None.

**Claims And Evidence:**

Yes

**Requested Changes:**

Please see the weaknesses and questions section. In particular, the definitions of the SBS phases and approximate winner phases need more discussions and justifications.

**Strengths And Weaknesses:**

**Strengths**
- The introduction of the significant Borda Winner switches (SBS) is meaningful, as there are lots of works in the nonstationary K-armed bandits literature who quantify the level of nonstationarity as the number of big distribution switches. The proposed notion, SBS, fills the gap in the nonstationary dueling bandits within the Borda winner setup.
- The proposed General Identifiability Condition further generalizes the SST and STI conditions presented in the Condorcet winner literature. Under this relaxed condition, the final upper bound is more refined and is superior to the previous ones under certain conditions.

**Weaknesses and Questions**

Weaknesses:
- The intuition for the definition of the SBS phase/Significant Borda regret, and approximate winner phase is not provided in detail.
- The usefulness of the approximate winner arm needs to be justified by concrete examples.

Questions:
- While the introduction of SBS is desired, the definition of the significant Borda regret is not clear to me.
	- According to equation (2), if the Borda winner gradually decrease the winning probability and the rest suboptimal arms gradually increase the winning probability such that (2) will not hold for any arm, then it may happen that the Borda winner has changed but there is not any SBS over the whole time horizon. Is this property desired?
- In Definition 4, is this requirement really useful?
	- When the top two arms have close winning probabilities against the rest arms and the winning probability of the top two arms are swapped frequently. It seems that although the Condorcet winer changes, the number of approximate winner phases does not increase.
	- As indicated by equations (3) and (4), the number of approximate winner phases is small only when the number of arms is large, which can even be linear in $T$.
Therefore, I am not perfectly convinced by the utility of this definition.
- Can the authors provide some insights for the choice of the threshold, in particular the orders, Definition 1 and 4? It seems that these orders are picked so that the expected regret in each phase can match that in the stationary setup.

---

> ### Author Response · Authors · 2024-11-14
>
> Thank you for the detailed review and comments.
>
> > Intuition for definition of the SBS phase/Significant Borda regret, approximate winner phase.
>
> Definition 1 is similar to the notion of _significant shift_ in the literature for non-stationary MAB (Suk & Kpotufe, 2022) and Condorcet dueling bandits (Buening and Saha, 2023; Suk and Agarwal, 2023). The key difference here is that significant shift is defined through the Borda notion of regret. Intuitively, for any notion of regret and minimax rate ($T^{2/3}$ for the Borda setting), one can track when each arm accrues significant regret (i.e., surpassing the minimax rate) and define a _significant shift_ as the first round when every arm has accrued significant regret.
>
> On an approximate winner phase, there is a fixed _approximate winner arm_ which exhibits similar preference behavior as the true winner, up to a tolerance decided by our minimax regret rate. An approximate winner change occurs when there is no such stable approximate winner.
>
> > The usefulness of the approximate winner arm needs to be justified by concrete examples.
>
> Following the thread of our previous answer, it may be the case that the winner arm changes every round but an approximate winner remains stable. For instance, consider a non-stationary environment with three arms $1,2,3$ initially ordered as $1 \succ 2 \succ 3$ and the gap $\delta_1(1,3) = 1/2$ is large but the gaps $\delta_1(1,2) , \delta_1(2,3) = o(1/T^{1/3})$ are small, with the identities of arms $1,3$ switching every round. Then, arm $2$ is a stable approximate winner (hence, $S_{\text{approx}}=1$) while the actual winner switches every round between arms $1$ and $3$ (number of winner switches $S_C=\Omega(T)$).
>
> Note here that previous upper bounds in terms of number of winner switches $S_C$ would be pessimistic for this environment, meaning the approximate winner arm is a more apt measure of non-stationarity.
>
> > According to equation (2), if the Borda winner gradually decrease the winning probability and the rest suboptimal arms gradually increase the winning probability such that (2) will not hold for any arm, then it may happen that the Borda winner has changed but there is not any SBS over the whole time horizon. Is this property desired?
>
> Your observation is correct and in fact this property is desired in our definition. To use an example, consider the 3-arm example in our previous answer. We see that not all changes in Borda winner may be _significant_ to performance and truly require restarting so long as there is a stable safe arm (arm $2$ in the example) that we can track. In Definition 1, arm 2 is the safe arm which does not satisfy (2) and hence an SBS will not be triggered. This is sensible because we can still maintain stationary $T^{2/3}$ regret by learning arm $2$ in this non-stationary environment.
>
> > When the top two arms have close winning probabilities against the rest arms and the winning probability of the top two arms are swapped frequently. It seems that although the Condorcet winer changes, the number of approximate winner phases does not increase.
>
> This observation is correct and in line with our previous answers above. To reiterate, the **benefit** of our approximate winner changes (Defn. 4) is that it does _not_ overcount superfluous changes which do not impede learning. Just as in the Borda case, not all changes in Condorcet winner may actually require a restart or need to be counted in the regret bound. Thus, our notion seeks to only count those changes from which we cannot recover.
>
> > As indicated by equations (3) and (4), the number of approximate winner phases is small only when the number of arms is large, which can even be linear in $T$. Therefore, I am not perfectly convinced by the utility of this definition.
>
> It's **not true** that "the number of approximate winner phases is small only when the number of arms is large". The number of arms can be small and there can still be one approximate winner phase, e.g. in a stationary environment with no changes (in such a case Eq. (3) will always be satisfied by the fixed winner regardless of the value of $K$). Note also that Eq. (4) is not a condition on the number of approximate winner phases, but a description of the regimes in which we recover/improve the state-of-art Condorcet regret rate of Buening and Saha, 2022.
>
> > Can the authors provide some insights for the choice of the threshold, in particular the orders, Definition 1 and 4? It seems that these orders are picked so that the expected regret in each phase can match that in the stationary setup.
>
> Indeed, you're correct that the threshold of Definition 1 is chosen to match the minimax stationary regret rate of $K^{1/3} T^{2/3}$ for the Borda setup (Definition 1). Definition 4's threshold is likewise chosen to match the minimax regret rate for the stationary _weighted Borda problem with unknown weights_ (Theorem 11).
>
> We'll elaborate and incorporate these answers in the future revision.

---

> > ### Comment · Reviewer_meZc · 2024-11-18
> >
> > Great thanks to the authors for the clarifications! Most of my questions have been addressed.
> >
> >  I have two more questions here:
> >
> > - Thanks for correcting the misunderstanding of equations (3) and (4). Actually, I am concerned about the utility of this definition, i.e., when the proposed definition is useful. While the proposed definition shows better performance under the two regimes given below equation (4), $K$ is required to be $\Theta(T)$ in some common (or benign) regimes where $S=o(T)$. Under such cases, the proposed definition seems to yield worse regret bounds.
> > - The proposed definitions of significant changes are designed to achieve improved instance-independent regret bounds. However, since there are researchers in the bandits community who are interested in instance-dependent regret bounds, people may wonder if the proposed definitions remain useful in that context. Could the authors kindly comment on this?

---

> > > ### Author Response · Authors · 2024-11-19
> > > **Response to Followup Questions**
> > >
> > > Thank you for your followup questions.
> > >
> > > **About Regimes where we have Superior Regret**: we believe the way (4) is written may have caused some confusion, for which we apologize. We mean to say that our regret rate is tighter than the $K\sqrt{ST}$ rate of Buening and Saha, 2022 (Corollary 3.1) when either $K^{1/3} \geq \frac{T^{1/6} S_{\text{approx}}^{1/3}}{S^{1/2}}$ or $K \geq \frac{T^{3/2} V_T^{1/3}}{S} \vee \frac{T^{1/2}}{S^{3/2}}$. Another way to state this is that we have a tighter regret rate when
> > > $$
> > > K \geq \min\left( \frac{T^{1/2} S_{\text{approx}} }{S^{3/2}} , \max\left( \frac{T^{3/2} V_T^{1/2} }{S} , \frac{T^{1/2} }{S^{3/2}} \right) \right)
> > > $$
> > > From this, it is **not** true that $K$ must be $\Theta(T)$ if $S=o(T)$ in order to get a better performance for our rate. First, note that if $K = \Omega(T^{1/2})$ then Buening & Saha (2022)'s $K\sqrt{ST}$ rate is already linear so that our regret rate is never worse regardless of the value of $S$ or other parameters.
> > >
> > > For $K = o(T^{1/2})$ we see from the above inequality that we still have better regret if $S$ is large relative to $S_{\text{approx}}$ or large relative to $V_T$.
> > >
> > > **Instance-Dependent Regret Bounds**: this is a great question and highly depends on what is meant by "instance-dependent regret bound". In classical stochastic bandits, our understanding is that this refers to bounds of the form $\frac{K\log(T)}{\Delta_{\min}}$ for minimal gap $\Delta_{\min}$. Unfortunately, it's well known in non-stationary bandits that such bounds cannot be achieved in changing environments (when summed over different stationary environments) adaptively without knowledge of non-stationarity [1, Theorem 31.2]. We believe such an impossibility result holds for dueling bandits as well. Thus, it remains open how best to define "instance-dependent regret rates" for the non-stationary problem, even in non-dueling bandits.
> > >
> > > We'll include discussion reflecting these answers in the revision.
> > >
> > > [1] Tor Lattimore and Csaba Szepesvári. Bandit Algoritms. 2020.

---

### Decision · Action_Editor_T4yT · 2024-12-05

**Recommendation:** Accept as is

**Comment:**

The setting and the results are novel. Together with the reviewers, I believe this contribution would be a good addition to the literature on dueling bandits.

**Audience:**

Yes. The bandit community would be interested in this work as it combines several interesting elements --- non-stationarity, dueling, etc.

**Claims And Evidence:**

The claims made in this paper are valid.